# PriorGuide: Test-Time Prior Adaptation for Simulation-Based Inference

**Yang Yang**[1], **Severi Rissanen**[2,3], **Paul E. Chang**[1,4], **Nasrulloh Loka**[1], **Daolang Huang**[2,3], **Arno Solin**[2,3], **Markus Heinonen**[3], **Luigi Acerbi**[1]

[1]Department of Computer Science, University of Helsinki, Finland      [2]ELLIS Institute Finland
[3]Department of Computer Science, Aalto University, Finland      [4]DataCrunch
`{yang.yang,paul.chang,nasrulloh.satrio,luigi.acerbi}@helsinki.fi`
`{severi.rissanen,daolang.huang,arno.solin,markus.o.heinonen}@aalto.fi`

## Abstract

Amortized simulator-based inference offers a powerful framework for tackling Bayesian inference in computational fields such as engineering or neuroscience, increasingly leveraging modern generative methods like diffusion models to map observed data to model parameters or future predictions. These approaches yield posterior or posterior-predictive samples for new datasets without requiring further simulator calls after training on simulated parameter-data pairs. However, their applicability is often limited by the prior distribution(s) used to generate model parameters during this training phase. To overcome this constraint, we introduce *PriorGuide*, a technique specifically designed for diffusion-based amortized inference methods. PriorGuide leverages a novel guidance approximation that enables flexible adaptation of the trained diffusion model to new priors at test time, crucially without costly retraining. This allows users to readily incorporate updated information or expert knowledge post-training, enhancing the versatility of pre-trained inference models.

## 1 Introduction

Simulation-based inference (SBI) has become a key tool across scientific disciplines, enabling Bayesian inference for complex systems where the likelihood function $p(\mathbf{x} \mid \boldsymbol{\theta})$ for data $\mathbf{x}$ and model parameters $\boldsymbol{\theta}$ is intractable, but one can easily simulate from the forward model $\mathbf{x} \sim p(\mathbf{x} \mid \boldsymbol{\theta})$ (Cranmer et al., 2020). Within the Bayesian paradigm, prior beliefs about parameters $\boldsymbol{\theta}$ are updated with observed data $\mathbf{x}$ to form a posterior distribution $p(\boldsymbol{\theta} \mid \mathbf{x})$ (Gelman et al., 2013). While traditional methods like Markov Chain Monte Carlo (MCMC; Robert, 2007) are effective when likelihoods are available, recent *amortized inference* methods can directly learn the inverse mapping $\mathbf{x} \mapsto p(\boldsymbol{\theta} \mid \mathbf{x})$ from observations to posteriors using neural networks (Lueckmann et al., 2017; Radev et al., 2020). These amortized approaches, once trained, can rapidly infer posterior (parameters) or posterior-predictive (data) distributions given new observations, significantly speeding up the inference process.

Modern generative models, including diffusion models (Sohl-Dickstein et al., 2015; Ho et al., 2020; Song et al., 2021), transformers (Vaswani et al., 2017), and flow-matching techniques (Lipman et al., 2023), have demonstrated state-of-the-art performance in tackling these inverse modeling tasks for amortized SBI (Müller et al., 2022; Wildberger et al., 2024; Schmitt et al., 2024; Gloeckler et al., 2024; Chang et al., 2025; Whittle et al., 2025; Mittal et al., 2025; Hollmann et al., 2025). These models are trained on vast numbers of simulated parameter-data pairs $(\boldsymbol{\theta}, \mathbf{x})$, often drawing parameters $\boldsymbol{\theta}$ from a broad, uniform prior distribution to ensure comprehensive coverage of the parameter space.

However, this reliance on a fixed *training prior* introduces significant limitations. Practitioners often possess domain knowledge that, if incorporated as a more specific prior, could substantially improve inference accuracy. Moreover, *prior sensitivity analysis*—a crucial step for assessing the robustness of scientific conclusions to modeling assumptions—becomes cumbersome. This practice is vital across many disciplines, *e.g.*, economists must validate the policy implications of macroeconomic models against different theoretical priors (Del Negro & Schorfheide, 2008), climate scientists need to validate the climate sensitivity estimates over multiple sets of assumptions (Sherwood et al., 2020), and epidemiologists need to assess the sensitivity of pandemic forecasts to assumptions

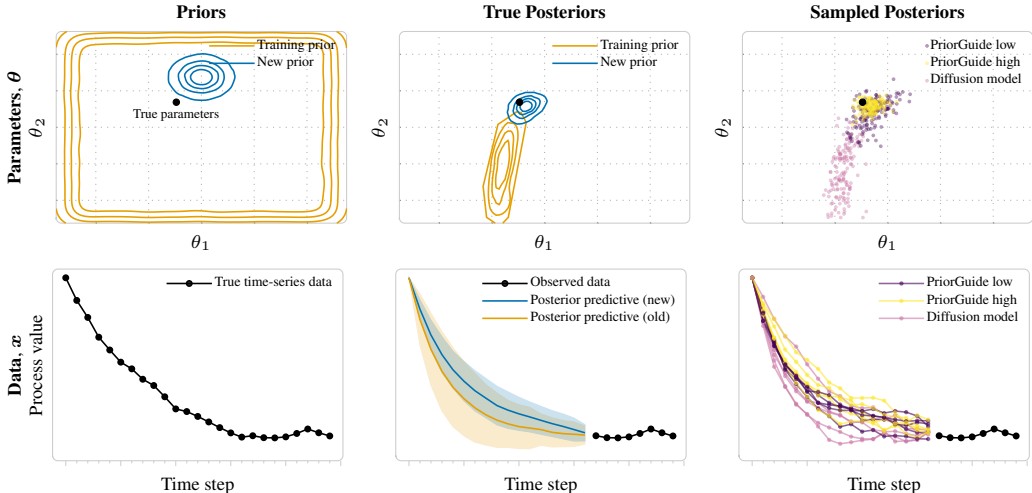

Figure 1: PriorGuide adapts a diffusion model to new prior information at test time for simulator-based inference—here for a time-series model. *Left:* Original broad **training prior** and a new, more specific **target prior**. *Middle:* Corresponding true posterior distributions over parameters (↑) and predictive data (↓) for each prior, given observed data. *Right:* Posterior (↑) and posterior-predictive (↓) samples from the diffusion model trained on the old prior vs. those from PriorGuide at *low* or *high* test-time cost, illustrating PriorGuide's ability to match the *new* posterior from the middle column.

about disease transmission (Flaxman et al., 2020). Changing priors is computationally challenging: non-amortized methods require costly re-simulation for each new prior, while amortized methods require retraining (Elsemüller et al., 2024). Though practitioners often resort to approximations like importance sampling to avoid these costs, such methods fail when priors differ substantially. This limitation becomes increasingly problematic as the field trends towards *foundation models* for SBI (Hollmann et al., 2025; Vetter et al., 2025). While some recent amortized methods offer inference-time prior adaptation, they are often restricted to specific families of priors (*e.g.*, factorized histograms or Gaussian mixtures pre-defined at training) or simple constraints (Elsemüller et al., 2024; Chang et al., 2025; Whittle et al., 2025; Gloeckler et al., 2024). The broader issue is the impracticality of pre-training over all potential tasks, such as all prior distributions a user might wish to employ. Recent successes of the *test-time compute* paradigm (Snell et al., 2025) suggest that rather than attempting exhaustive amortization for all scenarios, models could be designed to flexibly incorporate specific requirements, such as a user-defined prior, through dedicated computations at inference time—an ability which was unattained so far. For a comprehensive discussion of the related work, we refer the reader to Appendix A.1.

In this paper, we introduce PriorGuide, a novel method that empowers diffusion-based amortized inference models with the ability to adapt to new prior distributions $q(\boldsymbol{\theta})$ at inference time, without the need for retraining the original score model trained under prior $p_{\text{train}}(\boldsymbol{\theta})$.[1] PriorGuide leverages the guidance mechanisms inherent in diffusion models to steer the generative process according to the new target prior, seamlessly integrating new user-provided information during the sampling process. Crucially, this approach allows for a trade-off: users can invest more computational resources at inference time—such as more diffusion steps or adding refinement techniques like Langevin dynamics—to achieve higher inference fidelity. See Fig. 1 for a conceptual overview.

**Contributions** Our main contribution is a principled framework for flexibly incorporating new prior distributions at inference time into pre-trained diffusion models for SBI. We leverage a novel Gaussian mixture model approximation for effectively turning the target prior $q(\boldsymbol{\theta})$ into a tractable guidance term for the diffusion sampling process. We demonstrate empirically the effectiveness of PriorGuide on a range of SBI problems, showing its ability to accurately recover posterior and posterior-predictive distributions under various inference-time prior specifications. We show how sampling can be refined with additional Langevin dynamics steps, and study the tradeoff between test-time compute and inference accuracy within our framework.

---

[1]The intuitive requirement, quantified later, is that $q(\boldsymbol{\theta})$ should reside in regions of non-negligible mass under $p_{\text{train}}(\boldsymbol{\theta})$, to avoid out-of-distribution regions where the original score model would be poorly trained.

## 2 BACKGROUND

The primary goal in many scientific applications is to infer model parameters $\boldsymbol{\theta}$ given observed data $\mathbf{x}$, or to predict future data $\mathbf{x}^\star$. Bayesian inference provides a framework for computing the *posterior* distribution over parameters or *posterior predictive* distribution for new data:

$$p(\boldsymbol{\theta}\,|\,\mathbf{x}) \propto p(\mathbf{x}\,|\,\boldsymbol{\theta})\,p(\boldsymbol{\theta}), \qquad \text{(posterior)}$$

$$p(\mathbf{x}^\star\,|\,\mathbf{x}) = \int p(\mathbf{x}^\star\,|\,\boldsymbol{\theta},\mathbf{x})\,p(\boldsymbol{\theta}\,|\,\mathbf{x})\,\mathrm{d}\boldsymbol{\theta}, \quad \text{(posterior predictive)} \qquad (1)$$

where $p(\boldsymbol{\theta})$ is the prior and $p(\mathbf{x}\,|\,\boldsymbol{\theta})$ is the likelihood.

The *prior* $p(\boldsymbol{\theta})$ encodes the practitioner's beliefs about plausible parameter values *before* observing data—beliefs informed by physical constraints, previous experiments, or theoretical considerations (Gelman et al., 2013). In practice, priors are typically simple—Gaussians, uniforms, smooth distributions—reflecting broad domain knowledge rather than precise specifications. Yet even among simple priors, the specific choice varies by application: different analyses, constraints, or domain knowledge call for different specifications, and modern statistical practice recommends validation of results under multiple prior choices (Gelman et al., 2020).

The *likelihood* $p(\mathbf{x}\,|\,\boldsymbol{\theta})$ encodes the statistical or mechanistic description of how data are generated from the model. In many real-world scenarios from finance to physics, *evaluating* the likelihood is intractable, but *generating* samples $\mathbf{x} \sim p(\mathbf{x}\,|\,\boldsymbol{\theta})$ from a simulator is feasible, leading to the field of Simulation-Based Inference (SBI; Cranmer et al., 2020).

A powerful paradigm within SBI is *amortized inference*. Instead of performing inference from scratch for each $\mathbf{x}$, amortized methods train a neural network $q_\phi$ once on a large dataset of simulated parameter-data pairs, $\mathcal{D}_{\mathrm{sim}} = \{(\boldsymbol{\theta}_i, \mathbf{x}_i)\}_{i=1}^N$. Parameters $\boldsymbol{\theta}_i$ are drawn from a *training prior*, $p_{\mathrm{train}}(\boldsymbol{\theta})$, and data $\mathbf{x}_i \sim p(\mathbf{x}\,|\,\boldsymbol{\theta}_i)$. Once trained, $q_\phi$ provides rapid inference for new observations, amortizing the upfront computational cost. The network $q_\phi$ is usually trained to approximate the posterior $p(\boldsymbol{\theta}\,|\,\mathbf{x})$ (Lueckmann et al., 2017; Greenberg et al., 2019; Radev et al., 2020), the likelihood $p(\mathbf{x}\,|\,\boldsymbol{\theta})$ (Papamakarios et al., 2019), or the posterior predictive distribution $p(\mathbf{x}^\star\,|\,\mathbf{x})$ (Garnelo et al., 2018; Müller et al., 2022). Recently proposed architectures can flexibly perform *all* of these tasks, using transformers (Chang et al., 2025) or diffusion models (Gloeckler et al., 2024).

### 2.1 DIFFUSION MODELS

Diffusion models are a powerful framework for generative modeling that transforms samples from arbitrary to simple distributions and vice versa through a gradual noising and denoising process (Sohl-Dickstein et al., 2015). In the forward diffusion process, starting from a distribution $p(\mathbf{z}_0)$ we can draw samples from (*e.g.*, joint samples from the training prior and simulator), Gaussian noise is progressively added to the samples until, at the end of the process ($t = 1$), the distribution converges to a simple terminal distribution, typically Gaussian. In the Variance Exploding (VE) formulation (Song et al., 2021; Karras et al., 2022), the forward diffusion process can be described as:

$$p(\mathbf{z}_t) = \int p(\mathbf{z}_t\,|\,\mathbf{z}_0)\,p(\mathbf{z}_0)\,\mathrm{d}\mathbf{z}_0 = \int \mathcal{N}(\mathbf{z}_t\,|\,\mathbf{z}_0, \sigma(t)^2\mathbf{I})\,p(\mathbf{z}_0)\,\mathrm{d}\mathbf{z}_0, \qquad (2)$$

where $p(\mathbf{z}_t\,|\,\mathbf{z}_0)$ is the transition kernel (here Gaussian), $\sigma(t)^2$ defines the noise variance schedule as a function of time (typically increasing with $t$), and $\mathbf{z}_t$ represents the noisy samples at time $t$. The corresponding reverse process reconstructs the original sample distribution from noise, and can be formulated as either a stochastic differential equation (SDE) or an ordinary differential equation (ODE). The reverse SDE process takes the form (Song et al., 2021; Karras et al., 2022):

$$\mathrm{d}\mathbf{z}_t = -2\dot{\sigma}(t)\sigma(t)\nabla_{\mathbf{z}}\log p(\mathbf{z}_t)\,\mathrm{d}t + \sqrt{2\dot{\sigma}(t)\sigma(t)}\,\mathrm{d}\omega_t, \qquad (3)$$

where $\nabla_{\mathbf{z}}\log p(\mathbf{z}_t)$ is the *score function* (gradient of the log-density), $\mathrm{d}\omega_t$ is a Wiener process representing Brownian motion (noise), and $\dot{\sigma}(t)$ is the time derivative of the noise schedule.

**Learning the score function** The score function $\nabla_{\mathbf{z}}\log p(\mathbf{z}_t)$ can be approximated using a neural network $s(\mathbf{z}_t, t)$, trained to minimize the denoising score matching loss (Hyvärinen & Dayan, 2005; Vincent, 2011; Song et al., 2021):

$$\mathcal{L}_{\mathrm{DSM}} = \mathbb{E}_{t \sim p(t)}\mathbb{E}_{\mathbf{z}_0 \sim p(\mathbf{z}_0)}\mathbb{E}_{\mathbf{z}_t \sim \mathcal{N}(\mathbf{z}_t\,|\,\mathbf{z}_0, \sigma(t)^2\mathbf{I})}\left[\omega(t)\,\|s(\mathbf{z}_t, t) - \nabla_{\mathbf{z}_t}\log p(\mathbf{z}_t\,|\,\mathbf{z}_0)\|_2^2\right]. \qquad (4)$$

Here $p(t)$ is the distribution of noise levels sampled during training and $\omega(t)$ weights different noise levels in the loss. Once trained, the network $s(\mathbf{z}_t, t)$ approximates the gradient of the log-probability density of noised distributions and affords sampling through the reverse SDE; Eq. (3). Starting from a sample $\mathbf{z}_t \sim \mathcal{N}(\mathbf{z}_t \mid \mathbf{z}_0, \sigma_{\max}^2 \mathbf{I})$ for $t = 1$ with sufficiently large $\sigma_{\max}$, integrating the reverse process backward in time approximately reconstructs the original distribution $p(\mathbf{z}_0)$.

The diffusion framework's flexibility stems largely from its ability to incorporate *guidance mechanisms*, which afford steering the sampling process toward desired outcomes by including additional information or constraints. Notable examples include classifier guidance (Dhariwal & Nichol, 2021) and classifier-free guidance (Ho & Salimans, 2022), which enable variable-strength conditioned generation. For inverse problems, guidance methods exist to incorporate information about observations (Chung et al., 2023; Song et al., 2023a).

## 2.2 DIFFUSION-BASED AMORTIZED SBI

Modern amortized SBI methods leverage highly expressive generative models for multiple inference tasks. Early applications of diffusion models to SBI include Geffner et al. (2023), who demonstrated that learning conditional score functions via denoising score matching offers superior stability and sample quality compared to traditional flow-based baselines. Furthermore, Simformer (Gloeckler et al., 2024) trains a diffusion model on samples from the joint distribution $p(\boldsymbol{\theta}, \mathbf{x}) = p_{\text{train}}(\boldsymbol{\theta}) \, p(\mathbf{x} \mid \boldsymbol{\theta})$. Simformer employs a transformer architecture to model the score function $s_\phi(\boldsymbol{\xi}_t, t, \text{mask})$ of the noised joint variable $\boldsymbol{\xi}_t = (\boldsymbol{\theta}_t, \mathbf{x}_t)$ at diffusion time $t$. The mask specifies which components of $\boldsymbol{\xi}$ are conditioned upon and which are to be generated. By setting the mask appropriately (*e.g.*, conditioning on $\mathbf{x}$ to generate $\boldsymbol{\theta}$), Simformer can sample from various conditionals, including the posterior $p(\boldsymbol{\theta} \mid \mathbf{x})$ and posterior predictive $p(\mathbf{x}^\star \mid \mathbf{x})$. Crucially, these learned conditionals are implicitly tied to the training prior $p_{\text{train}}(\boldsymbol{\theta})$ used to generate the training data—applying a different prior $q(\boldsymbol{\theta})$ would require retraining the diffusion model with the new prior.

## 2.3 PRIOR ADAPTATION IN AMORTIZED SBI

Standard amortized SBI methods tie the learned posterior $q_\phi(\boldsymbol{\theta} \mid \mathbf{x})$ to the fixed prior $p_{\text{train}}(\boldsymbol{\theta})$ used during training. Changing this prior traditionally requires retraining the model from scratch, which is computationally prohibitive for expensive simulators. A few recent amortized methods achieve prior flexibility by training over a *meta-prior*—a distribution or predefined set of possible prior specifications—to learn how to incorporate different prior information at runtime. For instance, the Amortized Conditioning Engine (ACE; Chang et al., 2025) allows users to specify factorized priors at runtime by encoding each one-dimensional prior density as a normalized histogram over a predefined grid; its transformer architecture is trained to process these specific histogram-based prior encodings alongside observed data. Similarly, the Distribution Transformer (DT; Whittle et al., 2025) learns a direct mapping from a prior, itself represented as a Gaussian mixture model (GMM), to a GMM posterior, using attention mechanisms to transform the prior components based on the data. Sensitivity-aware SBI (Elsemüller et al., 2024) focuses on providing efficient sensitivity analysis to various modeling choices, including different priors, represented by a discrete set of possible alternative prior specifications. All of these methods enable prior adaptation by relying on their pre-training across a chosen meta-prior. Our approach, described next, sidesteps this requirement by preforming purely inference-time adaptation to flexibly handle new target priors.

## 3 PRIORGUIDE

PriorGuide offers a solution to take a diffusion-based amortized SBI model such as Simformer (Gloeckler et al., 2024), trained on a training prior $p_{\text{train}}(\boldsymbol{\theta})$, and adapt it to a new target prior $q(\boldsymbol{\theta})$ at inference time, *without retraining*. The objective is to perform standard SBI tasks such as sampling from the posterior or posterior predictive distribution *under the new prior*, that is $q(\boldsymbol{\theta} \mid \mathbf{x}) \propto p(\mathbf{x} \mid \boldsymbol{\theta})q(\boldsymbol{\theta})$ or $q(\mathbf{x}^* \mid \mathbf{x})$, respectively. The method achieves this by adjusting the score guidance during the diffusion sampling process. The key relationship for this adaptation is as follows:

**Proposition 1.** *Let the posterior under the original prior be $p(\boldsymbol{\theta} \mid \mathbf{x}) \propto p_{train}(\boldsymbol{\theta})p(\mathbf{x} \mid \boldsymbol{\theta})$, and let the target posterior—the posterior under the new prior—be $q(\boldsymbol{\theta} \mid \mathbf{x}) \propto q(\boldsymbol{\theta})p(\mathbf{x} \mid \boldsymbol{\theta})$. Then, sampling from $q(\boldsymbol{\theta} \mid \mathbf{x})$ is equivalent to sampling from $r(\boldsymbol{\theta})p(\boldsymbol{\theta} \mid \mathbf{x})$ with $r(\boldsymbol{\theta}) \equiv \frac{q(\boldsymbol{\theta})}{p_{train}(\boldsymbol{\theta})}$ the prior ratio.*

*Proof.* We can rewrite the target posterior $q(\boldsymbol{\theta} \mid \mathbf{x})$ as

$$q(\boldsymbol{\theta} \mid \mathbf{x}) \propto q(\boldsymbol{\theta})p(\mathbf{x} \mid \boldsymbol{\theta}) = \frac{q(\boldsymbol{\theta})}{p_{\text{train}}(\boldsymbol{\theta})} \, p_{\text{train}}(\boldsymbol{\theta})p(\mathbf{x} \mid \boldsymbol{\theta}) \propto \frac{q(\boldsymbol{\theta})}{p_{\text{train}}(\boldsymbol{\theta})} \, p(\boldsymbol{\theta} \mid \mathbf{x}) = r(\boldsymbol{\theta})p(\boldsymbol{\theta} \mid \mathbf{x}),$$

where the prior ratio $r(\boldsymbol{\theta}) \equiv \frac{q(\boldsymbol{\theta})}{p_{\text{train}}(\boldsymbol{\theta})}$ takes the role of an importance weighing function, analogous to the correction applied in multi-round neural posterior estimation (Lueckmann et al., 2017). □

Next, we focus on the task of sampling from the target posterior. First, we show in Section 3.1 how introducing the new prior amounts to adding a guidance term to the diffusion process for posterior sampling. In Section 3.2, we develop analytical approximations to make this tractable. Section 3.3 shows how we can provide guarantees using corrective Langevin steps. Finally, in Section 3.4 we show how our results for posterior sampling readily extend to the posterior predictive case.

## 3.1 TARGET PRIOR AS GUIDANCE

Assume we have a diffusion model trained under $p_{\text{train}}(\boldsymbol{\theta})$ to sample from $p(\boldsymbol{\theta} \mid \mathbf{x})$ with learnt score model $s(\boldsymbol{\theta}_t, t, \mathbf{x})$. PriorGuide leverages the fact that we can relate the score of the target posterior $q(\boldsymbol{\theta} \mid \mathbf{x})$ to the original score. The marginal pdf at time $t$ of the diffusion process for $q(\boldsymbol{\theta} \mid \mathbf{x})$ is:

$$q(\boldsymbol{\theta}_t \mid \mathbf{x}) \propto \int r(\boldsymbol{\theta}_0)p(\boldsymbol{\theta}_0 \mid \mathbf{x})p(\boldsymbol{\theta}_t \mid \boldsymbol{\theta}_0) \, \mathrm{d}\boldsymbol{\theta}_0, \tag{5}$$

which is written as an integral over $\boldsymbol{\theta}_0$ by noting that $q(\boldsymbol{\theta}_0 \mid \mathbf{x}) \propto r(\boldsymbol{\theta}_0)p(\boldsymbol{\theta}_0 \mid \mathbf{x})$ and then we propagated this to time $t$ by convolution with the transition kernel $p(\boldsymbol{\theta}_t \mid \boldsymbol{\theta}_0)$. Thus, the score is:

$$\nabla_{\boldsymbol{\theta}_t} \log q(\boldsymbol{\theta}_t \mid \mathbf{x}) = \nabla_{\boldsymbol{\theta}_t} \log \int r(\boldsymbol{\theta}_0)p(\boldsymbol{\theta}_0 \mid \mathbf{x})p(\boldsymbol{\theta}_t \mid \boldsymbol{\theta}_0, \mathbf{x}) \, \mathrm{d}\boldsymbol{\theta}_0 \tag{6}$$

$$= \nabla_{\boldsymbol{\theta}_t} \log \int r(\boldsymbol{\theta}_0)p(\boldsymbol{\theta}_0 \mid \boldsymbol{\theta}_t, \mathbf{x})p(\boldsymbol{\theta}_t \mid \mathbf{x}) \, \mathrm{d}\boldsymbol{\theta}_0 \tag{7}$$

$$= \nabla_{\boldsymbol{\theta}_t} \log p(\boldsymbol{\theta}_t \mid \mathbf{x}) + \nabla_{\boldsymbol{\theta}_t} \log \int r(\boldsymbol{\theta}_0)p(\boldsymbol{\theta}_0 \mid \boldsymbol{\theta}_t, \mathbf{x}) \, \mathrm{d}\boldsymbol{\theta}_0 \tag{8}$$

$$= \underbrace{s(\boldsymbol{\theta}_t, t, \mathbf{x})}_{\text{original score}} + \nabla_{\boldsymbol{\theta}_t} \log \underbrace{\mathbb{E}_{p(\boldsymbol{\theta}_0 \mid \boldsymbol{\theta}_t, \mathbf{x})}}_{\text{reverse kernel}} \Big[ \underbrace{r(\boldsymbol{\theta}_0)}_{\text{prior ratio}} \Big], \tag{9}$$

where in Eq. (7) we re-express the joint probability $p(\boldsymbol{\theta}_0 \mid \mathbf{x})p(\boldsymbol{\theta}_t \mid \boldsymbol{\theta}_0, \mathbf{x}) = p(\boldsymbol{\theta}_0, \boldsymbol{\theta}_t \mid \mathbf{x})$ as $p(\boldsymbol{\theta}_0 \mid \boldsymbol{\theta}_t, \mathbf{x})p(\boldsymbol{\theta}_t \mid \mathbf{x})$, which allows us to separate the contribution of the new prior guidance from the original score model $s(\boldsymbol{\theta}_t, t, \mathbf{x})$. In multiple steps we exploit the fact that multiplicative constants inside the integral disappear under the score. In conclusion, Eq. (9) expresses the score of the target (new) posterior as the old score, which we have, plus a guidance term which we can estimate.

**Guided diffusion** We can draw samples from the posterior distribution via the reverse diffusion process using the modified score in Eq. (9). The first term is the trained score model and the second term estimates how the new prior's influence propagates to time $t$ (guidance term). This is a common way to implement a guidance function (Chung et al., 2023; Song et al., 2023a;b; Rissanen et al., 2025), which now depends on the prior ratio. The core challenge lies in evaluating the expectation over $\boldsymbol{\theta}_0$, which is intractable and requires simulating the reverse SDE. To make this tractable, we develop analytical approximations in the following section.

**Ensuring prior coverage** A crucial consideration for stable guidance is ensuring the new prior $q(\boldsymbol{\theta})$ remains within regions adequately covered by the training prior $p_{\text{train}}(\boldsymbol{\theta})$.[2] If $q(\boldsymbol{\theta})$ assigns significant mass to regions where $p_{\text{train}}(\boldsymbol{\theta})$ has negligible support (*i.e.*, is *out-of-distribution* or OOD; Lee et al., 2018; Nalisnick et al., 2019), two related issues can arise: (a) in these regions, the learned score $s(\boldsymbol{\theta}_t, t, \mathbf{x})$ is likely a poor representation of the true score, as the training set contained few examples in these regions; and (b) the prior ratio $r(\boldsymbol{\theta}) = q(\boldsymbol{\theta})/p_{\text{train}}(\boldsymbol{\theta})$ can become arbitrarily large or ill-defined, destabilizing the guidance mechanism. Lack of coverage can be quantified using OOD metrics (Lee et al., 2018; Nalisnick et al., 2019; Schmitt et al., 2023; Huang et al., 2024). Notably, the requirement that $q(\boldsymbol{\theta})$ should be covered by $p_{\text{train}}(\boldsymbol{\theta})$ is typically not restrictive since the common practice is to train amortized models on broad training priors. This diagnostic check is detailed in Appendix A.4.

---

[2]Within this coverage, $q(\boldsymbol{\theta})$ can differ substantially from $p_{\text{train}}(\boldsymbol{\theta})$—for instance, being more concentrated, multimodal, or shifted—enabling meaningful prior adaptation.

## 3.2 APPROXIMATING THE GUIDANCE FUNCTION

To approximate the guidance term in Eq. (9) efficiently while maintaining flexible test-time priors, we introduce two approximations. Following recent work (Song et al., 2023a; Peng et al., 2024; Rissanen et al., 2025), we model the reverse transition kernel as a Gaussian. We then introduce a novel approach that represents $r(\boldsymbol{\theta})$ as a Gaussian mixture model. This yields an analytical solution for the guidance, circumventing the issue of estimating the score of an expectation via Monte Carlo, which would suffer from both bias and variance.

**Reverse transition kernel approximation** We first approximate the reverse transition kernel $p(\boldsymbol{\theta}_0 \mid \boldsymbol{\theta}_t, \mathbf{x})$ as a multivariate Gaussian distribution:

$$p(\boldsymbol{\theta}_0 \mid \boldsymbol{\theta}_t, \mathbf{x}) \approx \mathcal{N}\left(\boldsymbol{\theta}_0 \mid \boldsymbol{\mu}_{0|t}(\boldsymbol{\theta}_t, \mathbf{x}), \boldsymbol{\Sigma}_{0|t}\right) \tag{10}$$

whose mean is obtained from the score function via Tweedie's formula (Song & Ermon, 2019):

$$\boldsymbol{\mu}_{0|t}(\boldsymbol{\theta}_t, \mathbf{x}) = \boldsymbol{\theta}_t + \sigma(t)^2 \nabla_{\boldsymbol{\theta}_t} \log p(\boldsymbol{\theta}_t \mid \mathbf{x}). \tag{11}$$

This approximation is common in the guidance literature (Chung et al., 2023; Song et al., 2023a; Boys et al.; Peng et al., 2024; Rissanen et al., 2025; Finzi et al., 2023; Bao et al., 2022). For the covariance matrix $\boldsymbol{\Sigma}_{0|t}$, we adopt a simple yet effective approximation inspired by (Song et al., 2023a; Ho et al., 2022):

$$\boldsymbol{\Sigma}_{0|t} = \frac{\sigma(t)^2}{1 + \sigma(t)^2} \mathbf{I}. \tag{12}$$

This approximation acts as a time-dependent scaling factor that naturally aligns with the diffusion process—starting at the identity matrix when $t = 1$ and approaching zero as $t \to 0$, effectively increasing the precision of our prior guidance at smaller timesteps. This approximation becomes exact for all $t$ if the posterior under the original target distribution is $p(\boldsymbol{\theta}_0 \mid \mathbf{x}) = \mathcal{N}(\boldsymbol{\theta}_0 \mid \mathbf{0}, \mathbf{I})$.[3]

**Prior ratio approximation** With the goal of obtaining a closed-form solution for the guidance, we then approximate the prior ratio function $r(\boldsymbol{\theta}) = \frac{q(\boldsymbol{\theta})}{p(\boldsymbol{\theta})}$ as a generalized mixture of Gaussians:

$$r(\boldsymbol{\theta}) \approx \sum_{i=1}^{K} w_i \mathcal{N}(\boldsymbol{\theta} \mid \boldsymbol{\mu}_i, \boldsymbol{\Sigma}_i), \qquad r(\boldsymbol{\theta}) \geq 0, \tag{13}$$

where $\{w_i, \boldsymbol{\mu}_i, \boldsymbol{\Sigma}_i\}_{i=1}^{K}$ represent the weights, means and covariance matrices of the mixture, with $K$ a hyperparameter denoting the number of mixture components.[4] Since this represents a ratio rather than a distribution, the mixture weights need not be positive nor sum to one, as long as the ratio remains non-negative, potentially enabling more expressive approximations such as subtractive mixtures (Loconte et al., 2024). Notably, when $p(\boldsymbol{\theta})$ is uniform, $r(\boldsymbol{\theta}) \propto q(\boldsymbol{\theta})$. Since the guidance term is a gradient of a log-expectation, the constant factor vanishes, allowing us to directly specify $q(\boldsymbol{\theta})$ as a Gaussian mixture. For the more general case of non-uniform training distributions, obtaining the Gaussian mixture approximation for the ratio is a standard function approximation task (Sorenson & Alspach, 1971). Crucially, since the densities of both $p_{\text{train}}$ and $q$ are analytically known, fitting the ratio avoids the instability and high variance inherent to statistical density-ratio estimation from finite samples. We provide a straightforward gradient-based fitting procedure in Appendix A.2.

**Guidance term** Plugging in Eq. (10) and Eq. (13), the guidance from Eq. (9) becomes:

$$\nabla_{\boldsymbol{\theta}_t} \log \mathbb{E}_{p(\boldsymbol{\theta}_0 \mid \boldsymbol{\theta}_t, \mathbf{x})} [r(\boldsymbol{\theta}_0)] \approx \nabla_{\boldsymbol{\theta}_t} \log \int \sum_{i=1}^{K} w_i \mathcal{N}(\boldsymbol{\theta}_0 \mid \boldsymbol{\mu}_i, \boldsymbol{\Sigma}_i) \mathcal{N}(\boldsymbol{\theta}_0 \mid \boldsymbol{\mu}_{0|t}(\boldsymbol{\theta}_t, \mathbf{x}), \boldsymbol{\Sigma}_{0|t}) \, \mathrm{d}\boldsymbol{\theta}_0. \tag{14}$$

This integral can be solved analytically (full derivation in Appendix B), yielding:

$$\nabla_{\boldsymbol{\theta}_t} \log \mathbb{E}_{p(\boldsymbol{\theta}_0 \mid \boldsymbol{\theta}_t, \mathbf{x})}[r(\boldsymbol{\theta}_0)] \approx \sum_{i=1}^{K} \tilde{w}_i (\boldsymbol{\mu}_i - \boldsymbol{\mu}_{0|t}(\boldsymbol{\theta}_t))^{\top} \widetilde{\boldsymbol{\Sigma}}_i^{-1} \nabla_{\boldsymbol{\theta}_t} \boldsymbol{\mu}_{0|t}(\boldsymbol{\theta}_t), \tag{15}$$

---

[3]More advanced covariance approximations are explored in the literature (Boys et al.; Bao et al.; Peng et al., 2024; Finzi et al., 2023; Manor & Michaeli; Rozet et al., 2024; Rissanen et al., 2025), but introduce added computational costs or implementation complexity. For this work, we focus on the simple and effective Eq. (12), which already yields strong results especially when combined with the Langevin refinement in Section 4.4.

[4]We empirically analyze the sensitivity to $K$ in Appendix D.5, finding that performance is robust once $K$ provides sufficient expressivity (e.g., $K = 20$ in this paper), and increasing $K$ has negligible computational cost.

where $\widetilde{\boldsymbol{\Sigma}}_i = \boldsymbol{\Sigma}_i + \boldsymbol{\Sigma}_{0|t}$ and $\tilde{w}_i = w_i \mathcal{N}(\boldsymbol{\mu}_i \mid \boldsymbol{\mu}_{0|t}(\boldsymbol{\theta}_t, \mathbf{x}), \widetilde{\boldsymbol{\Sigma}}_i) / \sum_{j=1}^{K} w_j \mathcal{N}(\boldsymbol{\mu}_j \mid \boldsymbol{\mu}_{0|t}(\boldsymbol{\theta}_t, \mathbf{x}), \widetilde{\boldsymbol{\Sigma}}_j)$.

Finally, the PriorGuide update to the mean of the reverse kernel can be expressed concisely using Tweedie's formula, Eq. (11), and our derived guidance term, Eq. (15):

$$\boldsymbol{\mu}_{0|t}^{\text{new}}(\boldsymbol{\theta}_t, \mathbf{x}) = \boldsymbol{\mu}_{0|t}(\boldsymbol{\theta}_t, \mathbf{x}) + \sigma(t)^2 \sum_{i}^{K} \tilde{w}_i (\boldsymbol{\mu}_i - \boldsymbol{\mu}_{0|t}(\boldsymbol{\theta}_t, \mathbf{x}))^\top \widetilde{\boldsymbol{\Sigma}}_i^{-1} \nabla_{\boldsymbol{\theta}_t} \boldsymbol{\mu}_{0|t}(\boldsymbol{\theta}_t, \mathbf{x}). \quad (16)$$

This update intuitively combines the original prediction $\mu_{0|t}(\boldsymbol{\theta}_t)$ based on the training prior with a weighted sum of correction terms from our new prior. The correction magnitude is controlled by both the noise schedule $\sigma(t)^2$ and the distance between the mixture components and current prediction.

## 3.3 ASYMPTOTICALLY CORRECT SAMPLING WITH LANGEVIN DYNAMICS

The diffusion sampling process detailed in Section 3.2 is approximate due to the Gaussian approximation of the reverse transition kernel $p(\boldsymbol{\theta}_0 \mid \boldsymbol{\theta}_t, \mathbf{x})$. However, as stated in the following proposition, this Gaussian approximation becomes correct on low noise levels, rendering our guidance term accurate.

**Proposition 2.** *As* $t, \sigma(t) \to 0$, *the approximation* $\mathcal{N}(\boldsymbol{\theta}_0 \mid \boldsymbol{\theta}_t + \sigma(t)^2 \nabla_{\boldsymbol{\theta}_t} \log p(\boldsymbol{\theta}_t), \frac{\sigma(t)^2}{1+\sigma(t)^2} \mathbf{I})$ *converges to the true* $p(\boldsymbol{\theta}_0 \mid \boldsymbol{\theta}_t)$, *under mild regularity conditions on* $p(\boldsymbol{\theta}_0)$.

We provide the exact statement and a proof in Appendix B. Close-by statements are well-known in the diffusion literature (Finzi et al., 2023; Ho et al., 2022). Thus, the guidance approximation is correct at low noise levels (up to the GMM prior ratio fit accuracy), and we can run accurate Langevin dynamics MCMC sampling (Särkkä & Solin, 2019). To incorporate this with the regular diffusion process, we run $N_L$ Langevin steps after each regular diffusion step, effectively transforming the sampling into an annealed MCMC process that resembles methods used in compositional generation (Geffner et al., 2023; Du et al., 2023) and early unconditional diffusion models (Song et al., 2021). This refinement increases the total inference cost by a factor of $N_L + 1$.

Thus, PriorGuide has two main hyperparameters: the number of diffusion steps $N > 0$, as per any diffusion model, and the number of interleaved Langevin steps $N_L \geq 0$. For test-time sampling, the total number of function evaluations (NFE), or forward passes of the trained network, is $N \times (N_L + 1)$.

## 3.4 PRIORGUIDE POSTERIOR PREDICTIVE SAMPLING

PriorGuide is readily applied to compute posterior predictive distributions as well under a new prior. Starting from a diffusion model trained to generate samples from the joint posterior predictive distribution, $p(\mathbf{x}^\star, \boldsymbol{\theta} \mid \mathbf{x})$, we can marginalize over $\boldsymbol{\theta}$ to get the posterior predictive $p(\mathbf{x}^\star \mid \mathbf{x})$. The joint posterior predictive under the new prior becomes:

$$q(\mathbf{x}^\star, \boldsymbol{\theta} \mid \mathbf{x}) = q(\mathbf{x}^\star \mid \boldsymbol{\theta}, \mathbf{x}) q(\boldsymbol{\theta} \mid \mathbf{x}) \propto q(\mathbf{x}^\star \mid \boldsymbol{\theta}, \mathbf{x}) r(\boldsymbol{\theta}) p(\boldsymbol{\theta} \mid \mathbf{x}) = r(\boldsymbol{\theta}) p(\mathbf{x}^\star, \boldsymbol{\theta} \mid \mathbf{x}),$$

which results in a posterior predictive version of Eq. (9):

$$\nabla_{\mathbf{x}_t^\star, \boldsymbol{\theta}_t} \log q(\mathbf{x}_t^\star, \boldsymbol{\theta}_t \mid \mathbf{x}) = s(\mathbf{x}_t^\star, \boldsymbol{\theta}_t, t, \mathbf{x}) + \nabla_{\mathbf{x}_t^\star, \boldsymbol{\theta}_t} \log \mathbb{E}_{p(\boldsymbol{\theta}_0 \mid \mathbf{x}_t^\star, \boldsymbol{\theta}_t, \mathbf{x})} [r(\boldsymbol{\theta}_0)]. \quad (17)$$

The posterior predictive and posterior guidance terms differ only in the conditioning information for the score and reverse transition kernel. Thus, everything presented earlier in this section applies to this scenario, and the posterior predictive is obtained from the previous formulas with substitutions $p(\boldsymbol{\theta}_0 \mid \boldsymbol{\theta}_t, \mathbf{x}) \to p(\boldsymbol{\theta}_0 \mid \boldsymbol{\xi}_t^\star, \mathbf{x})$, $\boldsymbol{\mu}_{0|t}(\boldsymbol{\theta}_t, \mathbf{x}) \to \boldsymbol{\mu}_{0|t}(\boldsymbol{\xi}_t^\star, \mathbf{x})$ and $\nabla_{\boldsymbol{\theta}} \to \nabla_{\boldsymbol{\xi}^\star}$, where $\boldsymbol{\xi}_t^\star \equiv (\mathbf{x}_t^\star, \boldsymbol{\theta}_t)$.

## 4 EXPERIMENTS

We empirically evaluate PriorGuide across a range of SBI problems, focusing on its ability to adapt to new priors at test time for posterior and posterior predictive inference. First, Section 4.1 provides an intuitive demonstration on a 2D problem. In Section 4.2, we evaluate posterior inference on several SBI problems, comparing PriorGuide to existing methods that support test-time prior adaptation. Section 4.3 examines PriorGuide's performance on challenging posterior predictive tasks. Finally, Section 4.4 studies the trade-off between computational cost and inference accuracy. Full experimental details can be found in Appendix C, with Appendix D.3 reporting additional baseline

results. We also conduct two ablation studies, including an analysis of the sensitivity of PriorGuide to the distance between training and test-time priors (Appendix D.4) and a study of the impact of the number of GMM components used to approximate the prior ratio (Appendix D.5). The code is available at https://github.com/acerbilab/prior-guide.

## 4.1 ILLUSTRATIVE EXAMPLE OF TEST-TIME PRIOR ADAPTATION

We illustrate PriorGuide's capabilities on Two Moons, a two-dimensional SBI model with a bimodal posterior (Greenberg et al., 2019). We train the diffusion model under a uniform prior $p_{\text{train}}(\boldsymbol{\theta})$ over $[-1, 1]^2$, and test how PriorGuide handles a new prior $q(\boldsymbol{\theta})$ at test time. Fig. A1 shows that PriorGuide incorporates the new prior, matching well the true Bayesian posterior under the new prior.

## 4.2 TEST-TIME PRIOR ADAPTATION FOR POSTERIOR INFERENCE

We evaluate PriorGuide's posterior inference capabilities on six SBI problems (see Table A1), ranging from established SBI benchmarks to real models from engineering and neuroscience: Two Moons (Lueckmann et al., 2021); the Ornstein-Uhlenbeck Process (OUP; Uhlenbeck & Ornstein, 1930); the Turin model of radio propagation (Turin et al., 1972); the Gaussian Linear model (Lueckmann et al., 2021) and its high-dimensional variant; and the Bayesian Causal Inference model of multisensory perception (BCI; Körding et al., 2007). Training priors $p_{\text{train}}(\boldsymbol{\theta})$ for the base diffusion model (Simformer; Gloeckler et al., 2024) were uniform or Gaussian; details in Appendix C.

For baselines, we consider the base Simformer (no prior adaptation) and the Amortized Conditioning Engine (ACE; Chang et al., 2025), one of several approaches (Elsemüller et al., 2024; Whittle et al., 2025) that amortizes test-time prior adaptation for posterior inference by pre-training on a variety of possible (factorized) priors. PriorGuide is more flexible than ACE by not needing pretraining on specific priors—instead, it modifies a diffusion-based amortized inference model at runtime—, and can represent correlated and non-factorized priors. Detailed comparisons against additional non-amortized methods—including classic algorithms (rejection sampling and sampling-importance-resampling) and neural likelihood estimation (NLE; Papamakarios et al., 2019) with MCMC—are provided in Appendix D.3. While often computationally expensive, these methods serve as fundamental benchmarks for posterior sampling.

We consider three different families of target priors: *mild*, *strong* and *mixture*. Mild and strong priors are defined as multivariate Gaussian distributions with means drawn from a uniform box and diagonal covariance matrices, where the strong priors have smaller standard deviations; these represent scenarios with varying degrees of available information. Mixture priors are defined as a mixture distribution with two multivariate Gaussian components with the same setup as the strong priors; this can represent situations with distinct, competing hypotheses about the parameter values. For each prior family, we randomly generate ten possible prior parameterizations $q^{(i)}(\boldsymbol{\theta})$, sample ten parameter vectors $\boldsymbol{\theta}_{i,j} \sim q^{(i)}(\boldsymbol{\theta})$ from that prior, and simulate one observed dataset per $\boldsymbol{\theta}_{i,j}$, $\mathbf{x}_{i,j} \sim p(\mathbf{x} \,|\, \boldsymbol{\theta}_{i,j})$ to evaluate the methods. See Appendix C.3.1 for the full procedure.

We measure each method's performance using: 1) the root mean squared error (RMSE) between the true parameter and the samples from the estimated posterior; 2) the classifier two-sample test (C2ST) between the estimated posterior samples and *ground-truth* posterior samples; 3) the mean marginal total variation distance (MMTV) between the estimated vs. *ground-truth* posterior samples. For RMSE and MMTV, lower is better, while for C2ST, closer to 0.5 is better.

Results in Table 1 show that PriorGuide largely improves inference accuracy over the base Simformer model in all scenarios, making use of the prior information provided at test time, and achieves leading performance in most cases, especially when stronger prior beliefs (*strong* and *mixture*) are presented.

## 4.3 TEST-TIME PRIOR ADAPTATION FOR DATA PREDICTION

We next evaluate PriorGuide's ability to perform posterior predictive inference under new target priors, focusing on forecasting or retrocasting scenarios, as shown in Fig. 1. We use the OUP and Turin models, both of which generate time series trajectories (Fig. 2). We employ the same procedure and test-time prior setup (mild, strong, and mixture) from Section 4.2. For each target prior, we

Table 1: Posterior inference ($\boldsymbol{\theta}$). Mean (standard dev.) over 5 independent training runs (10 random target priors $\times$ 10 simulated datasets). Significantly best results (Wilcoxon signed-rank test) in bold.

| | | $q_{\mathrm{mild}}(\boldsymbol{\theta})$ | | | $q_{\mathrm{strong}}(\boldsymbol{\theta})$ | | | $q_{\mathrm{mixture}}(\boldsymbol{\theta})$ | | |
| --- | --- | --- | --- | --- | --- | --- | --- | --- | --- | --- |
| | | RMSE | C2ST | MMTV | RMSE | C2ST | MMTV | RMSE | C2ST | MMTV |
| Two Moons | Simformer | 0.39(0.19) | 0.56(0.05) | 0.21(0.10) | 0.39(0.20) | 0.75(0.06) | 0.54(0.13) | 0.33(0.22) | 0.68(0.08) | 0.42(0.17) |
| | ACE | **0.34**(0.16) | 0.83(0.04) | 0.23(0.07) | **0.09**(0.03) | 0.79(0.05) | 0.35(0.12) | **0.16**(0.15) | 0.80(0.07) | 0.34(0.14) |
| | PriorGuide | 0.37(0.16) | **0.52**(0.02) | **0.11**(0.04) | **0.09**(0.04) | **0.52**(0.02) | **0.08**(0.02) | 0.20(0.17) | **0.55**(0.05) | **0.16**(0.11) |
| OUP | Simformer | 0.18(0.07) | 0.58(0.08) | 0.18(0.11) | 0.24(0.08) | 0.73(0.08) | 0.37(0.10) | 0.23(0.09) | 0.70(0.08) | 0.33(0.11) |
| | ACE | **0.17**(0.07) | 0.58(0.03) | 0.11(0.05) | 0.14(0.04) | 0.56(0.03) | 0.12(0.05) | 0.17(0.07) | 0.62(0.08) | 0.20(0.11) |
| | PriorGuide | **0.17**(0.06) | **0.52**(0.02) | **0.08**(0.04) | **0.13**(0.04) | **0.51**(0.02) | **0.06**(0.02) | **0.15**(0.06) | **0.51**(0.02) | **0.07**(0.04) |
| Turin | Simformer | 0.22(0.03) | 0.80(0.03) | 0.31(0.04) | 0.23(0.03) | 0.95(0.02) | 0.56(0.07) | 0.23(0.03) | 0.94(0.02) | 0.50(0.06) |
| | ACE | 0.21(0.02) | 0.78(0.03) | 0.27(0.04) | 0.17(0.02) | 0.92(0.04) | 0.47(0.07) | 0.19(0.03) | 0.91(0.03) | 0.44(0.06) |
| | PriorGuide | **0.14**(0.02) | **0.64**(0.06) | **0.13**(0.03) | **0.06**(0.01) | **0.55**(0.04) | **0.08**(0.03) | **0.13**(0.06) | **0.62**(0.08) | **0.19**(0.12) |
| Gaussian Linear 10D | Simformer | 0.29(0.02) | 0.89(0.02) | 0.30(0.03) | 0.31(0.03) | 1.00(0.00) | 0.65(0.03) | 0.30(0.03) | 0.99(0.00) | 0.61(0.05) |
| | ACE | **0.22**(0.02) | 0.67(0.04) | 0.12(0.02) | **0.10**(0.01) | 0.78(0.06) | 0.19(0.04) | **0.19**(0.04) | 0.96(0.03) | 0.40(0.11) |
| | PriorGuide | 0.31(0.08) | **0.53**(0.03) | **0.05**(0.01) | 0.23(0.09) | **0.54**(0.05) | **0.06**(0.02) | 0.26(0.09) | **0.57**(0.06) | **0.12**(0.06) |
| Gaussian Linear 20D | Simformer | 0.29(0.02) | 0.95(0.01) | 0.29(0.02) | 0.30(0.02) | 1.00(0.00) | 0.64(0.02) | 0.30(0.02) | 1.00(0.00) | 0.63(0.03) |
| | ACE | **0.22**(0.02) | 0.72(0.06) | 0.11(0.02) | **0.10**(0.01) | 0.82(0.05) | 0.16(0.03) | **0.20**(0.04) | 0.99(0.01) | 0.44(0.09) |
| | PriorGuide | 0.27(0.07) | **0.54**(0.03) | **0.05**(0.01) | 0.28(0.12) | **0.58**(0.03) | **0.11**(0.03) | 0.28(0.10) | **0.59**(0.05) | **0.14**(0.06) |
| BCI | Simformer | 0.79(0.13) | 0.86(0.05) | 0.35(0.09) | 1.03(0.16) | 0.98(0.01) | 0.61(0.09) | 0.99(0.24) | 0.98(0.02) | 0.63(0.09) |
| | ACE | **0.55**(0.12) | **0.84**(0.08) | **0.28**(0.10) | 0.31(0.13) | 0.82(0.09) | 0.29(0.12) | 1.02(0.37) | 0.97(0.02) | 0.56(0.11) |
| | PriorGuide | 0.56(0.10) | 0.88(0.06) | 0.41(0.08) | **0.25**(0.04) | **0.72**(0.10) | **0.21**(0.12) | **0.87**(0.68) | **0.78**(0.13) | **0.37**(0.27) |

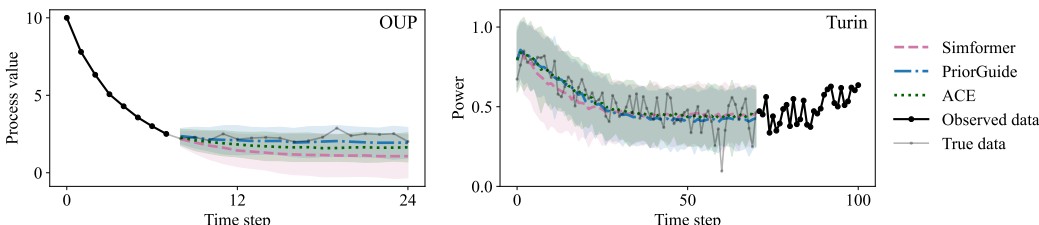

Figure 2: Example posterior predictive distributions for OUP and Turin models (strong priors).

condition the model on partial trajectories: in half of the cases, the first 30% of the trajectory, and for the other half, the last 30%. The task is always to predict the unobserved 70% of the trajectory.[5]

We evaluate the performance of all methods using RMSE and the maximum mean discrepancy (MMD) with an exponentiated quadratic kernel between the ground-truth trajectory $\mathbf{x}_o$ and generated posterior predictive samples. Fig. 2 shows how example posterior predictive distributions from PriorGuide closely match the true data. Results in Table 2 show that PriorGuide can generate reliable posterior predictive samples and achieve performance on par with or better than other methods.

Table 2: Data prediction ($\mathbf{x}$). Mean (standard dev.) over 5 independent training runs (10 random target priors $\times$ 10 simulated datasets). Significantly best results (Wilcoxon signed-rank test) in bold.

| | | $q_{\mathrm{mild}}(\boldsymbol{\theta})$ | | $q_{\mathrm{strong}}(\boldsymbol{\theta})$ | | $q_{\mathrm{mixture}}(\boldsymbol{\theta})$ | |
| --- | --- | --- | --- | --- | --- | --- | --- |
| | | RMSE | $\mathrm{MMD}_x$ | RMSE | $\mathrm{MMD}_x$ | RMSE | $\mathrm{MMD}_x$ |
| OUP | Simformer | 0.32(0.10) | 0.44(0.24) | 0.39(0.11) | 0.54(0.24) | 0.34(0.11) | 0.47(0.25) |
| | ACE | **0.26**(0.08) | **0.44**(0.31) | 0.22(0.06) | 0.30(0.20) | **0.24**(0.10) | 0.38(0.31) |
| | PriorGuide | 0.28(0.09) | 0.45(0.28) | **0.21**(0.05) | **0.29**(0.17) | 0.25(0.11) | **0.34**(0.22) |
| Turin | Simformer | 0.14(0.01) | 0.49(0.09) | 0.14(0.01) | 0.49(0.09) | 0.14(0.01) | 0.48(0.09) |
| | ACE | 0.16(0.03) | 0.62(0.19) | 0.16(0.03) | 0.61(0.20) | 0.16(0.03) | 0.61(0.19) |
| | PriorGuide | **0.13**(0.01) | **0.47**(0.08) | **0.13**(0.01) | **0.46**(0.07) | **0.13**(0.01) | **0.46**(0.09) |

## 4.4 TEST-TIME REFINEMENT VIA CORRECTIVE LANGEVIN DYNAMICS

PriorGuide supports improving the sampling quality by adding Langevin dynamic steps to the diffusion process, at the cost of additional test-time compute. We examine posterior inference accuracy—measured by MMTV, but similar results hold for other metrics—on the OUP and Turin models as a function of the number of diffusion steps $N$ and Langevin steps $N_L$. These two can be combined into a single computational cost metric, the *number of function evaluations* (NFEs), *i.e.*

---

[5]This task formulation prevents simple data interpolation as induced by sampling time indices randomly.

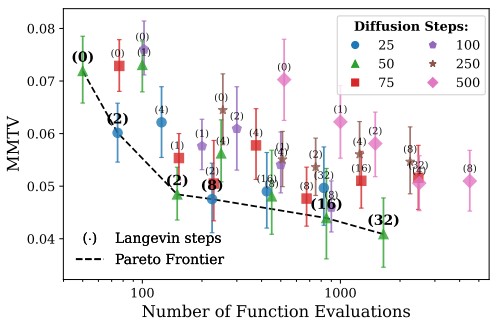 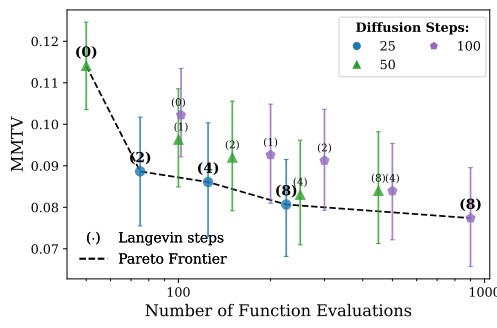

(a) Posterior inference on OUP, strong priors  (b) Posterior inference on Turin, strong priors

Figure 3: Pareto frontiers with respect to number of function evaluations (NFEs) and MMTV on posterior inference for OUP and Turin, with varying number of diffusion and Langevin steps.

calls to the score model. In Fig. 3, we visualize the relationship between MMTV, $N$ and $N_L$. The Pareto front shows that the best posterior inference is achieved by combining moderate diffusion steps ($N \sim 25$–$50$) with increasing Langevin corrections if the NFE budget allows it. The sample quality in general improves with more compute, implying that a simple way to calibrate the sampling parameters is to increase NFE until the output distribution does not change.

## 5    DISCUSSION

PriorGuide enables amortized diffusion-based SBI models to adapt to new prior distributions without retraining, an example of the *test-time compute* paradigm in extending pre-trained model capabilities with dedicated computations at test time which repurpose diffusion guidance for Bayesian inference. In practice, PriorGuide is recommended for moderate-to-high dimensional problems ($4 < D \lesssim 20$) where simulators are mildly-to-very expensive, making retraining a simulator model burdensome; for settings requiring complex, non-factorized priors, where amortized methods restricted by pre-defined meta-priors (*e.g.*, ACE) yield unsatisfactory performance; and for applications where prior adaptation needs to be fast but not strictly *instant*—in terms of pure speed, fully amortized methods without test-time compute remain the best choice.

**Limitations**  PriorGuide's effectiveness relies on the new prior $q(\boldsymbol{\theta})$ having substantial overlap with the training prior $p_{\text{train}}(\boldsymbol{\theta})$; out-of-distribution (OOD) target priors can lead to inaccurate learned scores and unstable guidance. The method also employs approximations—a Gaussian for the reverse transition kernel $p(\boldsymbol{\theta}_0 \,|\, \boldsymbol{\theta}_t, \mathbf{x})$ and a Gaussian mixture model for the prior ratio function $r(\boldsymbol{\theta})$—which can introduce inaccuracies, particularly for complex prior ratio shapes. Furthermore, current guidance calculations, involving matrix operations for the GMM components, may pose scalability challenges for high-dimensional parameter spaces ($\dim(\boldsymbol{\theta}) \gg 20$). PriorGuide sampling can be computationally intensive: although our method avoids retraining the base model—a key benefit with expensive simulators (*e.g.*, Turin model)—the iterative guided diffusion, particularly with interleaved Langevin refinement steps ($N_L$), incurs a cost. The number of function evaluations (NFEs) increases with diffusion ($N$) and Langevin steps, creating an accuracy-speed trade-off. This may render PriorGuide less suited than fully amortized methods for applications requiring very rapid inference. Advanced covariance approaches mentioned in Section 3.2 have the potential to speed up the method by requiring fewer NFEs, and represent an interesting future direction.

**Conclusions**  The ability of PriorGuide to decouple expensive simulator runs (for training the base model) from the specification of changing prior beliefs offers significant practical advantages. It allows for post-hoc prior sensitivity analyses and facilitates the direct incorporation of domain expert knowledge post-training, reducing the overall computational footprint in scientific workflows by avoiding the need for repeated model retraining when assumptions change.

ACKNOWLEDGEMENTS

This work was a part of Finland's Ministry of Education and Culture's Doctoral Education Pilot under Decision No. VN/3137/2024-OKM-6 (The Finnish Doctoral Program Network in Artificial Intelligence, AI-DOC). The project was also supported by the Research Council of Finland (Flagship programme: Finnish Center for Artificial Intelligence, FCAI). NL was funded by Business Finland (project 3576/31/2023) and LUMI AI Factory (EU Horizon Europe Joint Undertaking and its members including top-up funding by Ministry of Education and Culture). LA was supported by Research Council of Finland grants 356498 and 358980. SR, MH, and AS acknowledge funding from the Research Council of Finland (grants 339730, 362408, 334600). The authors also acknowledge the research environment provided by ELLIS Institute Finland.

We acknowledge CSC – IT Center for Science, Finland, for computational resources provided by the LUMI supercomputer, owned by the EuroHPC Joint Undertaking and hosted by CSC and the LUMI consortium (LUMI projects 462000864 and 462000873). Access was provided through the Finnish LUMI-OKM allocation. We acknowledge the computational resources provided by the Aalto Science-IT project.

Funded by the European Union. Views and opinions expressed are however those of the author(s) only and do not necessarily reflect those of the European Union or the granting authority. Neither the European Union nor the granting authority can be held responsible for them.

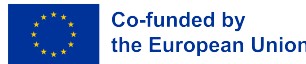
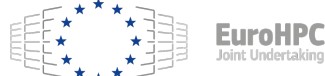

ETHICS STATEMENT

This work uses only synthetic datasets, with no sensitive data involved. The methods are for research purposes and pose no foreseeable ethical risks. We have followed the ICLR Code of Ethics.

REPRODUCIBILITY STATEMENT

The code is available at https://github.com/acerbilab/prior-guide. All experiments use synthetic datasets. Algorithmic details are presented in Appendix A, and all experimental details are specified in Appendix C.

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

# APPENDIX

The full appendix is organized as follows:

- Appendix A provides an extended description of related work and our method.
- Appendix B presents mathematical proofs and derivations.
- Appendix C describes our experimental and statistical procedures.
- Appendix D shows supplementary experimental results and analyses.
- Appendix E details computational and software resources.

## A  METHOD DETAILS

In this section we start with an extended discussion of related work (Appendix A.1). We then detail the main PriorGuide algorithm (Appendix A.2), the Langevin dynamics step size (Appendix A.3), and the prior coverage diagnostics (Appendix A.4).

### A.1  EXTENDED RELATED WORK

Section 2 in the main paper situates PriorGuide within the broader context of Simulation-Based Inference (SBI) and diffusion models. Here we explore those connections in more detail.

**Amortized SBI and prior specification**  The output of standard amortized SBI techniques, such as Neural Posterior Estimation (NPE) (Greenberg et al., 2019; Lueckmann et al., 2017; Papamakarios & Murray, 2016), is tied to the fixed prior, $p_{\text{train}}(\boldsymbol{\theta})$, used during their training phase. Modifying this prior traditionally requires retraining the entire amortized model, which can be prohibitive given expensive simulators. PriorGuide offers a solution specifically for diffusion-based amortized models, enabling adaptation to a new prior $q(\boldsymbol{\theta})$ by modifying the sampling process itself, thus bypassing the need for retraining. Other SBI techniques such as Neural Likelihood Estimation (NLE) (Papamakarios et al., 2019; Lueckmann et al., 2019) and Neural Ratio Estimation (NRE) (Hermans et al., 2020; Thomas et al., 2022) do not amortize posterior inference, in that they only approximate the likelihood (or likelihood ratio), and traditional approximate inference techniques such as MCMC or variational inference need to be run to obtain a posterior by combining the surrogate likelihood (or likelihood ratio) with a prior.

**Diffusion models for SBI**  PriorGuide enhances versatile diffusion-based SBI models like Simformer (Gloeckler et al., 2024), which we use as our base model. As described in the main text, Simformer leverages a transformer-based diffusion model over the joint space of parameters and data, $p(\boldsymbol{\theta}, \mathbf{x})$, allowing it to provide amortized samples from arbitrary conditionals (*e.g.*, posteriors, likelihoods) once trained, though this training inherently uses a fixed prior. Other diffusion-based SBI methods include techniques to combine learned scores from posteriors of individual observations to handle multiple data sources (Geffner et al., 2023; Linhart et al., 2024) and methods that focus on efficient (sequential) training of posterior score models (Sharrock et al., 2024). Since these approaches ultimately yield score functions for posteriors (conditioned on their respective training priors), PriorGuide's test-time score guidance mechanism could also adapt these trained models to new prior beliefs post-hoc.

**Amortized prior adaptation**  As discussed in Section 2.3 of the main text, several recent amortized SBI methods support prior changes by training over a *meta-prior*—a distribution or discrete set over possible prior specifications—including ACE (Chang et al., 2025), the Distribution Transformer (Whittle et al., 2025), and sensitivity-aware SBI (Elsemüller et al., 2024). These approaches learn to incorporate alternative priors at inference time, but rely on pre-training across the chosen meta-prior. PriorGuide, by contrast, performs prior adaptation entirely at test time, without amortizing over priors, and can therefore accommodate new target priors beyond any pre-specified meta-prior.

**Prior misspecification**  Effective PriorGuide use requires the target prior $q(\boldsymbol{\theta})$ to overlap substantially with the training prior $p_{\text{train}}(\boldsymbol{\theta})$ to prevent the trained score model from operating out-of-distribution (OOD). This concern for reliable inference echoes broader SBI efforts that address simulator misspecification and its impact on inference reliability using techniques like MMD or robust statistics (Schmitt

et al., 2023; Huang et al., 2024). Similarly, Yuyan et al. (2025) explore robust SBI with classes of priors and assess potential prior-likelihood conflicts. While this paper proposes a simple diagnostics to ensure the new prior is compatible with the learned score model, other techniques from the SBI literature could be used.

**Score-based guidance** PriorGuide's core mechanism is a novel application of score-based guidance. While the general idea of guiding diffusion models is well-established for inverse problems and conditional generation (Dhariwal & Nichol, 2021; Ho & Salimans, 2022; Chung et al., 2023; Song et al., 2023a;b), PriorGuide's specific contribution lies in deriving and applying a guidance term for the *prior ratio*, akin to an importance ratio term. Moreover, its analytical approximation using a GMM for the ratio is tailored to this prior adaptation task. This contrasts with other guidance methods that often focus on incorporating information from a known forward (observation) model (*e.g.*, linear operators imaging; Chung et al., 2023; Finzi et al., 2023) or specific loss functions (Song et al., 2023b). Further, the literature either tends to focus on the analytic case where the guidance function is Gaussian (Song et al., 2023a; Peng et al., 2024; Rissanen et al., 2025) or entirely forego an analytical integral while keeping the Gaussian reverse approximation (Song et al., 2023b). Our approach strikes a middle ground between analytic tractability and flexible, non-Gaussian guidance functions. The approximation of the reverse transition kernel and its covariance also adapts approaches seen in works like (Song et al., 2023a; Ho et al., 2022).

A number of works has considered Monte Carlo corrections to approximate inference time modifications to diffusion models, such as guidance. Wu et al. (2023); Dou & Song (2024); Cardoso et al.; Thornton et al.; Lee et al. (2025); Skreta et al. propose variants of Sequential Monte Carlo for asymptotically exact modifications to the generative distribution. Du et al. (2023); Geffner et al. (2023) use MCMC corrections for compositional generation. Early work on using Langevin dynamics in unconditional score based generative models includes Song & Ermon (2019) and the predictor-corrector sampler from Song et al. (2021).

## A.2 PRIORGUIDE INFERENCE ALGORITHM

Algorithm 1 details the PriorGuide posterior inference algorithm and Algorithm 2 contains the posterior predictive inference version.

---

**Algorithm 1** PriorGuide posterior inference

---

1: **Input:** Trained diffusion-based inference model $\mathcal{M}$, training prior $p_{\text{train}}(\boldsymbol{\theta})$, test-time prior $q(\boldsymbol{\theta})$, number of mixture components $K$ for the prior ratio, min diffusion time $T_{\min}$, max diffusion time $T_{\max}$, generation schedule nonlinearity parameter $\rho$, number of diffusion steps $N$, Langevin ratio $\eta$, number of Langevin steps $N_L$, conditioning information $\mathbf{x}$.

2: **Output:** Posterior samples $\boldsymbol{\theta}_{T_{\min}}$ at time $T_{\min}$.

3: $\{r(\boldsymbol{\theta}) \mid \{w_i, \boldsymbol{\mu}_i, \boldsymbol{\Sigma}_i\}_i^K\} \leftarrow \text{FITGMM}(p_{\text{train}}(\boldsymbol{\theta}), q(\boldsymbol{\theta}), K)$, $\qquad$ with $r(\boldsymbol{\theta}) \approx \frac{q(\boldsymbol{\theta})}{p_{\text{train}}(\boldsymbol{\theta})}$

4: $t_N, \dots, t_0 \leftarrow \text{Linspace}(1, 0, N)^\rho \cdot (T_{\max} - T_{\min}) + T_{\min}$

5: $\boldsymbol{\theta}_{t_N} \sim \mathcal{N}(0, \sigma(T_{\max})^2 \mathbf{I})$

6: **for** $j = N \to 1$ **do**

7: $\quad$ $t = t_j, \Delta t = t_{j-1} - t_j$

8: $\quad$ **for** $\ell = 1 \to N_L$ **do**

9: $\quad\quad$ Compute original score $s(\boldsymbol{\theta}_t, t, \mathbf{x})$ using $\mathcal{M}$

10: $\quad\quad$ Compute prior guidance $s_p \leftarrow \nabla_{\boldsymbol{\theta}_t} \log \mathbb{E}_{p(\boldsymbol{\theta}_0 \mid \boldsymbol{\theta}_t, \mathbf{x})}[r(\boldsymbol{\theta}_0)]$ with $\{w_i, \boldsymbol{\mu}_i, \boldsymbol{\Sigma}_i\}_i^K$

11: $\quad\quad$ Compute guided score $s_L \leftarrow s(\boldsymbol{\theta}_t, t, \mathbf{x}) + s_p$

12: $\quad\quad$ Langevin dynamics step $\boldsymbol{\theta}_t \leftarrow \boldsymbol{\theta}_t + \eta \frac{\dot{\sigma}(t)\sigma(t)}{2} s_L + \sqrt{\eta \dot{\sigma}(t)\sigma(t)}\varepsilon, \quad \varepsilon \sim \mathcal{N}(0, \mathbf{I})$

13: $\quad$ **end for**

14: $\quad$ Compute original score $s(\boldsymbol{\theta}_t, t, \mathbf{x})$ using $\mathcal{M}$

15: $\quad$ Compute prior guidance $s_p \leftarrow \nabla_{\boldsymbol{\theta}_t} \log \mathbb{E}_{p(\boldsymbol{\theta}_0 \mid \boldsymbol{\theta}_t, \mathbf{x})}[r(\boldsymbol{\theta}_0)]$ with $\{w_i, \boldsymbol{\mu}_i, \boldsymbol{\Sigma}_i\}_i^K$

16: $\quad$ Compute new guided score $\tilde{s} \leftarrow s(\boldsymbol{\theta}_t, t, \mathbf{x}) + s_p$

17: $\quad$ Euler-Maruyama step $\boldsymbol{\theta}_{t_{j-1}} \leftarrow \boldsymbol{\theta}_t + 2\dot{\sigma}(t)\sigma(t)\Delta t \tilde{s} + \sqrt{2\dot{\sigma}(t)\sigma(t)\Delta t}\varepsilon, \quad \varepsilon \sim \mathcal{N}(0, \mathbf{I})$

18: **end for**

19: **return** $\boldsymbol{\theta}_{t_0}$

---

**FITGMM subroutine** The FITGMM($p(\boldsymbol{\theta})$, $q(\boldsymbol{\theta})$, K) subroutine takes as input two distributions, $p(\boldsymbol{\theta})$ and $q(\boldsymbol{\theta})$, and fits a generalized Gaussian mixture model (GMM) with $K$ components to approximate the ratio $r(\boldsymbol{\theta}) = q(\boldsymbol{\theta})/p(\boldsymbol{\theta})$, as described in Section 3 of the main text. This is a standard GMM but coefficients are not constrained to sum to one. The subroutine returns the approximated ratio as the weights, means, and covariance matrices of the mixture. This is implemented as a stochastic optimization procedure over the parameters of the generalized mixture by minimizing the $L_2$ error between the GMM and the ratio (see Appendix C.3.2).

---

**Algorithm 2** PriorGuide posterior predictive inference

---

1: **Input:** Trained diffusion-based inference model $\mathcal{M}$, training prior $p_{\text{train}}(\boldsymbol{\theta})$, test-time prior $q(\boldsymbol{\theta})$, number of mixture components $K$ for the prior ratio, min diffusion time $T_{\min}$, max diffusion time $T_{\max}$, generation schedule nonlinearity parameter $\rho$, number of diffusion steps $N$, Langevin ratio $\eta$, number of Langevin steps $N_L$, conditioning information $\mathbf{x}$.
2: **Output:** Posterior predictive samples $\mathbf{x}^\star_{T_{\min}}$ at time $T_{\min}$.
3: $\{r(\boldsymbol{\theta}) \mid \{w_i, \boldsymbol{\mu}_i, \boldsymbol{\Sigma}_i\}_i^K\} \leftarrow \text{FITGMM}(p_{\text{train}}(\boldsymbol{\theta}), q(\boldsymbol{\theta}), K)$, $\qquad$ with $r(\boldsymbol{\theta}) \approx \frac{q(\boldsymbol{\theta})}{p_{\text{train}}(\boldsymbol{\theta})}$
4: $t_N, \ldots, t_0 \leftarrow \text{Linspace}(1, 0, N)^\rho \cdot (T_{\max} - T_{\min}) + T_{\min}$
5: $\mathbf{x}^\star_{t_N} \sim \mathcal{N}(0, \sigma(T_{\max})^2 \mathbf{I})$
6: $\boldsymbol{\theta}^\star_{t_N} \sim \mathcal{N}(0, \sigma(T_{\max})^2 \mathbf{I})$
7: $\boldsymbol{\xi}_{t_N} = (\boldsymbol{\theta}_{t_N}, \mathbf{x}^\star_{t_N})$
8: **for** $j = N \rightarrow 1$ **do**
9: $\quad$ $t = t_j, \Delta t = t_{j-1} - t_j$
10: $\quad$ **for** $\ell = 1 \rightarrow N_L$ **do**
11: $\quad\quad$ Compute original score $s(\boldsymbol{\xi}_t, t, \mathbf{x})$ using $\mathcal{M}$
12: $\quad\quad$ Compute prior guidance $s_p \leftarrow \nabla_{\boldsymbol{\xi}_t} \log \mathbb{E}_{p(\boldsymbol{\theta}_0 \mid \boldsymbol{\xi}_t, \mathbf{x})}[r(\boldsymbol{\theta}_0)]$ with $\{w_i, \boldsymbol{\mu}_i, \boldsymbol{\Sigma}_i\}_i^K$
13: $\quad\quad$ Compute guided score $s_L \leftarrow s(\boldsymbol{\xi}_t, t, \mathbf{x}) + s_p$
14: $\quad\quad$ Langevin dynamics step $\boldsymbol{\xi}_t \leftarrow \boldsymbol{\xi}_t + \eta \frac{\dot{\sigma}(t)\sigma(t)}{2} s_L + \sqrt{\eta \dot{\sigma}(t)\sigma(t)} \varepsilon, \quad \varepsilon \sim \mathcal{N}(0, \mathbf{I})$
15: $\quad$ **end for**
16: $\quad$ Compute original score $s(\boldsymbol{\xi}_t, t, \mathbf{x})$ using $\mathcal{M}$
17: $\quad$ Compute prior guidance $s_p \leftarrow \nabla_{\boldsymbol{\xi}_t} \log \mathbb{E}_{p(\boldsymbol{\theta}_0 \mid \boldsymbol{\xi}_t, \mathbf{x})}[r(\boldsymbol{\theta}_0)]$ with $\{w_i, \boldsymbol{\mu}_i, \boldsymbol{\Sigma}_i\}_i^K$
18: $\quad$ Compute new guided score $\tilde{s} \leftarrow s(\boldsymbol{\xi}_t, t, \mathbf{x}) + s_p$
19: $\quad$ Euler-Maruyama step $\boldsymbol{\xi}_{t_{j-1}} \leftarrow \boldsymbol{\xi}_t + 2\dot{\sigma}(t)\sigma(t)\Delta t \tilde{s} + \sqrt{2\dot{\sigma}(t)\sigma(t)\Delta t} \varepsilon, \quad \varepsilon \sim \mathcal{N}(0, \mathbf{I})$
20: **end for**
21: **return** $\mathbf{x}^\star_{t_0}$

---

## A.3 LANGEVIN DYNAMICS STEP SIZE

Since Langevin dynamics becomes exact only at small step sizes, a schedule for the step size is an important detail of the PriorGuide algorithm. At larger noise levels $\sigma(t)$, the distribution $p(\mathbf{z}_t)$ is more spread out and thus we can take larger steps, while a smaller step size is necessary for lower $\sigma(t)$ levels. We take inspiration from the similarity of the Euler-Maruyama reverse SDE sampler and the Langevin dynamics algorithm (see, *e.g.*, (Karras et al., 2022)), and calibrate the step size such that the noise added in the sampling step is proportional to the noise added in the Euler-Maruyama step when moving to the next noise level. In particular, the update rule for Langevin dynamics at noise level $\sigma(t)$ is

$$\mathbf{z}_t \leftarrow \mathbf{z}_t + \delta(t)\nabla_\mathbf{z} \log p(\mathbf{z}_t) + \sqrt{2\delta(t)}\varepsilon, \quad \varepsilon \sim \mathcal{N}(0, \mathbf{I}), \quad \delta(t) = \eta \frac{\dot{\sigma}(t)\sigma(t)\Delta t}{2}, \quad \text{(A1)}$$

where $\Delta t$ is the step size for the next step in the reverse SDE, and $\eta$ is a scaling parameter. $\eta = 1$ corresponds to the same noise level as the reverse Euler-Maruyama step. The overall scaling $\eta$ can be tuned to lower values for improved accuracy of the MCMC procedure, at the cost of slower mixing. We use $\eta = 0.5$ for all our experiments.

## A.4 OOD DIAGNOSTIC FOR TEST-TIME PRIORS

We assess out-of-distribution (OOD) behavior using a Monte Carlo sample-based diagnostic that estimates the mass of the test-time prior $q(\boldsymbol{\theta})$ falling into the $\alpha$-quantile of the training prior $p_{\text{train}}(\boldsymbol{\theta})$. We outline the procedures as follows:

- First, we compute the log-density threshold $t$ under $p_{\text{train}}$.
  - Draw $M_p$ samples $\{\boldsymbol{\theta}^{(i)}\}_{i=1}^{M_p} \sim p_{\text{train}}$.
  - Compute their log-densities $\ell_i = \log p_{\text{train}}(\boldsymbol{\theta}^{(i)})$.
  - Let $t$ be the empirical $\alpha$-quantile of $\{\ell_i\}$, i.e. $t = \text{quantile}(\{\ell_i\}, \alpha)$.
  - By construction, we have $\Pr_{p_{\text{train}}}[\log p_{\text{train}}(\boldsymbol{\theta}) < t] \approx \alpha$.
- Then, we estimate the OOD fraction under $q$.
  - Draw $M_q$ samples $\{\boldsymbol{\phi}^{(j)}\}_{j=1}^{M_q} \sim q$.
  - Evaluate each under $p$: $\ell'_j = \log p(\boldsymbol{\phi}^{(j)})$.
  - Count the fraction $\widehat{r}$ with $\ell'_j < t$: $\widehat{r} = \frac{1}{M_q} \sum_{j=1}^{M_q} [\ell'_j < t]$.
  - If $\widehat{r} > \alpha$, declare $q$ OOD.

Across all simulators, we employ a quantile threshold of $\alpha = 0.001$, chosen as $10/N_{\text{train}}$, where $N_{\text{train}}$ is the number of simulated parameters used to train the amortized method. We verify that each newly constructed prior (procedures detailed in Appendix C.3.1) successfully passes the above OOD diagnostics.

**Validity of the prior ratio** For the prior ratio $r(\boldsymbol{\theta}) = q(\boldsymbol{\theta})/p_{\text{train}}(\boldsymbol{\theta})$ to be well-defined, we require $p_{\text{train}}(\boldsymbol{\theta}) = 0 \rightarrow q(\boldsymbol{\theta}) = 0$, *i.e.* the target prior cannot have nonzero density where the training prior has zero density. This is not fully addressed by the OOD diagnostic, which looks for substantial overlap of prior mass. To avoid pointwise issues with zero-density regions (*e.g.*, when $p_{\text{train}}(\boldsymbol{\theta})$ is a bounded uniform distribution), we further truncate the tails of $q(\boldsymbol{\theta})$, setting its density to zero if $p_{\text{train}}(\boldsymbol{\theta}) = 0$. Note that this is done after the OOD check, which means that this adjustment only affects the far tails of $q(\boldsymbol{\theta})$, with negligible influence on inference performance.

## B    THEORETICAL RESULTS

We provide in this section full derivations and proofs of our theoretical results. This includes the derivation of the guidance term (Appendix B.1) and extended statements and proofs for Proposition 1 (Appendix B.2) and Proposition 2 (Appendix B.3) from the main text.

### B.1    DERIVATION OF THE GUIDANCE TERM

Here we provide a detailed derivation for the guidance term, Eq. 15 from the main text. We start from Eq. 14, which writes the guidance as the score of the expectation of the prior ratio under the reverse transition kernel, which are approximated by a Gaussian mixture model and a Gaussian, respectively:

$$\nabla_{\boldsymbol{\theta}_t} \log \mathbb{E}_{p(\boldsymbol{\theta}_0 \mid \boldsymbol{\theta}_t, \mathbf{x})}[r(\boldsymbol{\theta}_0)] \approx \nabla_{\boldsymbol{\theta}_t} \log \int \sum_{i=1}^{K} w_i \mathcal{N}(\boldsymbol{\theta}_0|\boldsymbol{\mu}_i, \boldsymbol{\Sigma}_i)\mathcal{N}(\boldsymbol{\theta}_0|\boldsymbol{\mu}_{0|t}(\boldsymbol{\theta}_t), \boldsymbol{\Sigma}_{0|t})d\boldsymbol{\theta}_0, \quad \text{(A2)}$$

$$= \nabla_{\boldsymbol{\theta}_t} \log \sum_{i=1}^{K} w_i \int \mathcal{N}(\boldsymbol{\mu}_i|\boldsymbol{\theta}_0, \boldsymbol{\Sigma}_i)\mathcal{N}(\boldsymbol{\theta}_0|\boldsymbol{\mu}_{0|t}(\boldsymbol{\theta}_t), \boldsymbol{\Sigma}_{0|t})d\boldsymbol{\theta}_0. \quad \text{(A3)}$$

The step above uses the symmetry property of Gaussian distributions: if $\boldsymbol{a} \sim \mathcal{N}(\boldsymbol{\mu}, \boldsymbol{\Sigma})$ then $\boldsymbol{\mu} \sim \mathcal{N}(\boldsymbol{a}, \boldsymbol{\Sigma})$. This allows us to swap $\boldsymbol{\theta}_0$ and $\boldsymbol{\mu}_i$ in the first Gaussian. Furthermore, using the standard result for the convolution of two Gaussian distributions:

$$\int \mathcal{N}(\mathbf{x}|\boldsymbol{\mu}_1, \boldsymbol{\Sigma}_1)\mathcal{N}(\boldsymbol{\mu}_1|\boldsymbol{\mu}_2, \boldsymbol{\Sigma}_2)d\boldsymbol{\mu}_1 = \mathcal{N}(\mathbf{x}|\boldsymbol{\mu}_2, \boldsymbol{\Sigma}_1 + \boldsymbol{\Sigma}_2), \quad \text{(A4)}$$

we get

$$\nabla_{\boldsymbol{\theta}_t} \log \mathbb{E}_{p(\boldsymbol{\theta}_0 \mid \boldsymbol{\theta}_t, \mathbf{x})}[r(\boldsymbol{\theta}_0)] \approx \nabla_{\boldsymbol{\theta}_t} \log \sum_{i=1}^{K} w_i \mathcal{N}(\boldsymbol{\mu}_i|\boldsymbol{\mu}_{0|t}(\boldsymbol{\theta}_t), \boldsymbol{\Sigma}_i + \boldsymbol{\Sigma}_{0|t}). \quad \text{(A5)}$$

For notational convenience, we define $\widetilde{\boldsymbol{\Sigma}}_i = \boldsymbol{\Sigma}_i + \boldsymbol{\Sigma}_{0|t}$ continuing with the derivation:

$$= \nabla_{\boldsymbol{\theta}_t} \log \sum_{i=1}^{K} w_i \mathcal{N}(\boldsymbol{\mu}_i | \boldsymbol{\mu}_{0|t}(\boldsymbol{\theta}_t), \widetilde{\boldsymbol{\Sigma}}_i), \tag{A6}$$

$$= \frac{\nabla_{\boldsymbol{\theta}_t} \sum_{i=1}^{K} w_i \mathcal{N}(\boldsymbol{\mu}_i | \boldsymbol{\mu}_{0|t}(\boldsymbol{\theta}_t), \widetilde{\boldsymbol{\Sigma}}_i)}{\sum_{j=1}^{K} w_j \mathcal{N}(\boldsymbol{\mu}_j | \boldsymbol{\mu}_{0|t}(\boldsymbol{\theta}_t), \widetilde{\boldsymbol{\Sigma}}_j)} \quad \text{(chain rule)}, \tag{A7}$$

$$= \frac{\sum_{i=1}^{K} w_i \mathcal{N}(\boldsymbol{\mu}_i | \boldsymbol{\mu}_{0|t}(\boldsymbol{\theta}_t), \widetilde{\boldsymbol{\Sigma}}_i) \nabla_{\boldsymbol{\theta}_t} \log \mathcal{N}(\boldsymbol{\mu}_i | \boldsymbol{\mu}_{0|t}(\boldsymbol{\theta}_t), \widetilde{\boldsymbol{\Sigma}}_i)}{\sum_{j=1}^{K} w_j \mathcal{N}(\boldsymbol{\mu}_j | \boldsymbol{\mu}_{0|t}(\boldsymbol{\theta}_t), \widetilde{\boldsymbol{\Sigma}}_j)} \quad \text{(since } \nabla f = f \nabla \log f), \tag{A8}$$

$$= \frac{\sum_{i=1}^{K} w_i \mathcal{N}(\boldsymbol{\mu}_i | \boldsymbol{\mu}_{0|t}(\boldsymbol{\theta}_t), \widetilde{\boldsymbol{\Sigma}}_i) \nabla_{\boldsymbol{\theta}_t} \left( -\frac{1}{2} (\boldsymbol{\mu}_{0|t}(\boldsymbol{\theta}_t) - \boldsymbol{\mu}_i)^\top \widetilde{\boldsymbol{\Sigma}}_i^{-1} (\boldsymbol{\mu}_{0|t}(\boldsymbol{\theta}_t) - \boldsymbol{\mu}_i) \right)}{\sum_{j=1}^{K} w_j \mathcal{N}(\boldsymbol{\mu}_j | \boldsymbol{\mu}_{0|t}(\boldsymbol{\theta}_t), \widetilde{\boldsymbol{\Sigma}}_j)}, \tag{A9}$$

$$= \frac{\sum_{i=1}^{K} w_i \mathcal{N}(\boldsymbol{\mu}_i | \boldsymbol{\mu}_{0|t}(\boldsymbol{\theta}_t), \widetilde{\boldsymbol{\Sigma}}_i)(\boldsymbol{\mu}_i - \boldsymbol{\mu}_{0|t}(\boldsymbol{\theta}_t))^{\mathbf{T}} \widetilde{\boldsymbol{\Sigma}}_i^{-1} \nabla_{\boldsymbol{\theta}_t} \boldsymbol{\mu}_{0|t}(\boldsymbol{\theta}_t)}{\sum_{j=1}^{K} w_j \mathcal{N}(\boldsymbol{\mu}_j | \boldsymbol{\mu}_{0|t}(\boldsymbol{\theta}_t), \widetilde{\boldsymbol{\Sigma}}_j)}. \tag{A10}$$

Finally, with the following definitions:

$$\widetilde{\boldsymbol{\Sigma}}_i = \boldsymbol{\Sigma}_i + \boldsymbol{\Sigma}_{0|t} \tag{A11}$$

$$\tilde{w}_i = \frac{w_i \mathcal{N}(\boldsymbol{\mu}_i | \boldsymbol{\mu}_{0|t}(\boldsymbol{\theta}_t, \mathbf{x}), \widetilde{\boldsymbol{\Sigma}}_i)}{\sum_{j=1}^{K} w_j \mathcal{N}(\boldsymbol{\mu}_j | \boldsymbol{\mu}_{0|t}(\boldsymbol{\theta}_t, \mathbf{x}), \widetilde{\boldsymbol{\Sigma}}_j)} \tag{A12}$$

we obtain

$$\nabla_{\boldsymbol{\theta}_t} \log \mathbb{E}_{p(\boldsymbol{\theta}_0 | \boldsymbol{\theta}_t, \mathbf{x})}[r(\boldsymbol{\theta}_0)] \approx \sum_{i=1}^{K} \tilde{w}_i (\boldsymbol{\mu}_i - \boldsymbol{\mu}_{0|t}(\boldsymbol{\theta}_t))^\top \widetilde{\boldsymbol{\Sigma}}_i^{-1} \nabla_{\boldsymbol{\theta}_t} \boldsymbol{\mu}_{0|t}(\boldsymbol{\theta}_t), \tag{A13}$$

which is Eq. 15 in the main text.

## B.2 PROOF OF PROPOSITION 1

This proposition is fully derived in the main paper, we only include here a natural assumption to guarantee the existence of the ratio (see also Appendix A.4).

Let $r(\boldsymbol{\theta}) \equiv \frac{q(\boldsymbol{\theta})}{p_{\text{train}}(\boldsymbol{\theta})}$ the prior ratio. We assume that $p_{\text{train}}(\boldsymbol{\theta}) = 0 \rightarrow q(\boldsymbol{\theta}) = 0$ almost everywhere, *i.e.* $q(\boldsymbol{\theta})$ needs to be zero outside the support of $p_{\text{train}}(\theta)$, except for a set of measure zero.

**Proposition** (Proposition 1). *Let the posterior under the original prior be $p(\boldsymbol{\theta} | \mathbf{x}) \propto p_{\text{train}}(\boldsymbol{\theta})p(\mathbf{x} | \boldsymbol{\theta})$, and let the target posterior—the posterior under the new prior—be $q(\boldsymbol{\theta} | \mathbf{x}) \propto q(\boldsymbol{\theta})p(\mathbf{x} | \boldsymbol{\theta})$. Then, sampling from $q(\boldsymbol{\theta} | \mathbf{x})$ is equivalent to sampling from $r(\boldsymbol{\theta})p(\boldsymbol{\theta} | \mathbf{x})$ with $r(\boldsymbol{\theta})$ the prior ratio.*

*Proof.* We can rewrite the target posterior $q(\boldsymbol{\theta} | \mathbf{x})$ as

$$q(\boldsymbol{\theta} | \mathbf{x}) \propto q(\boldsymbol{\theta})p(\mathbf{x} | \boldsymbol{\theta}) = \frac{q(\boldsymbol{\theta})}{p_{\text{train}}(\boldsymbol{\theta})} p_{\text{train}}(\boldsymbol{\theta})p(\mathbf{x} | \boldsymbol{\theta}) \propto \frac{q(\boldsymbol{\theta})}{p_{\text{train}}(\boldsymbol{\theta})} p(\boldsymbol{\theta} | \mathbf{x}) = r(\boldsymbol{\theta})p(\boldsymbol{\theta} | \mathbf{x}),$$

where the prior ratio $r(\boldsymbol{\theta}) \equiv \frac{q(\boldsymbol{\theta})}{p_{\text{train}}(\boldsymbol{\theta})}$ takes the role of an importance weighing function, and the above equality applies almost everywhere. □

## B.3 PROOF OF PROPOSITION 2

We provide here the extended statement, with explicit regularity conditions, and then the full proof.

**Proposition** (Proposition 2). *As $t, \sigma(t) \rightarrow 0$, the approximation $\mathcal{N}(\boldsymbol{\theta}_0 | \boldsymbol{\theta}_t + \sigma(t)^2 \nabla_{\boldsymbol{\theta}_t} \log p(\boldsymbol{\theta}_t), \frac{\sigma(t)^2}{1+\sigma(t)^2} \mathbf{I})$ converges to the true $p(\boldsymbol{\theta}_0 | \boldsymbol{\theta}_t)$, assuming that $p(\boldsymbol{\theta}_0)$ is two times differentiable everywhere and $\nabla_{\boldsymbol{\theta}_0}^2 p(\boldsymbol{\theta}_0)$ is bounded.*

**Notation** To be more precise about the different distributions involved, let us denote $p_0(\boldsymbol{\theta}_0)$ as the marginal distribution of the clean data, $p_t(\boldsymbol{\theta}_t)$ the marginal distribution at noise level $\sigma(t)$, $p_{t\,|\,0}(\boldsymbol{\theta}_t\,|\,\boldsymbol{\theta}_0) = \mathcal{N}(\boldsymbol{\theta}_t\,|\,\boldsymbol{\theta}_0, \sigma(t)^2\mathbf{I})$ and $p_{0\,|\,t}(\boldsymbol{\theta}_0\,|\,\boldsymbol{\theta}_t) = \frac{p_{t\,|\,0}(\boldsymbol{\theta}_t\,|\,\boldsymbol{\theta}_0)p_0(\boldsymbol{\theta}_0)}{p_t(\boldsymbol{\theta}_t)}$. Let us also drop the dependence $t$ from the notation $\sigma(t)$, and simply refer to $\sigma$, since we do not have to refer to derivatives of $\sigma(t)$ in the proof.

*Proof.* First, note that as $\sigma \to 0$, $\frac{\sigma^2}{1+\sigma^2} = \sigma^2 + O(\sigma^4)$ via a Taylor expansion. In other words, our denoising variance is $\sigma^2\mathbf{I}$ up to fourth-order or higher corrections in $\sigma$, which become negligible at low $\sigma$. We can thus first show the result for $\mathcal{N}(\boldsymbol{\theta}_0\,|\,\boldsymbol{\theta}_t + \sigma(t)^2\nabla_{\boldsymbol{\theta}_t}\log p(\boldsymbol{\theta}_t), \sigma^2\mathbf{I})$, and at the end we will see that it will trivially transfer to the $\frac{\sigma^2}{1+\sigma^2}\mathbf{I}$ case as well.

**Rescaled coordinates** Set

$$\mathbf{s} = \frac{\boldsymbol{\theta}_0 - \boldsymbol{\theta}_t}{\sigma}, \quad \phi_d(\mathbf{s}) = (2\pi)^{-d/2}\exp\left(-\frac{1}{2}||\mathbf{s}||^2\right) \tag{A14}$$

so that we can express the forward noising distribution as

$$p_{t\,|\,0}(\boldsymbol{\theta}_t\,|\,\boldsymbol{\theta}_0) = \sigma^{-d}\phi_d(\mathbf{s}) \tag{A15}$$

**Taylor expansion for the true posterior density** Since $p_0(\boldsymbol{\theta}_0)$ is two times differentiable everywhere, the multivariate Taylor theorem gives

$$p_0(\boldsymbol{\theta}_t + \sigma\mathbf{s}) = p_0(\boldsymbol{\theta}_t) + \sigma\nabla_{\boldsymbol{\theta}_t}p_0(\boldsymbol{\theta}_t)^\top\mathbf{s} + \frac{\sigma^2}{2}\mathbf{s}^\top\nabla^2_{\mathbf{b}(\mathbf{s})}p_0(\mathbf{b}(\mathbf{s}))\mathbf{s} \tag{A16}$$

where $\mathbf{b}(\mathbf{s}) \in \{t\mathbf{s} : 0 \leq t \leq 1\}$ is some point between $\boldsymbol{\theta}_t$ and $\boldsymbol{\theta}_0$, chosen separately for each $\mathbf{s}$.

Hence, the unnormalised posterior density can be written as

$$\sigma^{-d}\phi_d(\mathbf{s})\left[p_0(\boldsymbol{\theta}_t) + \sigma\nabla_{\boldsymbol{\theta}_t}p_0(\boldsymbol{\theta}_t)^\top\mathbf{s} + \frac{\sigma^2}{2}\mathbf{s}^\top\nabla^2_{\mathbf{b}(\mathbf{s})}p_0(\mathbf{b}(\mathbf{s}))\mathbf{s}\right] \tag{A17}$$

The normalizing constant can be expanded as

$$p_t(\boldsymbol{\theta}_t) = \int \sigma^{-d}\phi_d(\mathbf{s})\left[p_0(\boldsymbol{\theta}_t) + \sigma\nabla_{\boldsymbol{\theta}_t}p_0(\boldsymbol{\theta}_t)^\top\mathbf{s} + \frac{\sigma^2}{2}\mathbf{s}^\top\nabla^2_{\mathbf{b}(\mathbf{s})}p_0(\mathbf{b}(\mathbf{s}))\mathbf{s}\right]\sigma^d d\mathbf{s} \tag{A18}$$

$$= p_0(\boldsymbol{\theta}_t) + \sigma\underbrace{\int \phi_d(\mathbf{s})\nabla_{\boldsymbol{\theta}_t}p_0(\boldsymbol{\theta}_t)^\top\mathbf{s}d\mathbf{s}}_{=0} + \sigma^2\underbrace{\frac{1}{2}\int \phi_d(\mathbf{s})\mathbf{s}^\top\nabla^2_{\mathbf{b}(\mathbf{s})}p_0(\mathbf{b}(\mathbf{s}))\mathbf{s}d\mathbf{s}}_{=C_1(\boldsymbol{\theta}_t)} \tag{A19}$$

$$= p_0(\boldsymbol{\theta}_t) + \sigma^2 C_1(\boldsymbol{\theta}_t) \tag{A20}$$

where the odd-powered term in the Taylor expansion goes to zero due to the symmetry of the integral. The integral in the second-order term is finite since we assume $\nabla^2_{\mathbf{b}(\mathbf{s})}p_0(\mathbf{b}(\mathbf{s}))$ is finite everywhere.

The reciprocal of the normalizing constant is

$$\frac{1}{p_0(\boldsymbol{\theta}_t) + \sigma^2 C_1(\boldsymbol{\theta}_t)} = \frac{1}{p_0(\boldsymbol{\theta}_t)} - \frac{1}{p_0(\boldsymbol{\theta}_t)^2}\sigma^2 C_1(\boldsymbol{\theta}_t) + O(\sigma^4) \tag{A21}$$

obtained with the Taylor series $\frac{1}{a+\varepsilon} = \frac{1}{a} - \frac{1}{a^2}\varepsilon + O(\varepsilon^2)$. Thus, the normalised posterior can be expressed as

$$\frac{\sigma^{-d}\phi_d(\mathbf{s})\left[p_0(\boldsymbol{\theta}_t) + \sigma\nabla_{\boldsymbol{\theta}_t}p_0(\boldsymbol{\theta}_t)^\top\mathbf{s} + \frac{\sigma^2}{2}\mathbf{s}^\top\nabla^2_{\mathbf{b}(\mathbf{s})}p_0(\mathbf{b}(\mathbf{s}))\mathbf{s}\right]}{p_0(\boldsymbol{\theta}_t) + \sigma^2 C_1(\boldsymbol{\theta}_t)} \tag{A22}$$

$$= \sigma^{-d}\phi_d(\mathbf{s})\left[p_0(\boldsymbol{\theta}_t) + \sigma\nabla_{\boldsymbol{\theta}_t}p_0(\boldsymbol{\theta}_t)^\top\mathbf{s} + \frac{\sigma^2}{2}\mathbf{s}^\top\nabla^2_{\mathbf{b}(\mathbf{s})}p_0(\mathbf{b}(\mathbf{s}))\mathbf{s}\right]\left(\frac{1}{p_0(\boldsymbol{\theta}_t)} - \frac{1}{p_0(\boldsymbol{\theta}_t)^2}\sigma^2 C_1(\boldsymbol{\theta}_t) + O(\sigma^4)\right) \tag{A23}$$

$$= \sigma^{-d}\phi_d(\mathbf{s})\left[1 + \sigma\frac{\nabla_{\boldsymbol{\theta}_t}p_0(\boldsymbol{\theta}_t)^\top}{p_0(\boldsymbol{\theta}_t)}\mathbf{s} + \sigma^2\left(\frac{1}{2}\mathbf{s}^\top\nabla^2_{\mathbf{b}(\mathbf{s})}p_0(\mathbf{b}(\mathbf{s}))\mathbf{s} - \frac{1}{p_0(\boldsymbol{\theta}_t)}C_1(\boldsymbol{\theta}_t)\right) + O(\sigma^3)\right] \tag{A24}$$

$$= \sigma^{-d}\phi_d(\mathbf{s})\left[1 + \sigma\nabla_{\boldsymbol{\theta}_t}\log p_0(\boldsymbol{\theta}_t)^\top\mathbf{s} + \sigma^2 C_2(\mathbf{s}, \boldsymbol{\theta}_t) + O(\sigma^3)\right]. \tag{A25}$$

Here, we abstract away $C_2(\mathbf{s}, \boldsymbol{\theta}_t)$, since later in the proof we only care about the fact that it has a finite integral $\int |\phi_d(\mathbf{s}) C_2(\mathbf{s}, \boldsymbol{\theta}_t)| d\mathbf{s}$.

**Taylor expansion for our approximate posterior density** Note that the score $\nabla_{\boldsymbol{\theta}_t} \log p_t(\boldsymbol{\theta}_t) = \frac{\nabla_{\boldsymbol{\theta}_t} p_t(\boldsymbol{\theta}_t)}{p_t(\boldsymbol{\theta}_t)}$, and we can reuse our earlier calculation:

$$\nabla_{\boldsymbol{\theta}_t} \log p_t(\boldsymbol{\theta}_t) = \frac{\nabla_{\boldsymbol{\theta}_t} p_0(\boldsymbol{\theta}_t) + \sigma^2 \nabla_{\boldsymbol{\theta}_t} C_1(\boldsymbol{\theta}_t)}{p_0(\boldsymbol{\theta}_t) + \sigma^2 C_1(\boldsymbol{\theta}_t)} \tag{A26}$$

$$= \left(\nabla_{\boldsymbol{\theta}_t} p_0(\boldsymbol{\theta}_t) + \sigma^2 \nabla_{\boldsymbol{\theta}_t} C_1(\boldsymbol{\theta}_t)\right) \left(\frac{1}{p_0(\boldsymbol{\theta}_t)} - \frac{1}{p_0(\boldsymbol{\theta}_t)^2} \sigma^2 C_1(\boldsymbol{\theta}_t) + O(\sigma^4)\right) \tag{A27}$$

$$= \frac{\nabla_{\boldsymbol{\theta}_t} p_0(\boldsymbol{\theta}_t)}{p_0(\boldsymbol{\theta}_t)} + \sigma^2 \left(\nabla_{\boldsymbol{\theta}_t} C_1(\boldsymbol{\theta}_t) - \frac{\nabla_{\boldsymbol{\theta}_t} p_0(\boldsymbol{\theta}_t)}{p_0(\boldsymbol{\theta}_t)^2} C_1(\boldsymbol{\theta}_t)\right) + O(\sigma^4) \tag{A28}$$

$$= \nabla_{\boldsymbol{\theta}_t} \log p_0(\boldsymbol{\theta}_t) + O(\sigma^2). \tag{A29}$$

This yields a formula for our posterior mean:

$$\boldsymbol{\theta}_t + \sigma^2 \nabla_{\boldsymbol{\theta}_t} \log p_t(\boldsymbol{\theta}_t) = \boldsymbol{\theta}_t + \sigma^2 \nabla_{\boldsymbol{\theta}_t} \log p_0(\boldsymbol{\theta}_t) + O(\sigma^4). \tag{A30}$$

Now, define:

$$q(\boldsymbol{\theta}_0) = \mathcal{N}(\boldsymbol{\theta}_0 \,|\, \boldsymbol{\theta}_t + \sigma^2 \nabla_{\boldsymbol{\theta}_t} \log p(\boldsymbol{\theta}_t), \sigma^2 \mathbf{I}) \tag{A31}$$

$$= \sigma^{-d} \phi_d(s - \sigma \nabla_{\boldsymbol{\theta}_t} \log p(\boldsymbol{\theta}_t)), \quad s = \frac{\boldsymbol{\theta}_0 - \boldsymbol{\theta}_t}{\sigma}. \tag{A32}$$

Then we can Taylor expand our posterior approximation in terms of the shift $\mathbf{s} \to \mathbf{s} - \sigma \nabla_{\boldsymbol{\theta}_t} \log p(\boldsymbol{\theta}_t)$:

$$q(\sigma \mathbf{s} + \boldsymbol{\theta}_t) = \sigma^{-d} \phi_d(\mathbf{s}) [1 + \sigma \nabla_{\boldsymbol{\theta}_t} \log p_t(\boldsymbol{\theta}_t)^\top \mathbf{s}$$
$$+ \frac{1}{2} \sigma^2 \nabla_{\boldsymbol{\theta}_t} \log p_t(\boldsymbol{\theta}_t)^\top (\mathbf{s}\mathbf{s}^\top - I) \nabla_{\boldsymbol{\theta}_t} \log p_t(\boldsymbol{\theta}_t) + O(\sigma^3)] \tag{A33}$$

$$= \sigma^{-d} \phi_d(\mathbf{s}) [1 + \sigma \nabla_{\boldsymbol{\theta}_t} \log p_0(\boldsymbol{\theta}_t)^\top \mathbf{s}$$
$$+ \frac{1}{2} \sigma^2 \nabla_{\boldsymbol{\theta}_t} \log p_0(\boldsymbol{\theta}_t)^\top (\mathbf{s}\mathbf{s}^\top - I) \nabla_{\boldsymbol{\theta}_t} \log p_0(\boldsymbol{\theta}_t) + O(\sigma^3)] \tag{A34}$$

$$= \sigma^{-d} \phi_d(\mathbf{s}) [1 + \sigma \nabla_{\boldsymbol{\theta}_t} \log p_0(\boldsymbol{\theta}_t)^\top \mathbf{s} + \sigma^2 C_3(\mathbf{s}, \boldsymbol{\theta}_t) + O(\sigma^3)]. \tag{A35}$$

where in the second line we used Eq. (A29) and ignored resulting terms that are higher order than $\sigma^2$.

**Total variation bound**

The total variation distance is

$$\|p_{0\,|\,t} - q\|_{\mathrm{TV}} = \frac{1}{2} \int_{\mathbb{R}^d} |p_{0\,|\,t}(\boldsymbol{\theta}_0) - q(\boldsymbol{\theta}_0)| \, d\boldsymbol{\theta}_0. \tag{A36}$$

Now, plugging in Eq. (A25) and Eq. (A35), we get (note $d\boldsymbol{\theta}_0 = \sigma^d d\mathbf{s}$)

$$\frac{1}{2} \int \Big| \sigma^{-d} \phi_d(\mathbf{s}) \left[1 + \sigma \nabla_{\boldsymbol{\theta}_t} \log p_0(\boldsymbol{\theta}_t)^\top \mathbf{s} + \sigma^2 C_2(\mathbf{s}, \boldsymbol{\theta}_t) + O(\sigma^3)\right] \tag{A37}$$

$$- \sigma^{-d} \phi_d(\mathbf{s}) [1 + \sigma \nabla_{\boldsymbol{\theta}_t} \log p_0(\boldsymbol{\theta}_t)^\top \mathbf{s} + \sigma^2 C_3(\mathbf{s}, \boldsymbol{\theta}_t) + O(\sigma^3)] \Big| \sigma^d d\mathbf{s} \tag{A38}$$

$$= \frac{1}{2} \int \Big| \sigma^2 \phi_d(\mathbf{s}) (C_2(\mathbf{s}, \boldsymbol{\theta}_t) - C_3(\mathbf{s}, \boldsymbol{\theta}_t) + O(\sigma^3)) \Big| d\mathbf{s} \le \sigma^2 C_4(\boldsymbol{\theta}_t) + O(\sigma^3). \tag{A39}$$

Thus, the total variation distance converges at a rate $\sigma^2$. To see the final step more clearly, we can bound the integral with the triangle inequality:

$$\int \Big| \sigma^2 \phi_d(\mathbf{s}) (C_2(\mathbf{s}, \boldsymbol{\theta}_t) - C_3(\mathbf{s}, \boldsymbol{\theta}_t) + O(\sigma^3)) \Big| d\mathbf{s} \tag{A40}$$

$$\le \sigma^2 \int \big| \phi_d(\mathbf{s}) C_2(\mathbf{s}, \boldsymbol{\theta}_t) \big| d\mathbf{s} + \sigma^2 \int \big| \phi_d(\mathbf{s}) C_3(\mathbf{s}, \boldsymbol{\theta}_t) \big| d\mathbf{s} + O(\sigma^3). \tag{A41}$$

$$\tag{A42}$$

Recall the definitions of $C_3, C_2$ and $C_1$:

$$C_1(\boldsymbol{\theta}_t) = \frac{1}{2}\int \phi_d(\mathbf{s})\mathbf{s}^\top \nabla^2_{\mathbf{b(s)}}p_0(\mathbf{b(s)})\mathbf{s}\,d\mathbf{s} \tag{A43}$$

$$C_2(\mathbf{s}, \boldsymbol{\theta}_t) = \left(\frac{1}{2}\mathbf{s}^\top \nabla^2_{\mathbf{b(s)}}p_0(\mathbf{b(s)})\mathbf{s} - \frac{1}{p_0(\boldsymbol{\theta}_t)}C_1(\boldsymbol{\theta}_t)\right) \tag{A44}$$

$$\frac{1}{2}C_3(\mathbf{s}, \boldsymbol{\theta}_t) = \nabla_{\boldsymbol{\theta}_t}\log p_0(\boldsymbol{\theta}_t)^\top(\mathbf{s}\mathbf{s}^\top - I)\nabla_{\boldsymbol{\theta}_t}\log p_0(\boldsymbol{\theta}_t). \tag{A45}$$

We can see that the terms depending on $\sigma^2$ are finite, due to the assumption that the Hessian of $p_0$ is finite everywhere. Thus,

$$C_4(\boldsymbol{\theta}_t) = \int \big|\phi_d(\mathbf{s})(C_2(\mathbf{s}, \boldsymbol{\theta}_t)\big|d\mathbf{s} + \int \big|\phi_d(\mathbf{s})(C_3(\mathbf{s}, \boldsymbol{\theta}_t)\big|d\mathbf{s}. \tag{A46}$$

Finally, we note that $\frac{\sigma^2}{1+\sigma^2} \to \sigma^2$ as $\sigma \to 0$. Thus, the the convergence result here applies to the case where the posterior covariance approximation is $\frac{\sigma^2}{1+\sigma^2}\mathbf{I}$. $\square$

## C EXPERIMENTAL DETAILS

This section provides extended methodological and experimental details. We describe the simulator models used in our experiments (Appendix C.1), the training setup for each method (Appendix C.2), the testing procedure (Appendix C.3), and the statistical methodology for evaluation (Appendix C.4).

### C.1 SIMULATORS

**Two Moons** (Greenberg et al., 2019) is a widely used benchmark in SBI, designed as a two-dimensional task that presents a posterior distribution with both global (bimodality) and local (crescent shape) structure. For a given parameter vector $\boldsymbol{\theta} = (\theta_1, \theta_2) \in \mathbb{R}^2$, the simulator generates data $\mathbf{x} \in \mathbb{R}^2$ according to the following process:

$$\begin{aligned} a &\sim U(-\pi/2, \pi/2), \\ r &\sim \mathcal{N}(0.1, 0.01^2), \\ p &= (r\cos(a) + 0.25, r\sin(a)), \\ \mathbf{x}^T &= p + \left(\frac{-|\theta_1 + \theta_2|}{\sqrt{2}}, \frac{-\theta_1 + \theta_2}{\sqrt{2}}\right). \end{aligned}$$

The training prior we use is $p_{\text{train}}(\boldsymbol{\theta}) = U([-1, 1]^2)$. To obtain ground-truth posterior samples, we perform rejection sampling using the new prior as a proposal distribution. Rejection sampling relies on finding a constant $M$ such that $f(\boldsymbol{\theta}) \leq Mq(\boldsymbol{\theta})$, where $f(\boldsymbol{\theta}) = p(\mathbf{x}\,|\,\boldsymbol{\theta}) \cdot q(\boldsymbol{\theta})$ is the target density and $q(\boldsymbol{\theta})$ is the proposal density. In this Two Moons model, we set $M$ to be the upper bound of the likelihood $p(\mathbf{x}\,|\,\boldsymbol{\theta})$ which is achieved at $r = 0.1$.

**Ornstein-Uhlenbeck Process (OUP)** (Uhlenbeck & Ornstein, 1930) is a well-established stochastic process frequently applied in financial mathematics and evolutionary biology for modeling mean-reverting dynamics (Uhlenbeck & Ornstein, 1930). The model is defined as:

$$y_{t+1} = y_t + \Delta y_t, \quad \Delta y_t = \theta_1\left[\exp(\theta_2) - y_t\right]\Delta t + 0.5w, \quad \text{for } t = 1, \dots, T,$$

where we set $T = 25$, $\Delta t = 0.2$, and initialize $x_0 = 10$. The noise term follows a Gaussian distribution, $w \sim \mathcal{N}(0, \Delta t)$. The original prior is a uniform distribution, $U([0, 2] \times [-2, 2])$, over the latent parameters $\boldsymbol{\theta} = (\theta_1, \theta_2)$. For numerical convenience, we reparameterize the parameter space by mapping the original parameters to $[-1, 1]^2$, yielding $p_{\text{train}}(\boldsymbol{\theta}) = U([-1, 1]^2)$. We perform inference in this normalized parameter space and later rescale the sampled parameters to the original space during simulation. The simulated data are also normalized using standardization. We use normalized parameter-data pairs $(\tilde{\boldsymbol{\theta}}, \tilde{\mathbf{x}})$ to train all amortized inference models.

Since the OUP likelihood is implicit, obtaining ground-truth posterior samples is intractable—the kind of problem that requires simulation-based inference. As a practical surrogate, we adopt a neural

posterior estimation (NPE) model (Greenberg et al., 2019)—trained on *one million* simulations (compared to the 10,000 simulations used for our diffusion-based inference model)—to serve as ground truth. We then applied simulation-based calibration (SBC) (Talts et al., 2018) to verify that the NPE model remained well-calibrated across virtually all observation seeds for each new prior.

**Turin** (Turin et al., 1972) is a widely used time-series model for simulating radio wave propagation (Turin et al., 1972; Pedersen, 2019). This model generates high-dimensional, complex-valued time-series data and is governed by four key parameters: $G_0$ determines the reverberation gain, $T$ controls the reverberation time, $\lambda_0$ defines the arrival rate of the point process, and $\sigma_N^2$ represents the noise variance.

The model assumes a frequency bandwidth of $B = 0.5$ GHz and simulates the transfer function $H_k$ at $N_s = 101$ evenly spaced frequency points. The observed transfer function at the $k$-th frequency point, $Y_k$, is defined as:
$$Y_k = H_k + W_k, \quad k = 0, 1, \ldots, N_s - 1,$$
where $W_k$ represents additive zero-mean complex Gaussian noise with circular symmetry and variance $\sigma_W^2$. The transfer function $H_k$ is expressed as:

$$H_k = \sum_{l=1}^{N_{\text{points}}} \alpha_l \exp(-j2\pi\Delta f k \tau_l),$$

where the time delays $\tau_l$ are sampled from a homogeneous Poisson point process with rate $\lambda_0$, and the complex gains $\alpha_l$ are modeled as independent zero-mean complex Gaussian random variables. The conditional variance of the gains is given by:

$$\mathbb{E}[|\alpha_l|^2|\tau_l] = \frac{G_0 \exp(-\tau_l/T)}{\lambda_0}.$$

To obtain the time-domain signal $\tilde{y}(t)$, an inverse Fourier transform is applied:

$$\tilde{y}(t) = \frac{1}{N_s} \sum_{k=0}^{N_s-1} Y_k \exp(j2\pi k \Delta f t),$$

where $\Delta f = B/(N_s - 1)$ represents the frequency spacing. Finally, the real-valued output is computed by taking the absolute square of the complex signal and applying a logarithmic transformation:

$$y(t) = 10 \log_{10}(|\tilde{y}(t)|^2).$$

In Turin, the true parameter bounds are: $G_0 \in [10^{-9}, 10^{-8}]$, $T \in [10^{-9}, 10^{-8}]$, $\lambda_0 \in [10^7, 5 \times 10^9]$, $\sigma_N^2 \in [10^{-10}, 10^{-9}]$. We follow a similar normalization setup as in OUP. First, we define the training prior as $p_{\text{train}}(\boldsymbol{\theta}) = U([0,1]^4)$ and rescale the sampled parameters $\tilde{\boldsymbol{\theta}}$ to the original space using the true parameter bounds. Then, we normalize the simulator outputs using standardization and use normalized $(\tilde{\boldsymbol{\theta}}, \tilde{\mathbf{x}})$ pairs to train all inference models.

The Turin likelihood is also implicit. Therefore, we use a similar setup to the OUP case by training an NPE model with one million simulations and validating its reliability using SBC.

**Gaussian Linear** (Lueckmann et al., 2021) is a standard SBI benchmark task used to infer the mean of a multivariate Gaussian distribution when the covariance is fixed. In this model, both the parameters $\boldsymbol{\theta}$ and the data $\mathbf{x}$ are 10-dimensional vectors. The simulator is defined as:

$$\mathbf{x}|\boldsymbol{\theta} \sim \mathcal{N}(\boldsymbol{\theta}, \Sigma_s),$$

where $\Sigma_s = 0.1 \cdot \mathbf{I}_{10}$ with $\mathbf{I}_{10}$ is the 10-dimensional identity matrix. The training prior for the parameters $\boldsymbol{\theta}$ is a 10-dimensional Gaussian distribution $p_{\text{train}}(\boldsymbol{\theta}) = \mathcal{N}(\mathbf{0}, 0.1 \cdot \mathbf{I}_{10})$.

We also consider a 20-dimensional variant of the Gaussian Linear model, which follows the same fundamental setup as its 10-dimensional counterpart. In this version, both the parameters $\boldsymbol{\theta}$ and the data $\mathbf{x}$ are 20-dimensional vectors.

In all experiments, the test-time priors are constructed as Gaussian or mixture distributions with Gaussian components. Consequently, each new prior has a closed-form posterior, which we use as the ground-truth posterior.

**Bayesian Causal Inference model (BCI)** is a common model in computational cognitive neuroscience to represent how the brain determines whether multiple sensory stimuli originate from a common source (Körding et al., 2007). This BCI model simulates an observer's responses in an audiovisual localization task. The observer is presented with auditory ($S_A$) and visual ($S_V$) spatial cues—expressed as horizontal location in degrees of visual angle—and must report the perceived location of one of them. The model assumes the observer performs Bayesian causal inference to determine whether the cues originate from a common source or independent sources, and then makes a model-averaged estimate of the target stimulus location (Körding et al., 2007; Acerbi et al., 2018).

In this BCI model we consider five underlying physical parameters: standard deviation of visual sensory noise ($\sigma_V$), standard deviation of auditory sensory noise ($\sigma_A$), standard deviation of the Gaussian spatial prior over source locations ($\sigma_s$), prior probability that auditory and visual cues share a common cause ($p_{\text{same}}$), and standard deviation of motor noise in the response ($\sigma_m$). We assume the mean of the Gaussian spatial prior $\mu_p$ is set to 0 (central tendency), and a small lapse rate $\lambda = 0.02$, the probability of making a random response. Additionally, a fixed auditory rescaling factor $\rho_A = 4/3$ is applied to auditory stimulus locations to account for audiovisual adaptation in this experiment.

The simulation process for each trial $i$ (out of 98 fixed trials) proceeds as follows:

1. Given true stimulus locations $S_{V,i}$ and $S_{A,i}$, and the modality to be reported (visual or auditory).

2. Sensory measurements $x_{V,i}$ and $x_{A,i}$ are drawn:

$$x_{V,i} \sim \mathcal{N}(S_{V,i}, \sigma_V^2)$$
$$x_{A,i} \sim \mathcal{N}(\rho_A S_{A,i}, \sigma_A^2)$$

3. The observer combines these measurements with a spatial prior $p(s) = \mathcal{N}(s; \mu_p, \sigma_s^2)$.

4. Causal inference is performed to compute the posterior probability of a common source, $P(C = 1|x_{V,i}, x_{A,i})$, using the prior $p_{\text{same}}$ and the likelihoods of the measurements under common-cause ($C = 1$) and independent-causes ($C = 2$) hypotheses.

5. A model-averaged estimate of the relevant source location, $\hat{s}_i$, is computed:

$$\hat{s}_i = P(C = 1|x_{V,i}, x_{A,i}) \cdot \mu_{C=1}(x_{V,i}, x_{A,i})$$
$$+ (1 - P(C = 1|x_{V,i}, x_{A,i})) \cdot \mu_{C=2}(x_{V,i}, x_{A,i}, \text{report\_modality}_i),$$

   where $\mu_{C=1}$ and $\mu_{C=2}$ are the posterior mean estimates under the respective causal hypotheses.

6. A noisy motor response $R_i'$ is generated: $R_i' \sim \mathcal{N}(\hat{s}_i, \sigma_m^2)$.

7. With probability $\lambda$, the response $R_i'$ is replaced by a lapse response drawn uniformly from $U(-45°, 45°)$. The final response is $R_i$.

The output data $\mathbf{x}$ is a 98-dimensional vector $(R_1, \ldots, R_{98})$. These responses correspond to a fixed experimental design: a $7 \times 7$ Cartesian grid of stimulus locations $S_V, S_A \in \{-15, -10, -5, 0, 5, 10, 15\}°$. For each of the 49 unique $(S_V, S_A)$ pairs, one visual-report trial and one auditory-report trial are included, totaling 98 trials.

The model's five parameters are internally represented in an unconstrained space: $\sigma_V, \sigma_A, \sigma_s, \sigma_m$ are parameterized by their logarithms, and $p_{\text{same}}$ by its logit, denoted by the vector $\boldsymbol{\theta}$:

$$\boldsymbol{\theta} = (\theta_1, \theta_2, \theta_3, \theta_4, \theta_5) = (\log \sigma_V, \log \sigma_A, \log \sigma_s, \log \sigma_m, \text{logit } p_{\text{same}}). \quad \text{(A47)}$$

The original prior over these 5 unconstrained parameters follows a broad multivariate Gaussian distribution informed by the literature, with independent components:

$$\theta_1, \theta_2 \sim \mathcal{N}(\log 2, 0.35^2), \ \theta_3 \sim \mathcal{N}(\log 5, 0.5^2), \ \theta_4 \sim \mathcal{N}(\log 0.3, 0.35^2), \ \theta_5 \sim \mathcal{N}(0, 1^2). \quad \text{(A48)}$$

For numerical convenience, we reparameterize the parameter space by defining the training prior as a 5-dimensional Gaussian distribution $p_{\text{train}}(\boldsymbol{\theta}) = \mathcal{N}(\mathbf{0}, \mathbf{I}_5)$. Similar to the setup in OUP and Turin, we rescale the sampled parameters to the original space during simulation and normalize the

Table A1: Characteristics of the simulator models.

| Model | $\dim(\boldsymbol{\theta})$ | $\dim(\mathbf{x})$ | $p_{\text{train}}(\boldsymbol{\theta})$ |
|---|---|---|---|
| Two-Moons | 2 | 2 | Uniform |
| OUP | 2 | 25 | Uniform |
| Turin | 4 | 101 | Uniform |
| Gaussian Linear 10D | 10 | 10 | Gaussian |
| Gaussian Linear 20D | 20 | 20 | Gaussian |
| BCI | 5 | 98 | Gaussian |

simulated data using standardization. We use normalized parameter-data pairs $(\tilde{\boldsymbol{\theta}}, \tilde{\mathbf{x}})$ to train all amortized inference models.

The likelihood function for this model involves integrating over latent sensory measurements and is computationally intensive. To obtain ground-truth posterior samples for BCI, we run the Variational Bayesian Monte Carlo (VBMC) algorithm (Acerbi, 2018; Huggins et al., 2023). Using VBMC's internal diagnostics, we retain ten reliable variational posteriors and merge them via posterior stacking (Silvestrin et al., 2025) to obtain the final ground-truth posterior.

Table A1 summarizes the key properties of the simulator models in our experiments.

## C.2 TRAINING SETUP

For all methods that require training, we use 10,000 simulated datasets from the simulator to train the model. Note that PriorGuide is a test-time technique that does not require separate training. For PriorGuide, we use the same base diffusion model as Simformer. Details on the model configurations and dataset setups are provided below.

**Simformer** We adopt a similar setup as the Simformer paper Gloeckler et al. (2024), using the Variance Exploding Stochastic Differential Equation (VE-SDE) technique Song et al. (2021). It is defined by

$$f_{\text{VE-SDE}}(x,t) = 0, \qquad g_{\text{VE-SDE}}(t) = \sigma_{\min}\left(\frac{\sigma_{\max}}{\sigma_{\min}}\right)^t \sqrt{2\log\frac{\sigma_{\max}}{\sigma_{\min}}}. \tag{4}$$

Throughout all experiments, we set $\sigma_{\max} = 15$, $\sigma_{\min} = 10^{-4}$, and run the process over the time interval $t \in [10^{-5}, 1]$. We use a transformer configuration similar to Simformer Gloeckler et al. (2024), with 6 layers, 4 heads (size 10), a token dimension 40, and a 128-dimensional Gaussian Fourier embedding for diffusion time. MLP blocks use a hidden dimension of 150. In all experiments, the condition mask was sampled per batch by uniformly selecting one of the following: joint, posterior, likelihood, or two random masks. Random masks were drawn from Bernoulli distributions with $p = 0.3$ and $p = 0.7$, respectively. We use the same setup for all of the simulators.

We train all the Simformer models using a batch size of 1,000 and an initial learning rate of 0.001. A linear learning rate schedule is used to decay the learning rate to $1 \times 10^{-6}$, starting at half of the total number of training steps and completing by the final step. The optimizer combines adaptive gradient clipping with a maximum norm of 10.0 and the Adam optimizer Kingma & Ba (2015). Early stopping is applied based on validation loss, with the number of training steps constrained to a minimum of 5,000 and a maximum of 100,000 steps.

**Amortized Conditioning Engine (ACE)** ACE (Chang et al., 2025) is a type of Neural Process (NP) (Garnelo et al., 2018; Nguyen & Grover, 2022; Müller et al., 2022), a family of models that learn to perform amortized inference by conditioning on a *context set* of input-output pairs to predict outputs for a *target set* of inputs. Differently from other neural processes which focus on pure data prediction, ACE is trained to condition on, and predict, both data and latent variables (*e.g.*, model parameters in the case of SBI). During training, ACE was provided with simulator parameters that were randomly assigned to either the context or target set, so the model learns to generalize across varying observational conditions. For each experiment, a random number of data points $N_d$ were sampled for the context set, with the remaining used as targets. Specifically, $N_d \sim U(1,2)$ for Two-Moons, $N_d \sim U(7,25)$ for OUP, $N_d \sim U(30,101)$ for Turin, $N_d \sim U(3,10)$ for Gaussian Linear, and $N_d \sim U(29,98)$ for BCI.

Furthermore, ACE can be trained with a meta-prior (a distribution of priors) over latent variables, where each prior is expressed as a factorized histogram, affording amortized test-time prior adaptation. For the ACE baseline, we use the same prior generation process as the original ACE paper (Chang et al., 2025), which constructs diverse priors through a hierarchical method over a bounded range. For each latent variable $\theta_l$, the prior is sampled either from a mixture of Gaussians (with 80% probability) or a uniform distribution. When using a Gaussian mixture, the number of components $K$ is drawn from a geometric distribution with parameter $q = 0.5$, and one of three configurations is chosen at random: shared means with varying standard deviations, varying means with shared standard deviations, or both means and standard deviations varying. Means and standard deviations are sampled from predefined uniform distributions based on the latent variable's range, and mixture weights are drawn from a Dirichlet distribution with $\alpha_0 = 1$. The resulting mixture is discretized into a histogram over $N_{\text{bins}} = 100$ uniform bins by evaluating CDF differences at bin edges and normalizing. If a uniform prior is selected, equal probability is assigned to all bins.

For the network setup, we use a configuration similar to that of the ACE paper Chang et al. (2025). The ACE model has a 64-dimensional embedding, 6 transformer layers, 4 attention heads, and MLP blocks with hidden dimension 128. The output head includes 20 MLP components with hidden dimension 128. Training was carried out in $5 \times 10^4$ steps with batch size 32 and learning rate $5 \times 10^{-4}$ using the Adam optimizer with the cosine annealing scheduler.

## C.3 Testing procedure

### C.3.1 Test-time prior generations

We provide additional details about generating test-time priors $q$ in different scenarios. All the generated test-time priors have passed the OOD diagnostic detailed in Appendix A.4 with a quantile threshold $\alpha = 0.001$.

**Training prior definition**  For *uniform* training priors, let $p_{\text{train}}$ be defined as a uniform distribution over a $D$-dimensional hypercube $\prod_{i=1}^{D}[a_i, b_i]$. We define $s_i = (b_i - a_i)/\sqrt{12}$ for each dimension (the standard deviation of a uniform distribution over the range). For *Gaussian* training priors, $p_{\text{train}}$ is a multivariate normal distribution with diagonal covariance, with mean $m_i$ and standard deviation $s_i$ along each dimension.

**Target priors**  We first consider the cases where the test-time prior $q$ is a multivariate Gaussian distribution $q(\boldsymbol{\theta}) = \mathcal{N}(\boldsymbol{\theta} \,|\, \boldsymbol{\mu}, \boldsymbol{\sigma}^2 \mathbf{I})$. The procedure to generate a test prior is as follows, separately for each dimension $1 \le i \le D$:

- We first set the standard deviation $\sigma_i^{\text{mild}} = 0.5 \cdot s_i$ and $\sigma_i^{\text{strong}} = 0.2 \cdot s_i$ for mildly informative and strongly informative priors, respectively.

- We define the sampling lower ($L_i$) and upper ($U_i$) bounds for the mean of the target prior based on the class of training prior:

$$L_i = \begin{cases} a_i + 3\sigma_i & \text{(Uniform)} \\ m_i - 3s_i & \text{(Gaussian)} \end{cases}, \qquad U_i = \begin{cases} b_i - 3\sigma_i & \text{(Uniform)} \\ m_i + 3s_i & \text{(Gaussian)} \end{cases}. \tag{A49}$$

- With the bounds chosen, we sample the target prior mean for each coordinate, $\mu_i$, from

$$\mu_i \sim U(L_i, U_i). \tag{A50}$$

This procedure ensures that the target prior is well-overlapping with the training prior.

Finally, we define

$$q_{\text{mild}}(\boldsymbol{\theta}) = \mathcal{N}(\boldsymbol{\theta} \,|\, \boldsymbol{\mu}, \boldsymbol{\Sigma}^{\text{mild}}), \qquad q_{\text{strong}}(\boldsymbol{\theta}) = \mathcal{N}(\boldsymbol{\theta} \,|\, \boldsymbol{\mu}, \boldsymbol{\Sigma}^{\text{strong}}) \tag{A51}$$

where $\boldsymbol{\Sigma}^{\text{mild}} = \text{diag}\left[\sigma_1^{\text{mild}\,2}, \ldots, \sigma_D^{\text{mild}\,2}\right]$ and $\boldsymbol{\Sigma}^{\text{strong}} = \text{diag}\left[\sigma_1^{\text{strong}\,2}, \ldots, \sigma_D^{\text{strong}\,2}\right]$.

We then consider $q$ as a mixture Gaussian distribution with two components

$$q_{\text{mixture}}(\boldsymbol{\theta}) = \pi \mathcal{N}(\boldsymbol{\theta} \,|\, \boldsymbol{\mu}_1, \boldsymbol{\Sigma}^{\text{strong}}) + (1 - \pi)\mathcal{N}(\boldsymbol{\theta} \,|\, \boldsymbol{\mu}_2, \boldsymbol{\Sigma}^{\text{strong}}). \tag{A52}$$

We sample the mean vector for each component using the above procedures with *strongly informative* standard deviations. The mixture weights are sampled from $\pi \sim U(0.2, 0.8)$.

### C.3.2 POSTERIOR INFERENCE

For each prior type—*mild*, *strong* and *mixture*—we construct 10 priors $q_i$ following the procedure detailed in Appendix C.3.1. For each target prior $q_i$, we draw 10 parameter vectors $\boldsymbol{\theta}_{ij}$. For each parameter vector $\boldsymbol{\theta}_{ij}$, we simulate one observed dataset, yielding a pair $(\boldsymbol{\theta}_{ij}, \mathbf{x}_{ij})$. For a given tested method, we perform posterior inference by drawing 1,000 posterior samples $\boldsymbol{\theta}$ using the method conditioned on each $\mathbf{x}_{ij}$, repeating this evaluation across 5 independent training runs. Consequently, each amortized inference model produces $5 \times 10 \times 10 = 500$ collections of 1,000 $\boldsymbol{\theta}$ posterior samples for each prior type.

**Remark on prior specification**   By sampling the ground-truth parameters $\boldsymbol{\theta}_{ij}$ from the target prior $q_i$, we are ensuring that these priors are *informative* or well-specified, *i.e.*, the prior used for inference matches the true data generation process (Gelman et al., 2013). This is *not* a requirement of our method—PriorGuide can steer sampling towards any user-specified prior at test time (under the assumptions mentioned in the paper), steering the inference process towards the true Bayesian posterior under the given prior. Notably, a well-specified prior (or any reasonably specified prior) will *on average* improve inference accuracy, while a badly specified prior might hinder performance—this is a general property of Bayesian inference, which is reflected in PriorGuide. To avoid potential confusion, in this paper we focused on well-specified priors.

**Simformer**   With a trained Simformer model $\mathcal{M}$, we perform the same generation procedure as (Gloeckler et al., 2024), with a number of diffusion steps $N = 500$.

**PriorGuide**   Following Algorithm 1, we use the same set of core hyperparameters across all simulators—Two-Moons, OUP, Turin, Gaussian Linear, and BCI—unless otherwise noted. These include a trained diffusion-based inference model $\mathcal{M}$ (same as Simformer above), a training prior distribution $p_{\text{train}}(\boldsymbol{\theta})$, and a test-time prior distribution $q(\boldsymbol{\theta})$. The diffusion process is controlled by a minimum time $T_{\min} = 1 \times 10^{-10}$, a maximum time $T_{\max} = 1.0$, a generation schedule nonlinearity parameter $\rho = 2$ and a Langevin ratio $\eta = 0.5$. For Two Moons, OUP, Turin and Gaussian Linear, we set the number of diffusion steps as $N = 25$ and the number of Langevin steps as $N_L = 8$. For BCI, we set $N = 250$ and $N_L = 0$. In our experiments, we found that PriorGuide works better on BCI with a relatively large number of diffusion steps. The test-time compute across all simulators resides in the regime of $10^2$ NFEs (number of function evaluations, *i.e.* calls to the score model).

As detailed in Section 3 of the main text, we approximate the prior ratio $r(\boldsymbol{\theta})$ by fitting a Gaussian mixture model (cf. FITGMM in Algorithm 1). For Two Moons, OUP and Turin, we do not need to fit a GMM since the training prior $p_{\text{train}}$ is a uniform distribution, thus $r(\boldsymbol{\theta}) = q(\boldsymbol{\theta})$. For this reason, we can directly use the target prior $q(\boldsymbol{\theta})$ as the GMM parameterization for $r(\boldsymbol{\theta})$. For Gaussian Linear and BCI, which have a Gaussian training prior, we fit a GMM with $K = 20$ components to the ratio $r(\boldsymbol{\theta})$ by optimizing the L2 loss with gradient descent. For numerical convenience, we parameterize the GMM components with diagonal covariance matrices.[6] The optimization is carried out in $1 \times 10^6$ steps with batch size 1,000 and learning rate 0.01 using the Adam optimizer (Kingma & Ba, 2015). We verified both via the L2 loss and via visualizations that the GMM fits achieved a satisfactory ratio approximation.

**Amortized conditioning engine**   In ACE, first, we convert the test-time prior $q(\boldsymbol{\theta})$ generated in Appendix C.3.1 into a binned histogram distribution using the same steps discussed in Appendix C.2. Using the offline true data we have already sampled, we then condition on the full data $(x)$ in the context set and predict the parameters $\theta$ independently to obtain the posterior. Using this posterior distribution, which is a Gaussian Mixture Model (GMM), we then sample $\theta$ according to the number of posterior samples specified earlier. Note that ACE only supports factorized priors, so the *mixture* prior $q$ cannot be specified correctly within ACE (*i.e.*, ACE will represent the prior as a product of marginals).

### C.3.3 POSTERIOR PREDICTIVE INFERENCE

In the data prediction task (posterior predictive), with equal probability we condition on the first 30% or the last 30% of the values in $\mathbf{x}$, and then predict the remaining portion of $\mathbf{x}$. Note that in this task, we do not condition on $\boldsymbol{\theta}$. We draw 100 samples from the data predictions for evaluation.

---

[6]Note that PriorGuide also supports full covariance matrix representations for the GMM components of $r(\boldsymbol{\theta})$.

**Simformer and PriorGuide** We use the same hyperparameter settings and a similar procedure as in the posterior inference setup (Appendix C.3.2). In this case, the diffusion inference model will condition on the selected $30\%$ values of $\mathbf{x}$ to generate the remaining $70\%$. The steps for performing posterior predictive inference with PriorGuide are outlined in Algorithm 2.

**Amortized conditioning engine** For ACE, similar to the posterior inference setup, we first convert the prior into a binned histogram distribution. We then place the portion of $\mathbf{x}$ to be conditioned on into the context set. Data prediction is performed autoregressively by first randomly permuting the order of the target data points. This procedure is adapted from Bruinsma et al. (2023), as ACE is a neural process-based method. Consistent with the findings in Bruinsma et al. (2023), we observed that using random ordering for the autoregressive procedure yields the most robust performance.

### C.3.4 PARETO FRONTIERS OF TEST-TIME COMPUTE ASSIGNMENTS

PriorGuide supports improving the sampling quality by adding Langevin dynamic steps to the diffusion process, at the cost of additional test-time compute. With the same testing setup described above (Appendix C.3.2), we applied PriorGuide to posterior inference in the OUP and Turin simulators to investigate the influence of additional Langevin dynamic steps. We measure the test-time compute (including diffusion steps $N$ and Langevin steps $N_L$) in the number of function evaluations (NFEs), *i.e.* calls to the score model, recalling the basic relation NFE $= N \times (N_L + 1)$. We tested the posterior inference performance in the regime up to $10^3$ NFEs and presented Pareto-efficient assignments of test-time compute.

### C.4 STATISTICAL METHODOLOGY

### C.4.1 EVALUATION METRICS

**Root Mean Squared Error** (RMSE) measures the average magnitude of the errors between the predicted posterior samples and the ground-truth. A lower RMSE indicates a more concentrated prediction around the true parameters or data points. The RMSE is defined as

$$\text{RMSE} = \sqrt{\frac{1}{L\,N_{\text{post}}} \sum_{l=1}^{L} \sum_{j=1}^{N_{\text{post}}} \left(y_l - \widehat{y}_{l,j}\right)^2}, \tag{A53}$$

where $L$ is the feature dimension, $N_{\text{post}}$ is the number of posterior samples, $y_{i,l}$ is the ground-truth and $\widehat{y}_{l,j}$ is the prediction for the $l$-th feature (data point or parameter) and $j$-th posterior sample. In posterior inference experiments, we compute the RMSE between $N_{\text{post}} = 1,000$ samples and the ground-truth parameters. Conversely, in data prediction experiments, we compute the RMSE between $N_{\text{post}} = 100$ predicted data points and the ground-truth.

**Classifier Two-Sample Test** (C2ST; (Lopez-Paz & Oquab, 2017)) is a method to assess whether two sets of samples originate from the same distribution. In our context, it is used to compare the estimated posterior samples against samples from a reference posterior distribution. In our experiments, a random forest classifier is trained to distinguish between samples from the two distributions. An accuracy close to 0.5 suggests that the two distributions are indistinguishable.

**Mean Marginal Total Variation Distance** (MMTV) quantifies the dissimilarity between two multi-variate distributions by considering their marginal distributions, defined as

$$\text{MMTV}(p,q) = \sum_{d=1}^{D} \int_{-\infty}^{\infty} \frac{\left|p_d^{\text{M}}(x_d) - q_d^{\text{M}}(x_d)\right|}{2\,D} \, \mathrm{d}x_d, \tag{A54}$$

where $p_d^{\text{M}}$ and $q_d^{\text{M}}$ denote the marginal densities of $p$ and $q$ along the $d$-th dimension. An MMTV metric of 0.2 indicates that, on average across dimensions, the posterior marginals have an $80\%$ overlap, often indicated as a desirable threshold for an approximate posterior (Acerbi, 2018).

**Maximum Mean Discrepancy** (MMD) measures the distance between the mean embeddings of the distributions in a reproducing kernel Hilbert space. For our evaluations, we employ the MMD with with an exponentiated quadratic kernel with a lengthscale of 1.

### C.4.2 SIGNIFICANCE TESTING

To assess the statistical significance of the differences in performance between methods, we employ the Wilcoxon signed-rank test (Wilcoxon, 1945). In our experimental comparisons, we use this test to determine if the observed differences in metric scores between PriorGuide and baseline methods are statistically significant across multiple benchmark problems or datasets. The test evaluates the null hypothesis that the median difference between paired observations is zero, providing a p-value to indicate the significance of any observed deviation from this null hypothesis. We consider a result to be significantly different if the p-value is below 0.05.

## D ADDITIONAL EXPERIMENTAL RESULTS

We present here additional experimental results. First, in Appendix D.1, we provide a visual example of test-time adaptation to supplement the discussion in Section 4.1. In Appendix D.2, we demonstrate PriorGuide's applicability to structurally complex priors through a sequential Bayesian inference experiment on Two Moons. In Appendix D.3, we introduce additional baselines, including rejection sampling, sampling-importance-resampling and neural likelihood estimation (NLE; Papamakarios et al., 2019) with MCMC, to provide a broader comparison. We also present a study on the model's sensitivity to the distance between training and test-time priors (Appendix D.4) and conduct an ablation study on the GMM prior ratio approximation (Appendix D.5), which is further validated by a scalability analysis of the ratio fit in high-dimensional space (Appendix D.6). Furthermore, we include example posterior visualizations for the BCI model (Appendix D.7) and an analysis of training and test wall-clock time costs for Simformer and PriorGuide (Appendix D.8).

### D.1 ILLUSTRATION OF TEST-TIME PRIOR ADAPTATION ON TWO MOONS

This section provides the figure referenced in Section 4.1, illustrating the core capability of PriorGuide. We use the Two Moons model to provide an intuitive visualization of test-time prior adaptation. The base diffusion model is trained under a uniform prior $p_{\text{train}}(\boldsymbol{\theta})$ over $[-1, 1]^2$. At test time, we introduce a more localized Gaussian prior and employ PriorGuide to guide the diffusion sampling towards the correct posterior distribution (Fig. A1).

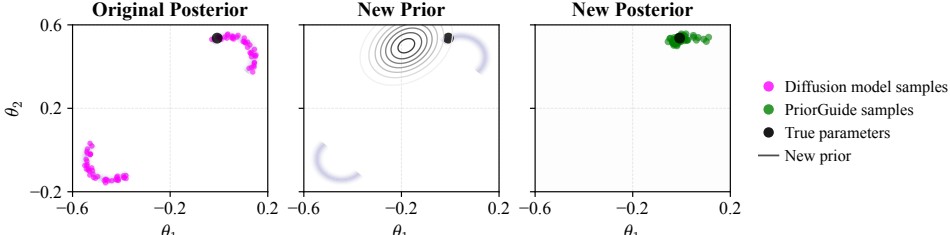

Figure A1: *Left:* Original Two Moons posterior and posterior samples from the base diffusion model trained on a uniform prior. *Middle:* New prior information about the parameters becomes available. *Right:* PriorGuide steers the diffusion process to match the Bayesian posterior under the new prior.

### D.2 SEQUENTIAL BAYESIAN UPDATING WITH INDEPENDENT OBSERVATIONS

To futher demonstrate PriorGuide's applicability to complicated and non-factorized priors, we examine a sequential Bayesian inference scenario. We utilize the Two Moons model under a uniform prior $p_{\text{train}}(\boldsymbol{\theta})$ over $[-1, 1]^2$, and consider a two-step Bayesian update scenario:

1. We simulate two observations $\mathbf{x}_1, \mathbf{x}_2$ given a parameter vector $\boldsymbol{\theta}^\star$.
2. Initial update: We calculate the intermediate posterior $p(\boldsymbol{\theta} \mid \mathbf{x}_1)$ given the first observation $\mathbf{x}_1$. In the Two Moons problem, this posterior is by construction a complex bimodal distribution (see Fig. A1 and Fig. A2). This becomes our ground-truth *target prior* $q(\boldsymbol{\theta})$.
   - For PriorGuide, we perform prior ratio fitting by fitting a Gaussian mixture to this intermediate posterior (the ratio is $\propto q(\boldsymbol{\theta})$ due to the uniform prior).

3. Sequential update: We employ the inference method of choice (PriorGuide or another baseline) to sample from the final posterior $p(\boldsymbol{\theta} \mid \mathbf{x}_1, \mathbf{x}_2)$, using $q(\boldsymbol{\theta})$ as the prior and conditioning on the second observation $\mathbf{x}_2$. Notably, the fact that $q(\boldsymbol{\theta}) = p(\boldsymbol{\theta}|\mathbf{x}_1)$ is itself a posterior does not require any special treatment: "yesterday's posterior is today's prior".

Using this setup, we evaluate PriorGuide and baselines across 10 true parameter points randomly sampled from $p_{\text{train}}$, simulating corresponding observation pairs for each. To establish ground truths for both the intermediate posteriors (and target priors) $p(\boldsymbol{\theta} \mid \mathbf{x}_1)$ as well as the joint posteriors $p(\boldsymbol{\theta} \mid \mathbf{x}_1, \mathbf{x}_2)$, we numerically calculate these posteriors on a dense discrete grid ($1000 \times 1000$ points) over the domain $[-1, 1]^2$.

On posterior quality metrics (C2ST, MMTV), PriorGuide strongly outperforms the baselines (Table A2). Here ACE underperforms, likely due to the fact that this complex prior differs significantly from the distribution of priors—the *meta-prior*—it has been pretrained on (see Appendix C.2), showing the importance of test-time prior adaptation. No difference is observed across methods for RMSE, but we recall that RMSE measures the error between the posterior and ground-truth parameter $\boldsymbol{\theta}^\star$. RMSE is not meaningful when the posterior is multimodal like in this case: the error is dominated by the regions of the posterior in the wrong mode, even though it is correct Bayesian behavior for the posterior to be multimodal. An illustration of such bimodality and of the update procedure for PriorGuide is visualized in Fig. A2 for one example pair $(\mathbf{x}_1, \mathbf{x}_2)$. Overall, these results confirm PriorGuide's capability to effectively handle complex, data-derived priors.

Table A2: Posterior inference ($p(\boldsymbol{\theta} \mid \mathbf{x}_1, \mathbf{x}_2)$). Mean $\pm$ standard dev. over 5 independent training runs (10 randomly generated true parameters and observation sets). Significantly best results (Wilcoxon signed-rank test) in bold.

| Dataset | Method | RMSE | C2ST | MMTV |
|---|---|---|---|---|
| Two Moons | Simformer | $0.63 \pm 0.34$ | $0.89 \pm 0.02$ | $0.66 \pm 0.07$ |
| | ACE | $0.63 \pm 0.37$ | $0.94 \pm 0.03$ | $0.71 \pm 0.07$ |
| | PriorGuide | $0.63 \pm 0.34$ | $\mathbf{0.60} \pm 0.07$ | $\mathbf{0.22} \pm 0.13$ |

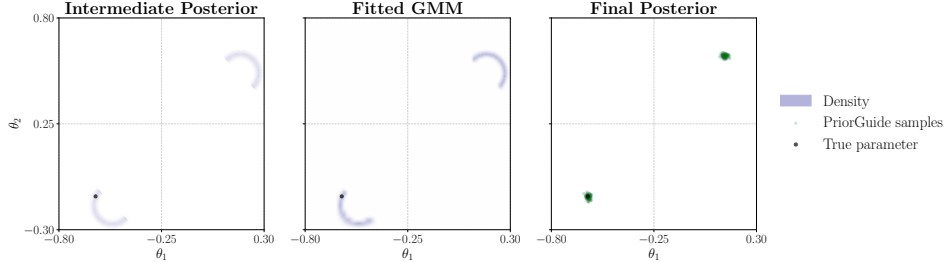

Figure A2: *Left:* The complex, bimodal posterior $p(\boldsymbol{\theta} \mid x_1)$ derived from the first observation $\mathbf{x}_1$. *Middle:* This posterior is fitted with a GMM to serve as the new target prior $q(\boldsymbol{\theta})$ for the next step. *Right:* PriorGuide samples from the final posterior $p(\boldsymbol{\theta}|\mathbf{x}_1, \mathbf{x}_2)$ by guiding the diffusion model with the new prior $q(\boldsymbol{\theta})$ and conditioning on the second observation $\mathbf{x}_2$. PriorGuide's samples match well the ground-truth posterior density.

## D.3 ADDITIONAL BASELINES

### D.3.1 REJECTION SAMPLING, AND SAMPLING IMPORTANT RESAMPLING

In addition to the main results, we compare PriorGuide against two straightforward sampling-based baselines: Sampling-Importance-Resampling (SIR) and Rejection Sampling (RS). Both methods are widely used and theoretically consistent. However, their efficiency deteriorates rapidly when the proposal distribution—in our case, the posterior from the base model under the training prior—is a poor match for the target posterior under the new test-time prior, particularly under high-dimensional or complex posterior geometries. Our experiments quantify this inefficiency and provide a direct

performance comparison under matched computational budgets, highlighting the practical advantages of PriorGuide.

**Setup.** For SIR, we drew 20,000 samples from the base diffusion model, applied importance weights, and evaluated performance alongside two diagnostics: effective sample size (ESS) and the $\hat{k}$ instability measure (Vehtari et al., 2024), where $\hat{k} > 0.7$ indicates instability. For RS, we continued sampling until either 1,000 accepted samples (matching PriorGuide) were obtained or a two-hour time limit on an Nvidia A100 GPU was reached. All three methods (PriorGuide, RS, and SIR) rely on samples from the same Simformer model, with PriorGuide requiring additional per-sample computation. This allows us to align costs directly. As shown in Table A8, generating a single PriorGuide sample is roughly as expensive as producing 20 base Simformer samples. For our comparison we use 1,000 PriorGuide samples, 20,000 SIR samples, and cap RS at two hours to equalize compute budgets. This cap is generous as RS acceptance rates are very low, while PriorGuide sampling itself takes about $\sim$150 seconds (see Table A8). Results are summarized below.

Table A3: Posterior inference ($\boldsymbol{\theta}$) results on Turin with RS and SIR, under *strong* (S) and *mixture* (M) test-time priors.

| Dataset | Method | RMSE | C2ST | MMTV | Acc Rate (%) | ESS (%) | $\hat{k}$ |
|---------|--------|------|------|------|--------------|---------|-----------|
| Turin (S) | Simformer | $0.23 \pm 0.04$ | $0.95 \pm 0.02$ | $0.55 \pm 0.07$ | — | — | — |
| | ACE | $0.18 \pm 0.02$ | $0.92 \pm 0.03$ | $0.47 \pm 0.08$ | — | — | — |
| | PriorGuide | $0.06 \pm 0.01$ | $0.55 \pm 0.03$ | $0.08 \pm 0.02$ | — | — | — |
| | Diffusion + SIR | $0.06 \pm 0.01$ | $0.81 \pm 0.07$ | $0.18 \pm 0.24$ | | $0.02 \pm 0.01$ | $0.35 \pm 0.42$ |
| | Diffusion + RS | $0.06 \pm 0.01$ | $0.56 \pm 0.04$ | $0.08 \pm 0.02$ | $0.63 \pm 0.33$ | — | — |
| Turin (M) | Simformer | $0.24 \pm 0.04$ | $0.95 \pm 0.02$ | $0.52 \pm 0.07$ | — | — | — |
| | ACE | $0.20 \pm 0.03$ | $0.91 \pm 0.03$ | $0.45 \pm 0.07$ | — | — | — |
| | PriorGuide | $0.14 \pm 0.05$ | $0.64 \pm 0.09$ | $0.21 \pm 0.12$ | — | — | — |
| | Diffusion + SIR | $0.09 \pm 0.03$ | $0.80 \pm 0.07$ | $0.19 \pm 0.22$ | | $0.02 \pm 0.01$ | $0.39 \pm 0.38$ |
| | Diffusion + RS | $0.09 \pm 0.02$ | $0.55 \pm 0.03$ | $0.08 \pm 0.04$ | $0.49 \pm 0.35$ | — | — |

Table A4: Posterior inference ($\boldsymbol{\theta}$) results on BCI with RS and SIR, under *strong* (S) and *mixture* (M) test-time priors.

| Dataset | Method | RMSE | C2ST | MMTV | Acc Rate (%) | ESS (%) | $\hat{k}$ |
|---------|--------|------|------|------|--------------|---------|-----------|
| BCI (S) | Simformer | $0.89 \pm 0.09$ | $0.97 \pm 0.02$ | $0.66 \pm 0.09$ | — | — | — |
| | ACE | $0.34 \pm 0.15$ | $0.91 \pm 0.05$ | $0.40 \pm 0.17$ | — | — | — |
| | PriorGuide | $0.24 \pm 0.04$ | $0.81 \pm 0.11$ | $0.33 \pm 0.19$ | — | — | — |
| | Diffusion + SIR | $0.22 \pm 0.03$ | $0.95 \pm 0.04$ | $0.63 \pm 0.32$ | | $0.01 \pm 0.01$ | $1.28 \pm 0.98$ |
| | Diffusion + RS | *Fail* | *Fail* | *Fail* | $\sim 0$ | — | — |
| BCI (M) | Simformer | $1.07 \pm 0.18$ | $0.99 \pm 0.01$ | $0.63 \pm 0.10$ | — | — | — |
| | ACE | $1.00 \pm 0.31$ | $0.97 \pm 0.02$ | $0.58 \pm 0.10$ | — | — | — |
| | PriorGuide | $0.79 \pm 0.70$ | $0.78 \pm 0.11$ | $0.35 \pm 0.24$ | — | — | — |
| | Diffusion + SIR | $0.25 \pm 0.03$ | $0.95 \pm 0.04$ | $0.79 \pm 0.31$ | | $0.00 \pm 0.00$ | $2.61 \pm 1.78$ |
| | Diffusion + RS | *Fail* | *Fail* | *Fail* | $\sim 0$ | — | — |

Table A5: Posterior inference ($\boldsymbol{\theta}$) results on Gaussian Linear 20D with RS and SIR, under *strong* (S) and *mixture* (M) test-time priors.

| Dataset | Method | RMSE | C2ST | MMTV | Acc Rate (%) | ESS (%) | $\hat{k}$ |
|---------|--------|------|------|------|--------------|---------|-----------|
| Gaussian Linear 20D (S) | Simformer | $0.30 \pm 0.02$ | $1.00 \pm 0.00$ | $0.64 \pm 0.02$ | — | — | — |
| | ACE | $0.10 \pm 0.01$ | $0.80 \pm 0.04$ | $0.15 \pm 0.02$ | — | — | — |
| | PriorGuide | $0.21 \pm 0.08$ | $0.58 \pm 0.03$ | $0.10 \pm 0.02$ | — | — | — |
| | Diffusion + SIR | $0.15 \pm 0.02$ | $\sim 1$ | $\sim 1$ | | $0.00 \pm 0.00$ | $10.85 \pm 1.32$ |
| | Diffusion + RS | *Fail* | *Fail* | *Fail* | $\sim 0$ | — | — |
| Gaussian Linear 20D (M) | Simformer | $0.30 \pm 0.01$ | $1.00 \pm 0.00$ | $0.64 \pm 0.02$ | — | — | — |
| | ACE | $0.24 \pm 0.03$ | $1.00 \pm 0.00$ | $0.52 \pm 0.07$ | — | — | — |
| | PriorGuide | $0.24 \pm 0.08$ | $0.59 \pm 0.06$ | $0.14 \pm 0.09$ | — | — | — |
| | Diffusion + SIR | $0.15 \pm 0.01$ | $\sim 1$ | $\sim 1$ | | $0.00 \pm 0.00$ | $10.61 \pm 1.12$ |
| | Diffusion + RS | *Fail* | *Fail* | *Fail* | $\sim 0$ | — | — |

**Results.** Under equal computational budgets, the results exhibit a dimensionality-driven behavior consistent with performance expectations of Monte Carlo methods in moderate dimensions (Robert & Casella, 2004). For the 4-dimensional Turin example (Table A3), the simple baselines remain viable, with performance roughly comparable to PriorGuide (sometimes slightly better, sometimes worse). RS achieves acceptable metrics but with highly variable acceptance rates, yielding only 98–126 accepted samples out of 20k.

Conversely, in the 5-dimensional BCI problem (Table A4), these baselines catastrophically fail. Rejection sampling produces zero accepted samples on most runs, while SIR's $\hat{k} > 0.7$ indicates statistical instability that makes its estimates unreliable. This transition due to dimensionality (4D vs. 5D) and the increased posterior complexity reveal a threshold where simple methods go from mildly unreliable to completely unusable, while PriorGuide maintains consistent performance across both problems.

Extending the analysis further to the 20D Gaussian Linear problem confirms this trend (Table A5). Despite being given a generous budget of 20k proposals (whereas each PriorGuide sample costs only a fraction of a Simformer sample), SIR and RS fail completely, while PriorGuide continues to deliver stable performance.

**Larger computational budget.** To examine behavior with substantially more compute, we ran rejection sampling until reaching 1,000 accepted samples or an 8-hour limit on a single Nvidia A100 GPU (up to ∼3.1M proposals) on Turin and BCI examples. While this setting is not typical for practitioners, it allows us to probe performance under very large budgets. In this regime, rejection sampling could sometimes match or slightly outperform PriorGuide. On Turin with a strong or mixture prior, both methods achieved similar accuracy (RMSE ≈0.06–0.09, C2ST ≈0.54–0.62, MMTV ≈0.07–0.19) with acceptance rates of 0.38–0.57%. On BCI, rejection sampling occasionally improved RMSE and MMTV (e.g., 0.22 vs. 0.63 RMSE on BCI-M), but acceptance was extremely low (0.06–0.16%), many runs hit the timeout, and some produced only a handful of accepted samples.

These results illustrate a trade-off: simple methods can approach strong performance if one is willing to spend hours of compute, but they remain unreliable and inefficient. PriorGuide achieves comparable quality in minutes (∼160 s for 1k samples), making it suitable for interactive use, repeated posterior evaluations, and sensitivity analysis, rather than only in very large-budget scenarios.

### D.3.2 NEURAL LIKELIHOOD ESTIMATION WITH MCMC

As an additional natural baseline, we evaluate a method based on neural likelihood estimation (NLE; Papamakarios et al., 2019) followed by MCMC sampling. This two-stage method is a representative of several likelihood-based approaches in SBI (Lueckmann et al., 2021), which trains a neural density estimator (e.g. a normalizing flow) to form a surrogate of the likelihood function $p(\mathbf{x} \,|\, \boldsymbol{\theta})$ (Papamakarios et al., 2019). Once the surrogate is trained, it is then used with a standard MCMC algorithm to draw samples from the posterior distribution $p(\boldsymbol{\theta} \,|\, \mathbf{x}_o)$ for a given observation $\mathbf{x}_o$.

While decoupling likelihood learning from posterior sampling is conceptually natural in SBI, we find this approach may lack robustness even in low-dimensional cases, particularly for problems with multi-modal posterior geometries. For instance, when this approach is applied in the bimodal Two Moons simulator Appendix C.1, the MCMC sampler guided by the learned likelihood often becomes trapped in a single posterior mode and fails to discover the other (Fig. A3). This failure is a known MCMC limitation that the sampler can easily get stuck in modes.

### D.4 SENSITIVITY TO THE DISTANCE BETWEEN TRAINING AND TEST-TIME PRIORS

To investigate the sensitivity of PriorGuide to the distance between the training and test-time priors, we conduct an experiment using the Gaussian Linear 10D model. We use the same training prior as in Section 4: a 10-dimensional Gaussian distribution $p_{\text{train}}(\boldsymbol{\theta}) = \mathcal{N}(\mathbf{0}, 0.1 \cdot \mathbf{I}_{10})$. For the test-time priors, we use strongly informative standard deviations ($\sigma_i^{\text{strong}} = 0.2 \cdot s_i$) and systematically shift their means ($\boldsymbol{\mu}^q$) away from the center of the training prior ($\boldsymbol{\mu}^p$). We quantify this shift using several metrics, including the Mahalanobis distance ($d'$), the 2-Wasserstein distance, and our proposed out-of-distribution (OOD) diagnostic (Appendix A.4).

The results, presented in Table A6, reveal how PriorGuide's performance responds to this increasing prior shift. The key take-home message is that PriorGuide is robust and degrades gracefully rather than failing catastrophically. For small to moderate shifts (e.g., $d' \leq 2$), performance remains nearly identical to the no-shift scenario, with all metrics (C2ST, MMTV, RMSE) showing negligible changes. As the prior is shifted into a substantially different region, we observe a smooth and predictable decline in accuracy. For instance, at an extreme shift where $d' = 4.4$, PriorGuide achieves an MMTV of 0.16, indicating that the generated posterior marginals have an 84% overlap with the ground truth on average across all dimensions, which corresponds to a reasonable posterior approximation (Acerbi,

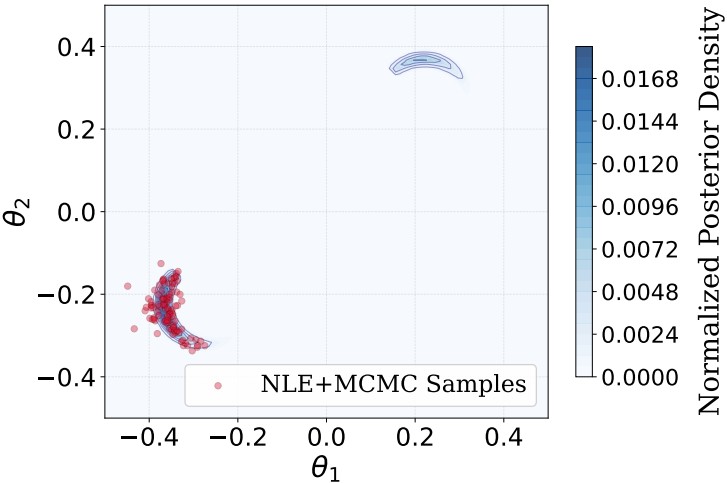

Figure A3: Failure case of the NLE with MCMC baseline on the Two Moons problem. The posterior is approximated by running an MCMC sampler on a learned neural likelihood.

2018). This analysis confirms that PriorGuide provides provides meaningful approximations even when the test-time prior is far from what was seen during training.

Table A6: Sensitivity analysis of posterior inference to prior shift. The degree of shift is quantified using both $d'$ (Mahalanobis distance) and 2-Wasserstein distance. The results indicate that PriorGuide maintains competitive performance even under substantial prior shifts.

| $\|\boldsymbol{\mu}^q - \boldsymbol{\mu}^p\|$ | $d'$ | 2-Wasserstein | OOD Frac. | C2ST | MMTV | RMSE |
|---|---|---|---|---|---|---|
| 0.00 | 0.00 | 0.84 | 0.00 | $0.52 \pm 0.05$ | $0.05 \pm 0.01$ | $0.21 \pm 0.10$ |
| 0.32 | 0.44 | 0.89 | 0.00 | $0.53 \pm 0.06$ | $0.06 \pm 0.02$ | $0.20 \pm 0.08$ |
| 0.63 | 0.88 | 1.05 | 0.00 | $0.54 \pm 0.07$ | $0.06 \pm 0.02$ | $0.23 \pm 0.09$ |
| 0.95 | 1.32 | 1.27 | 0.00 | $0.56 \pm 0.08$ | $0.07 \pm 0.03$ | $0.32 \pm 0.13$ |
| 1.26 | 1.76 | 1.52 | 0.00 | $0.59 \pm 0.09$ | $0.08 \pm 0.04$ | $0.41 \pm 0.15$ |
| 1.58 | 2.20 | 1.79 | 0.02 | $0.61 \pm 0.09$ | $0.10 \pm 0.04$ | $0.58 \pm 0.14$ |
| 1.90 | 2.64 | 2.07 | 1.00 | $0.62 \pm 0.09$ | $0.10 \pm 0.04$ | $0.73 \pm 0.16$ |
| 2.21 | 3.08 | 2.37 | 1.00 | $0.64 \pm 0.08$ | $0.11 \pm 0.04$ | $0.84 \pm 0.18$ |
| 2.53 | 3.52 | 2.66 | 1.00 | $0.65 \pm 0.08$ | $0.12 \pm 0.05$ | $1.04 \pm 0.19$ |
| 2.85 | 3.96 | 2.97 | 1.00 | $0.67 \pm 0.08$ | $0.14 \pm 0.05$ | $1.25 \pm 0.19$ |
| 3.16 | 4.40 | 3.27 | 1.00 | $0.70 \pm 0.08$ | $0.16 \pm 0.06$ | $1.42 \pm 0.24$ |

### D.5 IMPACT OF GMM COMPONENT COUNT ON PRIOR RATIO APPROXIMATION

We conduct an additional ablation study on the influence of the number of GMM components used to approximate the prior ratio. To isolate this effect, this experiment uses a controlled setup that differs from the main experiments: we evaluate performance using one target prior and one trained diffusion model, with metrics averaged across ten observations generated from that prior. As shown in Table A7, sampling speed remains efficient even with very large GMMs (e.g. 200 components), indicating that it is not a bottleneck at this stage. In contrast, the performance of PriorGuide is sensitive to the quality of the prior ratio approximation when using very few components. We therefore recommend that users fit a flexible and highly expressive GMM tailored to their specific application to ensure accurate and stable guidance. In practice, we see that 20 components are enough in our experiments.

D.6  SCALABILITY AND ROBUSTNESS OF THE PRIOR RATIO APPROXIMATION

To assess the scalability of our GMM fitting procedure for the prior ratio $r(\boldsymbol{\theta})$, we evaluate the quality of the approximation in a 10-dimensional setting. We use the Gaussian Linear 10$D$ setup and set $q(\boldsymbol{\theta})$ as a Gaussian mixture.

We execute the fitting procedure for 100 randomly sampled mixture target priors (using the procedures detailed in Appendix C.3.1). For each fit, we estimate the weighted $L_2$ distance in log space, that is the $L_2$ distance between the GMM approximation log predictive density (including the fitted log normalization constant $\ln Z$) and the true analytic log ratio using 10,000 samples from the fitted GMM. The method achieves a median $L_2$ error of 0.023 (2.5th-25th-75th-97.5th percentiles: 0.002-0.016-0.032-0.186), confirming that standard gradient-based optimization consistently converges to an accurate approximation of the ratio function in this 10$D$ scenario.[7] In practice, to address the small percentage of fits that converge to suboptimal solutions, we recommend running multiple optimizations with different initializations—which is the standard approach when solving a global nonconvex optimization problem.

To give an example of such fits, Fig. A4 displays a fitted GMM ratio against the true analytic ratio in log-space. We evaluate both the true analytic ratio and the fitted GMM passing through a fixed parameter vector $\theta^*$, illustrating the function's behavior for each pair of dimensions while holding the remaining dimensions constant. The figure demonstrates that the fitted GMM (orange lines) faithfully recovers the geometry of the true ratio (dashed black lines) across these conditional slices.

Table A7: Ablation study illustrating the influence of the number of GMM components used to approximate the prior ratio.

| Dataset | #GMM components | Time cost (s) per sample | C2ST | MMTV | RMSE |
|---|---|---|---|---|---|
| Gaussian Linear 10D | 2 | $0.02 \pm 0.00$ | $0.97 \pm 0.01$ | $0.31 \pm 0.05$ | $0.20 \pm 0.03$ |
| | 20 | $0.02 \pm 0.00$ | $0.55 \pm 0.04$ | $0.09 \pm 0.04$ | $0.23 \pm 0.07$ |
| | 200 | $0.02 \pm 0.00$ | $0.56 \pm 0.05$ | $0.11 \pm 0.04$ | $0.28 \pm 0.11$ |
| BCI | 2 | $0.18 \pm 0.01$ | $1.00 \pm 0.00$ | $0.72 \pm 0.04$ | $1.71 \pm 0.21$ |
| | 20 | $0.18 \pm 0.01$ | $0.88 \pm 0.10$ | $0.57 \pm 0.24$ | $1.36 \pm 0.66$ |
| | 200 | $0.18 \pm 0.01$ | $0.88 \pm 0.10$ | $0.57 \pm 0.24$ | $1.36 \pm 0.65$ |

D.7  BAYESIAN CAUSAL INFERENCE (BCI) MODEL POSTERIOR SAMPLES VISUALIZATION

In Fig. A5, we visualize the ground-truth posterior distribution alongside samples generated by Simformer (no prior adaptation) and with PriorGuide applied to the same base Simformer model. We use the `corner` package (Foreman-Mackey, 2016), which displays pairwise joint distributions and marginal histograms of the parameters.

D.8  TRAINING VS. TEST-TIME COST ANALYSIS

We summarize in Table A8 the estimated wall-clock time required by PriorGuide to produce 1,000 posterior samples under each simulator, measured on a system equipped with an Nvidia Ampere A100 GPU. For comparison, we also report the cost of retraining a diffusion inference model (Simformer, (Gloeckler et al., 2024)) with 10,000 simulations, which includes both the neural network training and the simulator call required to create its training set. As Table A8 demonstrates, PriorGuide achieves more than a tenfold speedup across most simulators, with the greatest gains observed in scenarios (*e.g.*, Turin) where the simulator calls are most computationally expensive.

E  COMPUTATIONAL RESOURCES AND SOFTWARE

**Computational resources**  Most experiments presented in this work are performed on a cluster equipped with AMD MI250X GPUs, while some additional experiments in the appendix are per-

---

[7]Note that 0.023 error in log space corresponds to $\sim 2\%$ error on the ratio, since $e^x \approx 1 + x$ for small $x$.

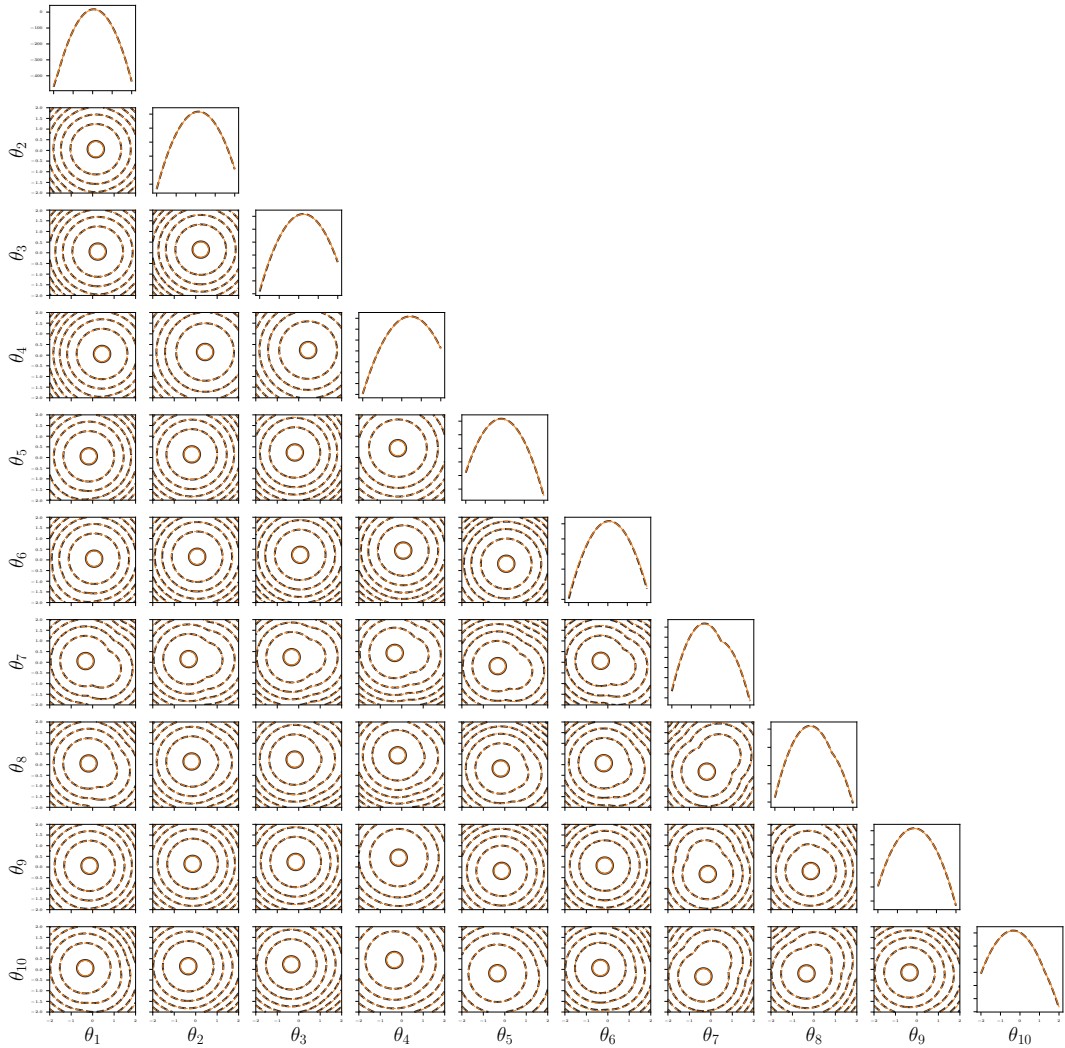

Figure A4: Example visualization of the prior ratio approximation in 10D (Gaussian Linear problem). Conditional 2D cross-sections of the true prior ratio $r(\boldsymbol{\theta})$ (dashed **black** lines) versus the fitted GMM approximation (orange lines). Note that the target ratio is not Gaussian, as visible in some panels. The GMM approximation achieves a near-perfect fit.

Table A8: A comparison of time costs (seconds) between performing PriorGuide posterior inference and retraining Simformer, including neural network training and generating new training data.

| Simulator | PriorGuide testing | Simformer retraining and testing | | | |
|---|---|---|---|---|---|
| | | Total | Simulation | Training | Sampling |
| Two Moons | 9.7 | 314.4 | 13.5 | 295.2 | 5.7 |
| OUP | 25.1 | 305.7 | 9.2 | 289.4 | 7.1 |
| Turin | 159.9 | 9346.6 | 8300.4 | 1037.4 | 8.8 |
| Gaussian Linear | 16.8 | 266.1 | 3.8 | 255.2 | 7.1 |
| BCI | 158.8 | 1063.1 | 49.1 | 1005.7 | 8.3 |

formed on a cluster equipped with Nvidia Ampere A100 GPUs. The total computational resources consumed for this research, including all development stages and experimental runs, are estimated to be approximately 30,000 GPU hours.

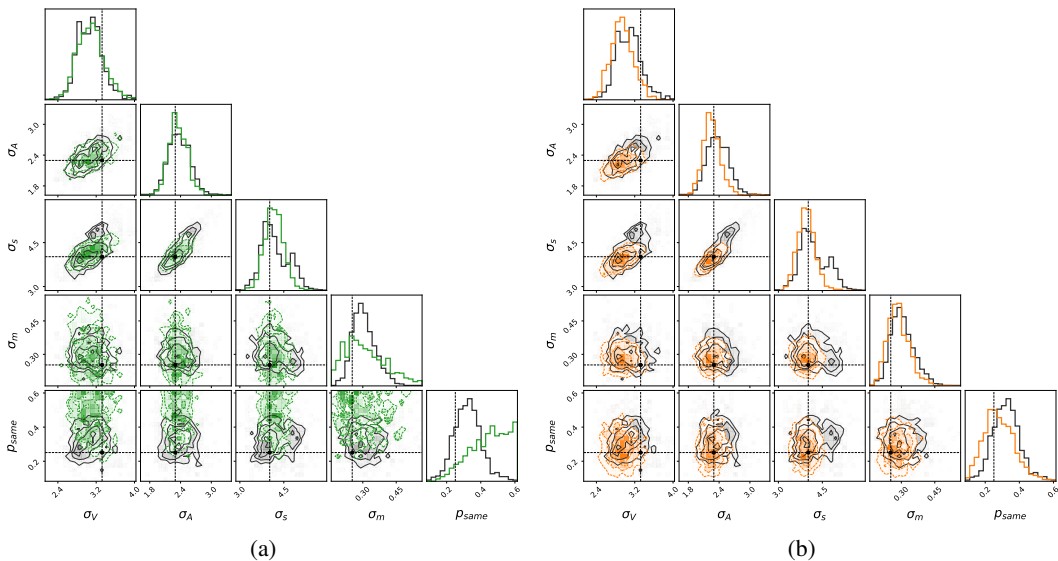

Figure A5: Posterior samples for the Bayesian causal inference (BCI) model, for a randomly chosen dataset. Ground-truth samples are shown in **black**, and the dashed black lines and black dot indicate the true parameter values. **(a)** Original Simformer samples (**green**) without prior adaptation, and **(b)** PriorGuide samples (**orange**) under a *mild* prior. Incorporating prior information with PriorGuide steers the inference toward more plausible parameter regions, especially for parameters such as $\sigma_m$ and $p_{\text{same}}$ which are less constrained by the data alone, resulting in posterior samples that more closely match the ground-truth posterior under the same prior, as shown in (b).

**Software** The core code base is built upon the `Simformer` repository (https://github.com/mackelab/simformer, License: MIT), using `JAX` (https://docs.jax.dev/en/latest/, License: Apache-2.0) and `PyTorch` (https://pytorch.org/, License: modified BSD license). The implementations of the Two Moons and Gaussian Linear simulators are based on `sbibm` (https://github.com/sbi-benchmark/sbibm, License: MIT). For the ground-truth posterior generation, we utilize `sbi` (https://sbi-dev.github.io/sbi/latest/, License: Apache-2.0) and `PyVBMC` (https://acerbilab.github.io/pyvbmc/, License: BSD 3-Clause). Our implementation of the ACE baseline uses the repository provided by (Chang et al., 2025) (https://github.com/acerbilab/amortized-conditioning-engine, License: Apache-2.0).

# F USE OF LARGE LANGUAGE MODELS

We acknowledge the use of Large Language Models (LLMs) to support various stages of the research. In the initial phase, LLMs were utilized for inspirations, helping to explore methodological approaches and find existing works. In the development phase, LLMs served as programming assistants to aid tasks such as implementing algorithms and debugging. LLMs also supported the writing process, providing assistance with polishing the manuscript for clarity, conciseness, and grammatical correctness.

