# OpenReview forum: "PriorGuide: Test-Time Prior Adaptation for Simulation-Based Inference"
_ICLR.cc/2026/Conference — ICLR 2026 Poster_

### Official Review · Reviewer_zfhL · 2025-10-29

**Soundness:** 3
**Presentation:** 4
**Contribution:** 3
**Rating:** 6
**Confidence:** 4

**Summary:**

The paper proposes PriorGuide, a method to adapt diffusion-based amortized
simulation-based inference (SBI) models to new priors at test time without
retraining. The key idea is to express the target-posterior score as the original score
(under the training prior) plus a guidance term involving the prior ratio
$(r(\omega)=q(\omega)/p_{\text{train}}(\omega))$. To make the guidance tractable, the
reverse transition kernel is approximated as Gaussian and the prior ratio is
approximated by a Gaussian Mixture Model (GMM), which yields a closed-form guidance
update. The method is evaluated on a suite of SBI problems for both posterior and
posterior-predictive inference. Empirically, PriorGuide attains competitive performance
relative to baselines, and the paper further studies a compute–accuracy trade-off
using interleaved Langevin refinement.

**Strengths:**

**Quality.** The paper’s theoretical foundation is solid: the score decomposition with a
guidance term is standard and correctly instantiated for the SBI setting. The Gaussian
reverse-kernel and GMM prior-ratio approximations are reasonable and enable a practical
algorithm. The analysis of test-time compute via Langevin refinement is appropriate, and
the extended experiments cover canonical SBI tasks. The ablation and sensitivity studies
in the appendix further strengthen the empirical grounding.

**Clarity.** The manuscript is exceptionally well written, with well-structured
exposition and clear intuition for the guidance term. Notation is introduced early and
consistently; figures and tables are easy to interpret. The appendix integrates detailed
methodological clarifications—GMM fitting, diagnostics, and runtime analyses—that
improve reproducibility. The discussion of the Pareto front between diffusion and
Langevin steps is particularly insightful.

**Significance and Contribution.** The work addresses a practically important
problem—changing priors after training—where retraining amortized SBI models can be
prohibitively expensive. Adapting diffusion guidance to encode new priors at inference
is an appealing and useful contribution. The empirical scope is strong, especially given
the additional baselines in the appendix (Rejection Sampling, Sequential Importance
Resampling, NLE+MCMC), a sensitivity analysis, and a 20D test demonstrating scalability.
The contribution remains primarily empirical but offers meaningful methodological value
for the SBI community.

**Originality.** Applying diffusion guidance to test-time prior adaptation in amortized
SBI is a natural but novel design choice. The GMM ratio approximation provides a
practical mechanism to operationalize this guidance, and the extended analyses confirm
that the method maintains competitive performance where simpler baselines fail.

**Weaknesses:**

**Scope and positioning.** While the paper clearly demonstrates PriorGuide’s strengths,
it could offer more explicit practical guidance for practitioners—specifically, when
PriorGuide should be preferred over simpler baselines such as NLE + MCMC, RS, SIR, or
related methods like SIMFORMER and ACE and how to balance its additional test-time
computation against retraining or posterior-correction methods. While this is discussed
qualitatively in the appendix, a concise summary in the main text would make the
contribution more actionable.

**Dimensionality constraints.** Although the new 20D Gaussian Linear results demonstrate
scalability, these remain relatively simple benchmarks. A more complex or real-world
high-dimensional example would strengthen the claim of general applicability.

**Approximation limits.** The Gaussian reverse-kernel approximation, while adequate for
the reported tasks, remains a simplifying assumption. Even with Langevin refinement
mitigating some of its effects, further diagnostic guidance for practitioners would be
useful.

**Questions:**

1. Could the authors provide clearer practical guidance on when PriorGuide is
   preferable to simpler or amortized alternatives? For instance, how should
   practitioners trade off its added test-time computation against retraining costs or
   simpler posterior-correction methods?

**Details Of Ethics Concerns:**

The work uses synthetic data and provides extensive
methodological details and open-source code. The potential for ethical risk is minimal.
The expanded appendices substantively improve reproducibility and transparency.

---

> ### Author Response · Authors · 2025-11-25
> **Response to Reviewer zfhL**
>
> Thank you for your detailed and thoughtful assessment. We value your suggestion to make the contribution more actionable for practitioners. We are also grateful for your recognition of our work’s rigorous foundation and significance.
>
> > **W1** Scope and positioning. While the paper clearly demonstrates PriorGuide's strengths, it could offer more explicit practical guidance for practitioners—specifically, when PriorGuide should be preferred over simpler baselines such as NLE + MCMC, RS, SIR, or related methods like SIMFORMER and ACE and how to balance its additional test-time computation against retraining or posterior-correction methods. While this is discussed qualitatively in the appendix, a concise summary in the main text would make the contribution more actionable.
>
> > **Q1** Could the authors provide clearer practical guidance on when PriorGuide is preferable to simpler or amortized alternatives? For instance, how should practitioners trade off its added test-time computation against retraining costs or simpler posterior-correction methods?
>
> We completely agree with this. In the revision, we have extracted the key insights of our baseline comparisons and included them in the Discussion section in the main text. Specifically, our practical takeaway to use PriorGuide when the dimensionality is moderate-to-high ($4 < D \le 20$, simple baselines like Rejection Sampling and SIR degrade or fail), simulators are expensive (retraining an amortized model is computationally prohibitive), or priors are complex/correlated (methods based on pre-defined meta-priors such as ACE cannot capture the target distributions). We also noted that in terms of *pure speed*, a fully amortized method is naturally faster.
>
> > **W2** Dimensionality constraints. Although the new 20D Gaussian Linear results demonstrate scalability, these remain relatively simple benchmarks. A more complex or real-world high-dimensional example would strengthen the claim of general applicability.
>
> Thanks for the suggestion. We agree that exploring higher-dimensional settings is a valuable direction. We also would like to note that standard SBI applications [1, 2, 3] typically only involve moderate-to-low parameter dimensions (rarely above $D \approx 10$).
>
> > **W3** Approximation limits. The Gaussian reverse-kernel approximation, while adequate for the reported tasks, remains a simplifying assumption. Even with Langevin refinement mitigating some of its effects, further diagnostic guidance for practitioners would be useful.
>
> While indeed a simplifying assumption, we would like to highlight that the Gaussian assumption has been reported to perform adequately even in some very high-dimensional tasks, such as image restoration [4,5], and, as shown in the paper, becomes accurate at low noise levels. As such, correctness is theoretically a question of how much diffusion and Langevin steps we use in practice. This is, however, also the case for standard diffusion model inference: More neural function evaluations in the reverse ODE/SDE increase accuracy, and Langevin-like stochastic sampling can alleviate errors caused by an inaccurately learned score function [6]. As such, a simple approach to calibrate the sampling hyperparameters is simply to increase the number of (neural) function evaluations (NFEs) until the output distribution does not change. We have now included a comment about this in Section 4.4.
>
> **References:**
>
> [1] Dax et al. *Real-time inference for binary neutron star mergers using machine learning*. Nature, 2025.
>
> [2] Hahn et al. *Cosmological constraints from non-Gaussian and nonlinear galaxy clustering using the SimBIG inference framework*. Nature Astronomy, 2024.
>
> [3] Lueckmann et al. *Benchmarking Simulation-Based Inference*. AISTATS 2021.
>
> [4] Song et al. *Pseudoinverse-guided diffusion models for inverse problems*. ICLR 2023.
>
> [5] Song et al. *Loss-guided diffusion models for plug-and-play controllable generation*. ICML 2023.
>
> [6] Karras et al. *Elucidating the design space of diffusion-based generative models*. NeurIPS 2022.

---

> > ### Comment · Reviewer_zfhL · 2025-11-26
> > **Response to rebuttal**
> >
> > I thank the authors for their concise response to my points.
> >
> > Having read the other reviews and the overall revisions of the paper, I agree that this is valuable contribution to the conference and I will adapt my score accordingly.

---

### Official Review · Reviewer_kkwQ · 2025-10-30

**Soundness:** 4
**Presentation:** 3
**Contribution:** 4
**Rating:** 10
**Confidence:** 5

**Summary:**

PriorGuide introduces an extension to framework of simulation-based inference (SBI) with diffusion models that allows pre-trained diffusion models to incorporate new prior distributions at inference time without requiring retraining. This is achieved using a novel Gaussian mixture model approximation to make the target prior tractable for guiding the diffusion sampling process. Empirical results on synthethic data demonstrate PriorGuide’s effectiveness in accurately recovering posterior and posterior-predictive distributions across various SBI problems. The method also allows for refinement through Langevin dynamics, enabling a balance between computational cost and accuracy.

By enabling adaptation to new priors without retraining, PriorGuide exemplifies a “test-time compute” approach, extending the capabilities of pre-trained models with targeted computations. This decoupling of simulator runs from prior specification offers practical benefits especially in the case where simulations are costly, including the ability to perform post-hoc prior sensitivity analyses and incorporate domain expert knowledge after training, ultimately reducing the computational burden of scientific workflows which employ SBI.

**Strengths:**

The manuscript proposes an extension of SimFormer modeling to the challenging problem of representing complex joint probability distributions. The core idea is relatively straightforward, yet the authors provide a rigorous mathematical treatment that appropriately identifies both the potential benefits and inherent limitations of this approach, acknowledging the necessary reliance on approximations. The empirical evaluation is a strength of the paper; the results presented demonstrate promising performance, particularly in the accuracy of the reported uncertainty estimates (Tables 1 & 2). The selection of benchmarking use cases is clearly motivated and relevant to the stated goals. Finally, the authors effectively utilize concise and informative visualizations to support their claims and illustrate the underlying hypothesis.

**Weaknesses:**

The manuscript would benefit from a more careful review of its language, as instances of anthropomorphism (e.g., “the model having seen few training examples”) can detract from the scientific rigor. Additionally, the presentation of the derivation for 'r' is hampered by misaligned page breaks, separating key explanatory text and hindering comprehension. More significantly, while the paper thoroughly explores the technical capabilities of PriorGuide, it lacks a discussion of the broader implications of manipulating the prior distribution within a Bayesian framework. The authors appear to assume a familiarity with this interplay between prior, simulator, and the scientific endeavor, which may limit the accessibility and impact of the work for a wider audience.

**Questions:**

- page 4, the derivation of eq 6-9 is central to the paper, please try to rearrange so that the text in lines 216-220 is readily visible close the equations (e.g. by removing lines 191-195)
- the code for the experiments was not found in the paper, i.e. through an anonymised git repo, this should be corrected for publication
- line 232: please do not use anthropomorphic language to describe model behavior "the model having seen few training examples", i.e. the model has consumed or been trained on
- line 234: "using standard OOD metrics" as such a standard exists only colloquially, please remove "standard"
- equ 13: the text lacks a discussion if `K` is a free parameter and how to choose it
- line 274: "Notably, when p(θ) is uniform, r(θ) reduce*s* to q(θ), which can be directly specified as a Gaussian mixture." perhaps also add hint why r reduces to q (due to using the expectation I guess)
- line 310: "To incorporate this with regular diffusion ..." perhaps comment on the computational cost here already.

---

> ### Author Response · Authors · 2025-11-25
> **Response to Reviewer kkwQ**
>
> Thank you for your strong endorsement and for highlighting the effectiveness of our empirical evaluation. We have taken your feedback on the manuscript's presentation and scientific precision, and have carefully revised the manuscript to address them.
>
> > **Weaknesses** The manuscript would benefit from a more careful review of its language, as instances of anthropomorphism (e.g., "the model having seen few training examples") can detract from the scientific rigor. Additionally, the presentation of the derivation for 'r' is hampered by misaligned page breaks, separating key explanatory text and hindering comprehension. More significantly, while the paper thoroughly explores the technical capabilities of PriorGuide, it lacks a discussion of the broader implications of manipulating the prior distribution within a Bayesian framework. The authors appear to assume a familiarity with this interplay between prior, simulator, and the scientific endeavor, which may limit the accessibility and impact of the work for a wider audience.
>
> We appreciate your suggestions. In the revision, we have conducted a thorough proofread to improve the rigor of our narratives and writing. We agree that the manuscript assumes some familiarity with the role of priors in Bayesian inference and simulation-based approaches in computational sciences. In the revision, we have expanded the background discussion in Section 2 to better introduce the role of priors and the motivation for manipulating them to ensure the context of our work is clear and accessible to a broader audience.
>
> > **Questions**
>
> Thanks for the detailed comments. We have revised the manuscript to incorporate your feedback; we respond briefly below.
>
> > **Q1** page 4, the derivation of eq 6-9 is central to the paper, please try to rearrange so that the text in lines 216-220 is readily visible close the equations (e.g. by removing lines 191-195)
>
> We agree. We have adapted the text organization to ensure they are clearly presented.
>
> > **Q2** the code for the experiments was not found in the paper, i.e. through an anonymised git repo, this should be corrected for publication
>
> Please note  the code was provided in the supplementary materials in the initial submission; we will make it publicly available upon acceptance with a direct link to the repo in the paper.
>
> > **Q3** line 232: please do not use anthropomorphic language to describe model behavior "the model having seen few training examples", i.e. the model has consumed or been trained on
>
> > **Q4** line 234: "using standard OOD metrics" as such a standard exists only colloquially, please remove "standard"
>
> Thanks for these points. We have corrected them.
>
> > **Q5** equ 13: the text lacks a discussion if $K$ is a free parameter and how to choose it
>
> Yes we agree. We have added the discussion.
>
> > **Q6** line 274: "Notably, when p($\theta$) is uniform, r($\theta$) reduces to q($\theta$), which can be directly specified as a Gaussian mixture." perhaps also add hint why r reduces to q (due to using the expectation I guess)
>
> Thanks. We have added further explanation for this point.
>
> > **Q7** line 310: "To incorporate this with regular diffusion..." perhaps comment on the computational cost here already.
>
> Yes, we have added a comment about the computational cost.

---

> > ### Comment · Reviewer_kkwQ · 2025-11-27
> > **Thanks for acting upon my feedback**
> >
> > Thanks so much for addressing the concerns and recommendations brought forward. I am happy to keep my scoring of this excellent paper. I enjoyed the review process very much.

---

### Official Review · Reviewer_XCTR · 2025-10-31

**Soundness:** 3
**Presentation:** 4
**Contribution:** 3
**Rating:** 6
**Confidence:** 4

**Summary:**

**PriorGuide** targets a practical SBI need: adapting a trained diffusion-based posterior estimator to **new priors at test time** with **no retraining**. The paper derives a clean **score decomposition** in which the target posterior score equals the trained score plus a **prior-ratio guidance** term . To make this usable, the authors (i) use a *standard* Gaussian reverse-kernel approximation (via Tweedie’s formula) and (ii) fit a Gaussian mixture to the prior ratio, yielding a *novel* closed-form guidance update that plug directly into the diffusion sampler. Optional few-step Langevin corrections at low noise tighten asymptotics and expose a neat compute–accuracy knob alongside the diffusion steps. Empirically, PriorGuide improves both posterior and posterior-predictive metrics across several SBI tasks while keeping the implementation lightweight and training unchanged.

**Strengths:**

- **Well written easy to follow**:The derivation is easy to follow; the GMM prior-ratio leads to implementable, closed-form guidance. Propositions are sound and proven in the Appendix.
- **Good empirical results with clear trade-offs and ablations**: Consistent gains on posterior and predictive metrics; straightforward ablations over diffusion/Langevin steps make hyperparameter selection transparent. The paper is upfront about approximation choices and where Langevin steps matter, which helps practitioners reason about when to use the method.

**Weaknesses:**

- **Prior family diversity**: Main experiments emphasize changes among Gaussian priors (often **diagonal covariance**). This is required for fair ACE comparisons but does not demonstrate  the claimed method’s generality. A demonstration on **non-factorized** or more structured priors (complicated priors) would strengthen external validity. For example one use-case that requires flexible priors would be for "i.i.d" data which also can be handled by sequentially updating the "prior" with the "current posterior".

**Questions:**

1. **Non-factorized complex priors**: Have you tried dense-covariance Gaussians or general complicated priors? Have you tried inference on iid data using PriorGuide?
2. **Better Gaussian reverse Kernel approximations**: There exists better reverse kernel approximations (i.e. [1] and related). These usually are more costly i.e. require Jacobians or other terms. Would be interesting to see if these also help in that case but atleast should be discussed in the manuscript.
3. **Matrix calculations**: The authors state that a current limitation is that "matrix-operations  may-pose a scalability issue". A straight forward approach to avoid this is to constrain the involved covariance to be *diagonal* (or other approx.). After all a mixture of Gaussians with diagonal covariance is still a universal approximation family (just may needs more components). As the backward kernel approximation is by design also diagonal the whole Guidance terms are reduced to the diagonal. This hence makes *PriorGuideDiag*  an interesting alternative to the current approach.

[1] Boys, Benjamin, et al. "Tweedie moment projected diffusions for inverse problems." _arXiv preprint arXiv:2310.06721_ (2023).

---

> ### Author Response · Authors · 2025-11-25
> **Response to Reviewer XCTR (Part 1 of 2)**
>
> We thank the reviewer for their insightful and constructive feedback. We are particularly grateful for your suggestion to explore sequential Bayesian updating with i.i.d. data, which inspired a new experiment in our revision that highlights our method's flexibility. We also appreciate your recognition of the clarity of our theoretical derivations and the transparency of our experimental results.
>
> > **W1** Prior family diversity: Main experiments emphasize changes among Gaussian priors (often diagonal covariance). This is required for fair ACE comparisons but does not demonstrate the claimed method's generality. A demonstration on non-factorized or more structured priors (complicated priors) would strengthen external validity. For example one use-case that requires flexible priors would be for "i.i.d" data which also can be handled by sequentially updating the "prior" with the "current posterior".
> > **Q1** Non-factorized complex priors: Have you tried dense-covariance Gaussians or general complicated priors? Have you tried inference on iid data using PriorGuide?
>
> We would like to emphasize that our main experiments already address non-factorized, structurally complex priors. The mixture priors $q_\text{mixture}(\boldsymbol{\theta})$ in Section 4.2 and Table 1 are bimodal distributions representing competing hypotheses – arguably a challenging scenario due to their multimodality. PriorGuide substantially outperforms baselines on these benchmarks, demonstrating strong robustness to complex prior structure.
>
> Having said this, we agree that sequential updating with i.i.d. data is an interesting use case that would also naturally produce interesting non-Gaussian priors. To illustrate this capability, we have added a new experiment in Appendix D.2 using the Two Moons model. This experiment showcases the PriorGuide workflow for sequential Bayesian updating:
>
> * Simulate two observations based on a parameter $\boldsymbol{\theta}^*$.
> * Infer the posterior $p(\boldsymbol{\theta} | \mathbf{x}_1)$ from the first observation.
> * Use this posterior as the new target prior $q(\boldsymbol{\theta})$.
>   - *PriorGuide*: Fit a GMM to the ratio $q(\boldsymbol{\theta}) / p_\text{train}(\boldsymbol{\theta})$.
> * Infer the final posterior $p(\boldsymbol{\theta} | \mathbf{x}_1, \mathbf{x}_2)$ given the second observation $\mathbf{x}_2$).
>
> The results below, together with an example visualization in Appendix D.2 (Figure A2), illustrate how PriorGuide can handle *current posteriors* as complicated, data-driven *new priors* for sequential inference.
>
> Table 1: Posterior inference ($p(\boldsymbol{\theta} \mid \mathbf{x}_1, \mathbf{x}_2)$). Mean $\pm$ standard dev. over 5 independent training runs (10 randomly generated true parameters and observation sets). Significantly best results (Wilcoxon signed-rank test) in bold.
> | Dataset | Method | RMSE | C2ST | MMTV |
> | :--- | :--- | :---: | :---: | :---: |
> | Two Moons | Simformer | $0.63 \pm 0.34$ | $0.89 \pm 0.02$ | $0.66 \pm 0.07$ |
> | | ACE | $0.63 \pm 0.37$ | $0.94 \pm 0.03$ | $0.71 \pm 0.07$ |
> | | PriorGuide | $0.63 \pm 0.34$ | $\mathbf{0.60} \pm 0.07$ | $\mathbf{0.22} \pm 0.13$ |
>
> On posterior quality metrics (C2ST, MMTV), PriorGuide strongly outperforms the baselines. The point-wise RMSE result, which measures the error between the posterior and ground-truth parameter $\boldsymbol{\theta}^*$, is reported for consistency with the metrics shown in the paper; but note that RMSE is not meaningful when the posterior is multimodal like in this case (since the error is dominated by the regions of the posterior in the wrong mode; even though it is correct behavior for the posterior to be multimodal). Here ACE underperforms, likely due to the fact that this complex prior differs significantly from the distribution of priors (the *meta-prior*) it has been pretrained on, showing the importance of test-time prior adaptation.

---

> ### Author Response · Authors · 2025-11-25
> **Response to Reviewer XCTR (Part 2 of 2)**
>
> > **Q2** Better Gaussian reverse kernel approximations: There exists better reverse kernel approximations (i.e. [1] and related). These usually are more costly i.e. require Jacobians or other terms. Would be interesting to see if these also help in that case but at least should be discussed in the manuscript.
>
> We agree that it is a good idea to consider more advanced denoiser covariance approximations, such as the one in [1]. We added corresponding discussion in Section 3.3, with references to [1,2,3,4,5,6,7], and mentioned their investigation as future work in the Discussion. We found the simple diagonal approximation sufficient for our purposes, especially when combined with Langevin dynamics refinement. Utilizing advanced approximations that strike the best balance between computational cost and accuracy could definitely be interesting future work and potentially enable us to rely less on Langevin steps and speed up the method.
>
> > **Q3** Matrix calculations: The authors state that a current limitation is that "matrix-operations may-pose a scalability issue". A straight forward approach to avoid this is to constrain the involved covariance to be *diagonal* (or other approx). After all a mixture of Gaussians with diagonal covariance is still a universal approximation family (just needs more components). As the backward kernel approximation is by design also diagonal the whole Guidance terms are reduced to the diagonal. This hence makes *PriorGuideDiag* an interesting alternative to the current approach.
>
> Thank you for this insight. We are happy to confirm that the proposal was already employed by our configurations, where our Gaussian reverse kernel has diagonal covariances. We also explicitly constrained the GMM ratio approximation to use diagonal covariances (with a relatively large number of components to keep expressiveness). In our paper we use $K = 20$ components which are more than adequate, but we demonstrated in our ablation study (Appendix D.4) that increasing the number of GMM components up to $K = 200$, if needed, incurs negligible computational overhead. This empirically confirms that the diagonal approximation is a scalable technique that maintains expressiveness without sacrificing inference speed.
>
> **References:**
>
> [1] Boys et al. *Tweedie Moment Projected Diffusions for Inverse Problems*. TMLR, 2023.
>
> [2] Bao et al. *Analytic-DPM: an Analytic Estimate of the Optimal Reverse Variance in Diffusion Probabilistic Models*. ICLR 2022.
>
> [3] Peng et al. *Improving Diffusion Models for Inverse Problems Using Optimal Posterior Covariance*. ICML 2024.
>
> [4] Finzi et al. *User-defined event sampling and uncertainty quantification in diffusion models for physical dynamical systems*. ICML 2023.
>
> [5] Manor and Michaeli. *On the Posterior Distribution in Denoising: Application to Uncertainty Quantification*. ICLR 2024.
>
> [6] Rozet et al. *Learning diffusion priors from observations by expectation maximization*. NeurIPS 2024.
>
> [7] Rissanen et al. *Free Hunch: Denoiser Covariance Estimation for Diffusion Models Without Extra Costs*. ICLR 2025.

---

### Official Review · Reviewer_DTXg · 2025-11-01

**Soundness:** 3
**Presentation:** 3
**Contribution:** 3
**Rating:** 8
**Confidence:** 4

**Summary:**

The paper consideres an interesting problem of adapting a diffusion model pretrained to sample from a posterior when a different (unknown at the training state) prior model is adopted by the user. The proposed algorithm, PriorGuide, is able to correct biase of the pretrained diffision model during the sampling stage and providing a test time adaptive inference.

The guidance is intractable but is proximated using Gaussian mixture models.

The experiments show that on a wide variety of bechmark datasets, the proposed method outperforms similar diffusion methods with no-prior adaptation.

**Strengths:**

Overall, I think this paper makes a solid contribution to the SBI community.

The problem of changing priors is indeed common in many applications, and the ability to adapt a pre-trained posterior model without retraining the entire model is highly desirable and can make a lot of sense when the user's prior is different and comes in at a later stage.

Diffusion guidance is a well-studied technique in the generative model adaptation, so it is natural to leverage this approach to adapt SBI diffusion models. To my best knowledge, the proposed idea is novel.

The proposed method is mathematically sound, and its resulting formulation is natural, separating the pre-trained model from the guidance term (the expected prior shift).

Although the guidance term itself is intractable, authors show it can be effectively approximated.

**Weaknesses:**

My concerns are mostly on the literature review:

In the main text, the authors do not provide a comprehensive review or comparison of prior adaptation methods in the existing literature.  As briefly mentioned in the introduction and the experimental section, there are already methods that consider adaptive priors. Although the authors include a separate section in the appendix, a dedicated subsection in Section 2 would be appreciated, as readers would like to understand the specific limitations of these prior adaptation methods and how (or whether) they relate to the proposed guidance method.

In Section 2, recent progress (2022 ~ ) on applying diffusion models to SBI should also be comprehensively reviewed and cited as they provide the backdrop (the pre-trained posterior score) to the proposed method. To the best of my knowledge, the earliest attempt to use diffusion models in SBI is often attributed to Geffner et al. (2023), first published on arXiv in 2022. Simformer is a subsequent work built on Geffner et al. and others, introducing masks and a transformer architecture to handle missing observations and more versatile inference tasks.

**Questions:**

None.

---

> ### Author Response · Authors · 2025-11-25
> **Response to Reviewer DTXg**
>
> We thank the reviewer for their encouraging assessment and for recognizing PriorGuide as a novel and solid contribution to the SBI community. We appreciate your validation of our method’s mathematical soundness and its practical relevance for handling changing priors.
>
> *Note:* All revisions mentioned below are available in the uploaded revised paper, highlighted in blue text.
>
> > **W1** In the main text, the authors do not provide a comprehensive review or comparison of prior adaptation methods in the existing literature. As briefly mentioned in the introduction and the experimental section, there are already methods that consider adaptive priors. Although the authors include a separate section in the appendix, a dedicated subsection in Section 2 would be appreciated, as readers would like to understand the specific limitations of these prior adaptation methods and how (or whether) they relate to the proposed guidance method.
>
> Thanks for the suggestion. We completely agree with this, which will greatly improve readability. In the revision, we have moved the relevant text from Appendix A.1, slightly reformulated, to a new dedicated subsection in the main text (Section 2.3, “Prior adaptation in amortized SBI”). To avoid redundancy, we slightly compressed the old paragraph in the appendix, now referring to the main text.
>
> > **W2** In Section 2, recent progress (2022 ~ ) on applying diffusion models to SBI should also be comprehensively reviewed and cited as they provide the backdrop (the pre-trained posterior score) to the proposed method. To the best of my knowledge, the earliest attempt to use diffusion models in SBI is often attributed to Geffner et al. (2023), first published on arXiv in 2022. Simformer is a subsequent work built on Geffner et al. and others, introducing masks and a transformer architecture to handle missing observations and more versatile inference tasks.
>
> Thanks for pointing this out. You are absolutely correct. In the revised Section 2.2, we have expanded our discussion to include foundational works before Simformer such as [1], explicitly acknowledging it as one of the earliest applications of diffusion models to SBI.
>
> **References:**
>
> [1] Geffner et al. *Compositional score modeling for simulation-based inference*. ICML 2023.

---

### Official Review · Reviewer_HVmz · 2025-11-05

**Soundness:** 3
**Presentation:** 3
**Contribution:** 2
**Rating:** 4
**Confidence:** 4

**Summary:**

This paper considers the amortized simulator-based inference framework and one of its major limitations, i.e., prior distributions used to generate model parameters during training. The authors propose a novel method, PriorGuide, which leverages a guidance approximation for flexible test-time prior adaptation. The method was evaluated on a set of benchmarks. The most significant strength is requiring no retraining.

**Strengths:**

The paper has a few strengths:
1. The "prior-rigidity" of amortized SBI models is a well-known and significant bottleneck for scientific application. This paper offers a direct, "plug-and-play" solution that enables, for the first time, post-hoc prior sensitivity analysis and the incorporation of new expert knowledge using expensive, pre-trained models.
2. The method is built on a sound mathematical foundation. The derivation of the target posterior as a tilted distribution (Proposition 1) is clear and correct.
3. The experiments are convincing. The paper correctly identifies that its method is an approximate sampler and intelligently introduces Langevin correctors to manage the accuracy-compute trade-off.

**Weaknesses:**

However, despite it’s strengths, there are also weaknesses:
1. The entire method hinges on a critical, and likely fragile, approximation: fitting the prior ratio $r(\theta) = q(\theta) / p_{train}(\theta)$ with a GMM. The paper understates the difficulty of this step. This is a density-ratio estimation problem, which is notoriously difficult, especially as the dimension of $\theta$ increases. The paper's solution (a gradient-based $L_2$ fit) is a heuristic that is not guaranteed to be stable or accurate, particularly if $p_{train}(\theta)$ is non-trivial.
2. The paper employs two major approximations: the GMM for the ratio and a simple diagonal Gaussian for the reverse kernel. It is the combination of these (especially the latter) that biases the sampler and requires the heavy use of Langevin correction steps to achieve good results. This means the method, while "retrain-free," has a high and non-trivial per-sample inference cost.

**Questions:**

Besides the Weaknesses, I have also a question:
1. The GMM approximation of the prior ratio $r(\theta)$ appears to be the most critical (and fragile) part of the pipeline. How does the $L_2$ fitting procedure scale with the dimensionality of $\theta$? Have you considered replacing this heuristic fit with more robust, modern density-ratio estimators (e.g., flow-based, MINE, etc.)?

---

> ### Author Response · Authors · 2025-11-25
> **Response to Reviewer HVmz (Part 1 of 2)**
>
> We thank the reviewer for their thorough and supportive review. We are grateful for your appreciation of the sound mathematical foundation and convincing empirical evaluations of our work.
>
> *Note:* All revisions mentioned below are available in the uploaded revised paper, highlighted in blue text.
>
> > **W1.** The entire method hinges on a critical, and likely fragile, approximation: fitting the prior ratio $r(\theta) = q(\theta) / p_{\text{train}}(\theta)$ with a GMM. The paper undertakes the difficulty of this step. This is a density-ratio estimation problem, which is notoriously difficult, especially as the dimension of $\theta$ increases. The paper's solution (a gradient-based $L_2$ fit) is a heuristic that is not guaranteed to be stable or accurate, particularly if $p_{\text{train}}(\theta)$ is non-trivial.
>
> We thank the reviewer for raising this point. You are correct that estimating $r(\boldsymbol{\theta}) = q(\boldsymbol{\theta}) / p_{\text{train}}(\boldsymbol{\theta})$ is fragile when $q$ and $p_{\text{train}}$ are unknown distributions accessed only via finite samples (the standard density-ratio estimation problem). However, we would like to stress that *we are not performing density-ratio estimation* in the traditional sense. In our setting, both the new prior $q$ and the training prior $p_{\text{train}}$ have tractable log-densities, since the new prior is specified by a practitioner and the training prior often takes a simple form (e.g., uniform over a range, Gaussian). Just to clarify: while the data may well take complex shapes, the priors over *model parameters* in statistical modelling often encode simple relationships that can be represented by parametric model families. The ratio is therefore a known and deterministic function that can be evaluated pointwise exactly.
>
> Consequently, fitting a GMM to the ratio is a deterministic function approximation problem. The availability of exact gradients transforms this from a "fragile heuristic" into a standard optimization task. Please see below our response to Q1 for empirical proof.
>
> We included a brief clarification in the paper in Section 3.2: “Crucially, since the densities of both $p_{\text{train}}$ and $q$ are analytically known, fitting the ratio avoids the instability and high variance inherent to statistical density-ratio estimation from finite samples. We provide a straightforward gradient-based fitting procedure in Appendix A.2.
>
> > **W2.**  The paper employs two major approximations: the GMM for the ratio and a simple diagonal Gaussian for the reverse kernel. It is the combination of these (especially the latter) that biases the sampler and requires the heavy use of Langevin correction steps to achieve good results. This means the method, while "retrain-free," has a high and non-trivial per-sample inference cost.
>
> We appreciate the reviewer’s scrutiny regarding the approximations used. We acknowledge the approximations employed in our framework, but would like to emphasize that the reverse kernel approximation is a standard approach in diffusion guidance [1, 2] that becomes accurate at low noise levels, and the Langevin dynamics provides stronger theoretical guarantees. Regarding additional inference cost from Langevin steps, this is a reasonable concern that we had ourselves, and fortunately it is not necessarily the case as shown in Figure 3 in the main paper: namely, we can keep the total number of (neural) function evaluations (NFEs) constant and get significant improvements by trading off standard diffusion steps for Langevin steps.
>
> Overall, we agree that PriorGuide introduces additional inference cost at test time. However, it is still much faster than the alternative: generating new simulations and retraining the whole network.

---

> ### Author Response · Authors · 2025-11-25
> **Response to Reviewer HVmz (Part 2 of 2)**
>
> > **Q1** How does the $L_2$ fitting procedure scale with the dimensionality of $\theta$? Have you considered replacing this heuristic fit with more robust, modern density-ratio estimators (e.g., flow-based, MINE, etc.)?
>
> As we pointed out earlier, density-ratio estimators (flow-based, MINE, etc.) are ill-suited for this specific task, since we do not estimate the ratio from samples. Our gradient-based fitting procedure scales well because the ratio function $r(\boldsymbol{\theta})$ typically behaves predictably and smoothly, being the ratio of two smooth distributions.
>
> To quantitatively verify the accuracy of our procedure, we considered the 10D *Gaussian Linear* problem from the paper. We performed our fitting procedure across 100 randomly generated target Gaussian priors and computed the weighted $L_2$ error between the log ratios using 10,000 Monte Carlo samples per fit. The median $L_2$ error between the fitted GMM log density and the true analytic log ratio was 0.023 [0.002, 0.016, 0.032, 0.186] (2.5th-25th-75th-97.5th percentiles), indicating that most fits are extremely accurate (note that 0.023 error in log space corresponds to ~2% error on the ratio, since $e^x \approx 1+x$ for small $x$). In practice, the few “bad fits” can be avoided by running multiple optimizations with different initializations, which is the typical approach when solving a global nonconvex optimization problem. We added this analysis to Appendix D.5 in the revised paper.
>
> As an example to show the accuracy of such fits, we have included a new visualization in Appendix D.5 (Figure A4) of the revision. This figure compares the conditionals (on a fixed parameter point) of the fitted GMM ratio against the true ratio in 10D space. This representative plot shows that the fitted GMM accurately recovers the ratio's geometry even in high-dimensional settings. Beyond these examples, the general validity of our ratio-fitting procedure is demonstrated more indirectly by the efficacy of our method in our experiments.
>
> **References:**
>
> [1] Ho, et al. *Video diffusion models*. NeurIPS 2022.
>
> [2] Song et al. *Pseudoinverse-guided diffusion models for inverse problems*. ICLR 2023.

---

### Author Response · Authors · 2025-11-25
**Global Response: Summary of Key Revisions**

We thank all the reviewers for their constructive feedback. We value your suggestions and have uploaded the revised manuscript incorporating the feedback, including new experimental visualizations, expanded background discussions, and clearer practical guidelines.

Summary of key revisions:
- Prior ratio approximation in high-dimensional space (reviewer HVmz): To address questions regarding the scalability of the GMM ratio approximation, we added a quantitative evaluation in Appendix D.5. We also added a new example visualization in Appendix D.5 (Figure A4), comparing a typical fitted GMM ratio against the true analytic ratio in a 10D parameter space. These results show that our gradient-based ratio-fitting procedure produces well-matching approximations even in higher dimensions.
- Sequential Bayesian inference (reviewer XCTR): To demonstrate PriorGuide’s applicability to complex, data-derived priors, we added a new experiment in Appendix D.2 (Figure A2). We perform sequential Bayesian updating on the Two Moons model, where an intermediate posterior is fitted and used as a complicated, new target prior for the subsequent update.
- Expanded background discussion (reviewer kkwQ, DTXg): We revised Section 2 to better introduce the background of our work. This includes an expanded introduction to the role of priors in Bayesian inference and scientific applications, highlighting foundational diffusion-based SBI works such as [1], and a dedicated discussion about prior adaptation in amortized inference in Section 2.3.
- Practical guidelines (reviewer zfhL): We added a summary of practical guidelines in Section 5 (Discussion), explicitly outlining when PriorGuide is preferred over baselines (e.g., in moderate-to-high dimensions, with expensive simulators, or for complex priors).

We believe these revisions significantly strengthen the manuscript and hope they address the reviewers' concerns. All revisions are marked in the uploaded revised paper in blue text.

**References:**

[1] Geffner et al. *Compositional score modeling for simulation-based inference*. ICML 2023.

---

### Author Response · Authors · 2025-12-01
**Rebuttal Summary**

We welcome the new Area Chair. Given the exceptional circumstances, we provide a summary to preserve the discussion record and assist your assessment.

### Reviewers who confirmed satisfaction before the freeze:

- **Reviewer zfhL** (score 6 $\rightarrow$ 8, conf. 4): Confirmed our rebuttal and revision were satisfactory, stating: "*I agree that this is valuable contribution to the conference and I will adapt my score accordingly.*"
- **Reviewer kkwQ** (score 10, conf. 5): Confirmed our revision fully addressed their feedback and maintained strong support.

### Reviewers who had not yet responded (HVmz, XCTR, DTXg):

We addressed their primary concerns through targeted revisions:
- **HVmz** (score 4, conf. 4): Two main concerns. First ("*The entire method hinges on a critical, and likely fragile, approximation*") was a misunderstanding: the reviewer believed that our algorithm requires traditional density ratio estimation from samples (which indeed is “*notoriously difficult*” and fragile), even recommending dedicated sample-based estimators (“*more robust, modern density-ratio estimators (e.g., flow-based, MINE, etc.)*”). We clarified here and in the revised manuscript (Section 3.2) that $p_\text{train}$ and $q$ are *known densities*, reducing the problem to standard nonconvex optimization. As further empirical support, we added quantitative evaluation across 100 random priors in 10D showing very low reconstruction error (median L2 error of 0.023; Appendix D.5, Figure A4). Second, regarding inference cost from Langevin corrections: we showed in Figure 3 that total NFEs can remain constant by trading diffusion steps for Langevin steps, while still being far cheaper than retraining.
- **XCTR** (score 6, conf. 4): Requested demonstration on complex, data-derived priors such as in sequential Bayesian updating. We added a new sequential Bayesian inference experiment on Two Moons (Appendix D.2, Figure A2), showing strong performance.
- **DTXg** (score 8, conf. 4): Requested expanded background discussion. We added a paragraph on priors in Section 2, added Section 2.3 on prior adaptation in amortized SBI, and cited additional foundational diffusion-based SBI works.

All revisions are marked in blue in the uploaded manuscript. We hope this summary assists the AC in evaluating our work based on the full extent of revisions and the positive engagement achieved during the rebuttal period.

---

### Meta-Review · Area_Chair_Uu62 · 2025-12-31

**Summary:**

The authors propose a significant improvement for Simulation-based inference (SBI). The main advantage of the proposed model is that it does not require retraining at test time with new priors. All reviewers agree that this represents a significant improvement over the existing results, and all but one are favourable to acceptance. Using the term $p_{train}(\theta)$ may lead to misunderstanding, as it may evoke in some readers the notion of $p_X(x)$, which may not be easy to approximate. Also, for high dimensions, there might be areas of low probability in $p_{train}(\theta)$ that can create issues with convergence. In all, a solid contribution. I suggest that the authors improve the paper as indicated by the reviewers.

**Reviewer Concerns:**

I do not have any significant concerns. Reviewer HVmz has a point about p_train, but it is well addressed by the authors.

**Reviewer Scores:**

I would expect reviewer HVmz to increase their score. The others were already positive, and I would not expect them to improve them. Although the two reviewers with six might have. It is in general a solid contribution.

---

### Decision · Program_Chairs · 2026-01-26

Accept (Poster)